# SARS-CoV-2 RNAemia and proteomic trajectories inform prognostication in COVID-19 patients admitted to intensive care

Clemens Gutmann [1,16], Kaloyan Takov [1,16], Sean A. Burnap [1,16], Bhawana Singh [1,16], Hashim Ali[1,16], Konstantinos Theofilatos [1], Ella Reed[1], Maria Hasman[1], Adam Nabeebaccus[1,2], Matthew Fish [3,4], Mark JW. McPhail[2,5,6], Kevin O'Gallagher[1,2], Lukas E. Schmidt [1], Christian Cassel[1], Marieke Rienks [1], Xiaoke Yin [1], Georg Auzinger[2], Salvatore Napoli[5], Salma F. Mujib[6], Francesca Trovato[2,5,6], Barnaby Sanderson [4], Blair Merrick [7], Umar Niazi [8], Mansoor Saqi[8], Konstantina Dimitrakopoulou[8], Rafael Fernández-Leiro [9], Silke Braun [10], Romy Kronstein-Wiedemann[11], Katie J. Doores [3], Jonathan D. Edgeworth[3,7], Ajay M. Shah[1,2], Stefan R. Bornstein [12,13], Torsten Tonn [11,14], Adrian C. Hayday [3,15], Mauro Giacca [1], Manu Shankar-Hari [3,4 ✉] & Manuel Mayr [1,12 ✉]

Prognostic characteristics inform risk stratification in intensive care unit (ICU) patients with coronavirus disease 2019 (COVID-19). We obtained blood samples ($n = 474$) from hospitalized COVID-19 patients ($n = 123$), non-COVID-19 ICU sepsis patients ($n = 25$) and healthy controls ($n = 30$). Severe acute respiratory syndrome coronavirus 2 (SARS-CoV-2) RNA was detected in plasma or serum (RNAemia) of COVID-19 ICU patients when neutralizing antibody response was low. RNAemia is associated with higher 28-day ICU mortality (hazard ratio [HR], 1.84 [95% CI, 1.22–2.77] adjusted for age and sex). RNAemia is comparable in performance to the best protein predictors. Mannose binding lectin 2 and pentraxin-3 (PTX3), two activators of the complement pathway of the innate immune system, are positively associated with mortality. Machine learning identified 'Age, RNAemia' and 'Age, PTX3' as the best binary signatures associated with 28-day ICU mortality. In longitudinal comparisons, COVID-19 ICU patients have a distinct proteomic trajectory associated with mortality, with recovery of many liver-derived proteins indicating survival. Finally, proteins of the complement system and galectin-3-binding protein (LGALS3BP) are identified as interaction partners of SARS-CoV-2 spike glycoprotein. LGALS3BP overexpression inhibits spike-pseudoparticle uptake and spike-induced cell-cell fusion in vitro.

A full list of author affiliations appears at the end of the paper.

Coronavirus disease 2019 (COVID-19) caused by the severe acute respiratory syndrome coronavirus 2 (SARS-CoV-2; a single-stranded RNA virus) poses an unprecedented challenge to health care systems globally. It is increasingly apparent that conventional prognostic scores for patients admitted to intensive care units (ICUs) such as the Acute Physiology and Chronic Health Evaluation (APACHE II) score[1] and Sequential Organ Failure Assessment (SOFA) score[2], are unsuitable for outcome prediction in COVID-19 ICU patients[3–6].

In this context, circulating SARS-CoV-2 RNA (RNAemia) has been highlighted as a promising prognostic marker in hospitalized COVID-19 patients, as it is associated with disease severity[7] and mortality[8–10], with an estimated prevalence of 10% (95% CI: 5–18%, random-effects model)[7]. Further, we hypothesized that the acute and profound alterations in the innate and adaptive immune system in COVID-19 patients[3,11–13], especially in RNAemic patients[14–18], will be accompanied by marked changes in the circulating proteome and interactome that will highlight mechanistically relevant signatures and trajectories when compared to non-COVID-19 sepsis and healthy controls. Thus far, proteomics studies have focused on the determination of protein markers of COVID-19 severity[19–22], often with healthy individuals as a comparator, but have not assessed the longitudinal relationship between proteomic changes, RNAemia, and 28-day ICU mortality.

In this study, we assessed RNAemia, antibody response against SARS-CoV-2, and proteomic profiles in serial blood samples from COVID-19 patients admitted to two ICUs. Controls included hospitalized, non-ICU patients with and without COVID-19 as well as SARS-CoV-2-negative ICU sepsis patients. Sepsis is defined as organ dysfunction caused by a dysregulated host response to infection[23,24]. As SARS-CoV-2 infection causes organ dysfunction (pulmonary and extrapulmonary)[25,26] and there is overlap in immunological changes between SARS-CoV-2 infection and sepsis[11,27], this formed the rationale for using SARS-CoV-2-negative ICU sepsis patients as additional comparators. We compared the associations of RNAemia and protein measurements with 28-day ICU mortality, including established protein markers of acute respiratory distress syndrome (ARDS), i.e., the receptor for advanced glycation end-products (RAGE)[28–30], and prognosis in ICU patients with sepsis, i.e., pentraxin-3 (PTX3)[31–34]. In the context of RNAemia, we explored the plasma protein interactions with the SARS-CoV-2 spike glycoprotein, identifying galectin-3-binding protein (LGALS3BP) as a novel binding partner with antiviral activities.

## Results

**Demographics and clinical characteristics of COVID-19 patients.** 474 blood samples were available for analysis (Fig. 1, Supplementary Fig. 1): 295 longitudinal samples from ICU patients with COVID-19 admitted to two university hospitals (GSTT; $n = 62$ and KCH; $n = 16$) and samples from hospitalized, non-ICU COVID-19 patients for comparison ($n = 45$); ICU and non-ICU patients without COVID-19 served as controls ($n = 55$). The baseline clinical characteristics of all COVID-19 ICU patients are shown in Supplementary Table 1. The primary outcome measure was defined as mortality 28 days after ICU admission. As expected[35], non-survivors (23%) were older than survivors ($P = 0.0004$). COVID-19 patients admitted to ICU were predominantly males (72%). All other characteristics, including common comorbidities, the time from symptom onset to ICU admission, APACHE II score, and SOFA score, were similar between ICU survivors and non-survivors. The mortality rate in COVID-19 ICU patients was twice as high as in hospitalized,

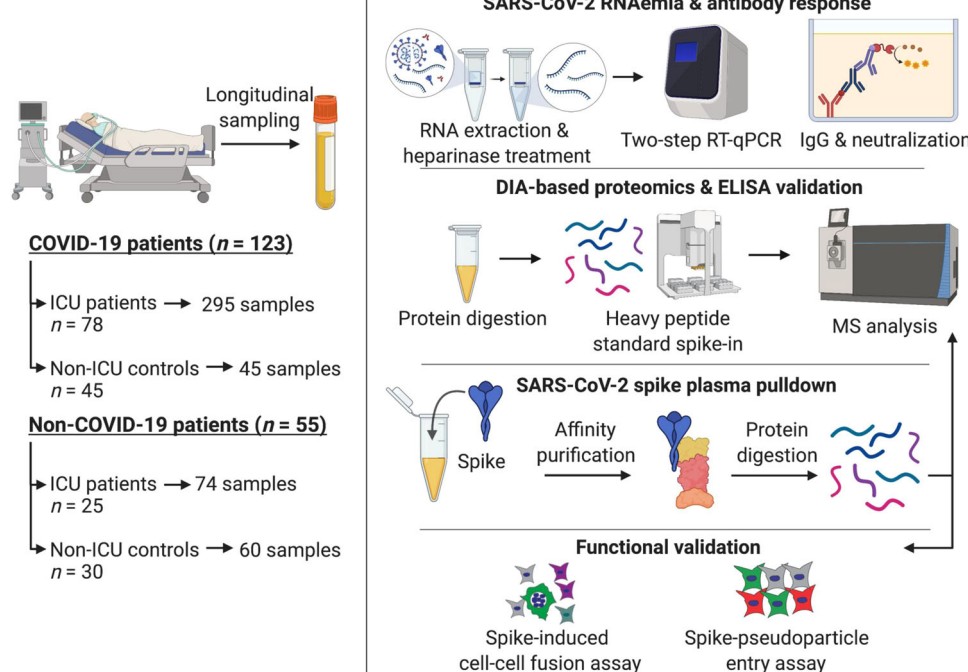

**Fig. 1 Schematic of study design.** Plasma and serum samples were obtained from multiple patient cohorts across two UK-based university hospitals, including 123 COVID-19 patients: 78 SARS-CoV-2 positive patients in ICU were sampled at multiple time points over a 2-week period and compared to hospitalized non-ICU SARS-CoV-2 positive patients ($n = 45$). We used non-COVID-19 ICU patients ($n = 25$) and patients before undergoing elective cardiac surgery ($n = 30$) as controls. Patient samples were assessed for SARS-CoV-2 RNAemia, antibody responses, and protein changes in the circulation by data-independent acquisition (DIA) mass spectrometry (MS) analysis. Plasma protein interactions with SARS-CoV-2 spike glycoprotein were determined using a pulldown assay followed by data-dependent acquisition (DDA) MS analysis. Functional effects of LGALS3BP were assessed in two assays: SARS-CoV-2 spike-mediated cell-cell fusion (syncytia formation) and cell entry through SARS-CoV-2 spike pseudoparticle assays.

non-ICU COVID-19 patients (23% vs. 11%; Supplementary Table 2).

**Frequency of SARS-CoV-2 RNAemia and association with mortality in COVID-19 ICU patients**. The presence of circulating viral RNA was analyzed by reverse transcription-quantitative polymerase chain reaction (RT-qPCR). Serum (GSTT; $n = 62$) and plasma (KCH; $n = 16$) samples were collected within 24 h of admission to ICU with COVID-19 and thereafter during week 1, week 2, and again before discharge. Out of 78, 18 (23%) COVID-19 ICU patients had detectable RNAemia within the first 6 days of admission to ICU (Supplementary Table 1). RNAemia was more common early after symptom onset (Supplementary Fig. 2). RNAemia within 6 days of admission to ICU was detectable in 56% of non-survivors but only in 13% of survivors ($P = 0.0006$, Supplementary Table 1). RNAemia was associated with a higher risk of 28-day mortality (hazard ratio [HR], 2.05 [95% CI: 1.38–3.04]), that was comparable to age (2.89 [1.66–5.03] Fig. 2a) and maintained after correction for age and sex (HR, 1.84 [95% CI: 1.22–2.77], Fig. 2b). In comparison, only 2 out of 45 (4%) non-ICU COVID-19 patients tested positive for RNAemia upon hospitalization (Supplementary Table 2). General demographics and baseline clinical characteristics of COVID-19 patients with and without RNAemia in the first 6 days of admission to ICU are presented in Supplementary Table 3. Hypertension ($r = 0.33$, $P = 0.003$), bilirubin ($r = 0.32$, $P = 0.005$), respiration rate ($r = 0.27$, $P = 0.018$), and elevated potassium levels ($r = 0.26$, $P = 0.023$) were positively correlated to RNAemia, while monocyte counts were inversely correlated ($r = -0.23$, $P = 0.047$, Fig. 2c). Hierarchical clustering analysis of all clinical variables and RNAemia is presented in Supplementary Fig. 3. To confirm the specificity of our RT-qPCR assay, we measured SARS-CoV-2 RNAemia in 134 plasma samples from 55 non-COVID-19 patients, all of which tested negative (Supplementary Table 4).

**Humoral immune response during SARS-CoV-2 RNAemia**. In COVID-19 ICU patients with detailed information on the days, post-onset of symptoms (POS, $n = 70$), IgG antibodies to the S1 domain of SARS-CoV-2 spike glycoprotein and SARS-CoV-2 neutralizing capacity were measured by ELISA and Surrogate Virus Neutralization Test[36], respectively. The latter test evaluates the inhibition of binding of the receptor-binding domain (RBD) of SARS-CoV-2 spike to ACE2. COVID-19 ICU patients who tested positive ($n = 15$) or negative ($n = 55$) for RNAemia within the first six days in ICU showed no difference in their strong IgG response to SARS-CoV-2 S1 or in their neutralization capacity (Fig. 2d). However, when individual samples ($n = 232$) were compared, RNAemia positive samples ($n = 29$) had lower anti-SARS-CoV-2 spike IgG levels and lower SARS-CoV-2 neutralization capacity (Fig. 2e).

**Plasma proteome alterations in COVID-19 ICU patients**. To capture the host response of COVID-19 ICU patients, we interrogated their plasma proteome. Baseline plasma samples from COVID-19 ICU patients (KCH cohort, $n = 12$) were compared to plasma samples from COVID-19-negative sepsis ICU patients (sepsis, $n = 12$) and patients prior to undergoing elective cardiac surgery (controls, $n = 30$) (Supplementary Table 4). The plasma proteome was quantified by a data-independent acquisition–mass spectrometry (DIA–MS) approach, using authentic heavy peptide standards representing 500 proteins[37], revealing 100 significantly altered proteins across the three patient groups ($q < 0.05$) (Fig. 3a). Hierarchical cluster analysis highlighted a cluster of 47 plasma proteins enriched in COVID-19, including members of the

complement cascade, as well as proteins involved in platelet degranulation, the acute phase response, and coagulation (Fig. 3a, b).

Of the 100 circulating proteins altered across control, sepsis ICU, and COVID-19 ICU patients, 29 overlapped with previous proteomic reports identifying markers of COVID-19 severity[19,20] (Supplementary Fig. 4). However, only a few were associated with 28-day mortality, as determined through DIA–MS analysis of baseline serum samples obtained from a larger COVID-19 ICU patient cohort (GSTT, $n = 62$) (Fig. 3c). Complement factor B (CFB), carboxypeptidase N (CPN1), and alpha-1-antichymotrypsin (SERPINA3) were all negatively associated with outcome but none of these three associations remained significant after correcting for multiple testing. An independent, publicly available dataset utilizing proximity-extension assays in plasma (Olink Explore 1536, $n = 264$ survivors, $n = 42$ non-survivors, Fig. 4a, Supplementary Table 5)[38] also confirmed the lack of outcome association for three other proteins identified as markers of COVID-19 severity in previous proteomics studies[19,20]: lipopolysaccharide-binding protein, CD14, and inter-alpha-trypsin inhibitor heavy chain H3 (ITIH3) (Fig. 3c).

Protein changes that emerged as significantly associated with mortality in ICU patients but have not been previously linked to the severity of COVID-19, included an elevation of mannose binding lectin 2 (MBL2) and reductions in protein C (PROC), plasminogen (PLG), coagulation factor 7 (F7) and vitamin D-binding protein (GC) (Fig. 3d). A correlation matrix of clinical variables and proteins associated with COVID-19 severity (Fig. 3c) and outcome (Fig. 3c, d) measured in serum of the GSTT cohort is presented in Supplementary Fig. 5.

**Predictors of 28-day mortality in COVID-19 patients identified by machine learning**. Next, we compared the predictive performance of RNAemia against protein markers and clinical characteristics for 28-day mortality in ICU patients. In the external validation cohort of hospitalized COVID-19 patients described above ($n = 264$ survivors; $n = 42$ non-survivors)[38], PROC and F7 were the only proteins associated with 28-day mortality in our DIA–MS data (Fig. 3c, d) that were also measured by Olink proximity-extension assays[38]. Reduced PROC and F7 were confirmed to be associated with 28-day mortality (Fig. 4a, Supplementary Table 5)[38]. MBL2 was not part of the Olink panel (Explore 1536) but MBL2 is known to form complexes with PTX3[39]. Interestingly, PTX3 emerged as one of the proteins most strongly associated with mortality among 1472 unique proteins measured in the external validation, outperforming most measured cytokines and chemokines, showing a larger fold change than PROC or F7 (Fig. 4a, Supplementary Table 5)[38]. PTX3, a protein we and others have previously highlighted as a prognostic marker in ICU patients with sepsis[31–34], also positively associated with COVID-19 mortality in our ICU cohort when measured by ELISA (Fig. 4b). In contrast, RAGE, an established protein marker of ARDS[28–30], remained unaffected by SARS-CoV-2 RNAemia and mortality (Fig. 4c, Supplementary Fig. 6a). PTX3 forms multimers that accumulated in the high molecular weight fraction after high-performance size-exclusion chromatography of plasma (Fig. 4d). PTX3 multimers correlated to neutrophil-related proteins such as S100A8/A9, defensin (DEFA1), and SERPINA3 as well as to monocyte/macrophage-related markers such as CD14[40] (Fig. 4e). A machine learning-based approach was adopted to determine the best prognostic markers (Supplementary Fig. 7). Based on statistical significance ($P < 0.05$, Fig. 4b, Supplementary Table 1), age, RNAemia, urea, and PTX3 were shortlisted as singleton markers. In non-survivors, PTX3 was elevated (median circulating levels, 4.93 ng/ml [IQR: 2.78–6.20])

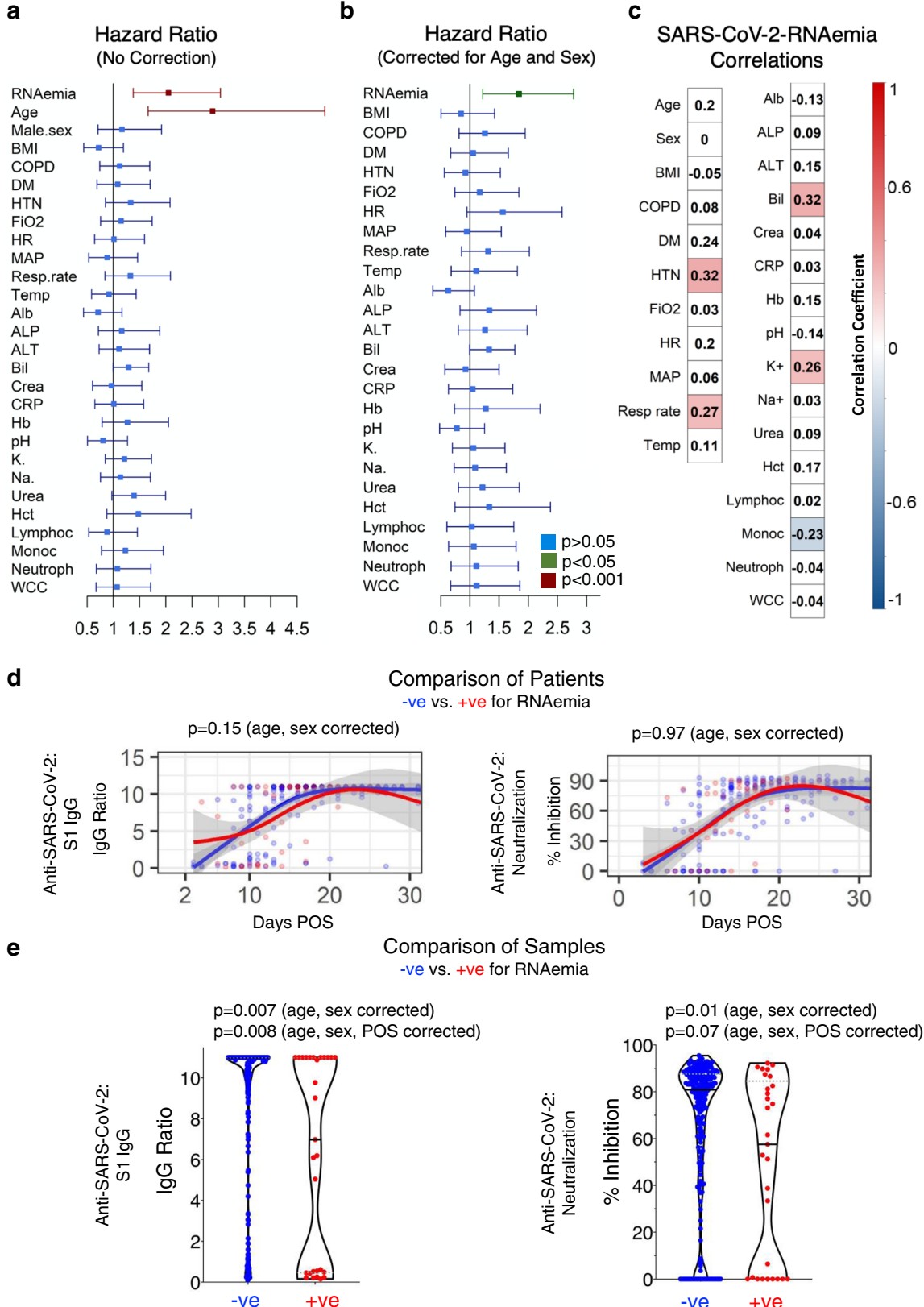

compared to survivors (median circulating levels, 2.16 ng/ml [IQR: 1.41–3.64], $P = 0.018$, adjusted for age and sex) (Fig. 4b). Although RNAemia surfaced as the best singleton predictor, its sensitivity was low (Supplementary Table 6). Age was the next best singleton predictor, however, its positive predictive value (PPV) of 44% demonstrates low probability confidence in

predicting mortality. When binary combinations were analyzed, two signatures emerged with comparable classification performance: "Age, RNAemia" and "Age, PTX3" (Supplementary Table 6). Although the triplet combination of age, $FiO_2$, and RNAemia achieved a ROC of ~86% with a sensitivity of 72.22% and a specificity of 88.33% (Supplementary Table 6), the gain in

**Fig. 2 SARS-CoV-2 RNAemia and the humoral immune response. a** Unadjusted hazard ratios with 95% confidence interval (CI) based on two ICU patient cohorts (n = 60 survivors and n = 18 non-survivors, KCH and GSTT). Green indicates P value < 0.05, maroon indicates P value < 0.001 and blue indicates P value > 0.05. **b**, Hazard ratios with 95% CI after adjustment for age and sex (n = 60 survivors and n = 18 non-survivors, KCH and GSTT). **c** Association of SARS-CoV-2 RNAemia with binary variables (Cohen's Kappa correlation) and continuous variables (point-biserial correlation). Red indicates positive and blue negative correlation with P value < 0.05. *Abbreviations*: Alb albumin, ALP alkaline phosphatase, ALT alanine aminotransferase, Bil bilirubin, COPD chronic obstructive pulmonary disease, Crea creatinine, CRP C-reactive protein, DM diabetes, Hct hematocrit, Hb hemoglobin, HR heart rate, HTN hypertension, Lymphoc lymphocytes, MAP mean arterial pressure, Monoc monocytes, Neutroph neutrophils, $K^+$ potassium, Resp. rate respiratory rate, Na $^+$ sodium, Temp body temperature, WCC white cell count. **d** Anti-SARS-CoV-2 spike IgG and anti-SARS-CoV-2 neutralization response based on days post-onset of symptoms (POS) in patients who tested positive (red) or negative (blue) for plasma/serum SARS-CoV-2 RNA within the first 6 ICU days (261 samples from n = 55 RNAemia negative and n = 15 RNAemia positive patients). Lines show fitted generalized additive models (GAM) with gray bands indicating the 95% CI, correcting for age and sex. **e** Anti-SARS-CoV-2 spike IgG levels and anti-SARS-CoV-2 neutralization capacity in individual samples negative (232 samples) or positive (29 samples) for SARS-CoV-2 RNA (n = 70 patients). Lines inside violin plots show median (continuous line) and interquartile range (dotted lines). Significance was determined through a Mann–Whitney U test. P values are corrected for age, sex, and days POS. All statistical analyses are two-tailed.

PPV was nominal with no uplift in specificity when compared to "Age, RNAemia", suggesting the binary combination to be an optimal signature to choose. The technical validation of our "Age, RNAemia" model was undertaken using a permutation test for statistical significance of the classifier performance (Supplementary Fig. 8a); and stability of feature importance in an alternate machine learning feature ranking model, i.e., random forest with resampling (Supplementary Fig. 8b). Kaplan–Meier plots (Fig. 5) also illustrate that the binary combinations "Age, RNAemia" (P < 0.0001) and "Age, PTX3" (P < 0.0001) provide improved stratification.

**Longitudinal protein associations with SARS-CoV-2 RNAemia and clinical improvement.** To explore whether the association of RNAemia with 28-day mortality may reflect distinct pathological processes, we identified proteins that associate with RNAemia at baseline (Fig. 6a) and over time (Fig. 6b). Nine proteins were significantly associated with RNAemia at baseline which included an increase in plasma protease C1 inhibitor (SERPING1) and complement component C4A (C4A); paralleled by a reduction in VE-cadherin (CDH5) and CFH-related protein 1 (CFHR1) (Fig. 6a). A correlation matrix of clinical variables and proteins associated with RNAemia or outcome is presented in Supplementary Fig. 5. In longitudinal serum samples from the GSTT cohort (baseline, week 1 and week 2; n = 47), a greater increase of polymeric immunoglobulin receptor (PIGR) was observed in RNAemia positive, compared to RNAemia negative ICU patients (Fig. 6b). In contrast, plasma kallikrein (KLKB1) levels significantly increased over time but tended to be higher in RNAemia negative ICU patients (Fig. 6b).

Hierarchical cluster analysis upon significantly changing serum proteins over the two-week period (baseline, week 1 and week 2, n = 47) revealed four distinct protein clusters (Fig. 6c), which were annotated by gene ontology enrichment analysis. Alterations in PIGR correlated closely with neutrophil degranulation proteins such as S100A8 and S100A9 (Fig. 6c, Cluster 2), while KLKB1 kinetics followed members of the coagulation system such as F11 and SERPIND1 (Fig. 6c, Cluster 4). A comparison of the trajectories of individual proteins between patients who survived and died is shown in Supplementary Fig. 9. The most pronounced changes were observed among proteins constituting cluster 3 (P = 0.003) and cluster 4 (P < 0.001) (Fig. 6c). L-selectin levels (cluster 3) declined over time but were higher in patients who survived (Supplementary Fig. 9). In contrast, the recovery of many liver-derived proteins was suppressed in patients who died (cluster 4), including apolipoproteins linked to lipid metabolism (i.e., ApoB, ApoC1, and ApoE), biotinidase, complement factor H (CFH), and kininogen (Supplementary Fig. 9). Significant

correlations between proteins constituting these clusters and clinical variables are depicted in Supplementary Fig. 5.

**LGALS3BP is enriched in COVID-19 and binds to SARS-CoV-2 spike glycoprotein.** To further explore potential mechanistic links between RNAemia and circulating proteins, we searched for binding partners of the SARS-CoV-2 spike glycoprotein. The SARS-CoV-2 spike glycoprotein is the largest protein in the viral envelope, responsible for cell entry, and is the main target of neutralizing antibodies[41]. A magnetic affinity pulldown of a His-tagged SARS-CoV-2 spike glycoprotein mixed with plasma from COVID-19 ICU patients was coupled with proteomics to determine interaction partners. Proteomics analysis identified 28 spike-binding proteins after excluding contaminants and non-specific binders (Fig. 7a, b, Supplementary Table 7). Eight of them were immunoglobulins (Fig. 7a) and five were members of the complement system, which are known to directly interact with antigen-bound antibodies (i.e., C1 complement complex, Fig. 7b, Supplementary Table 7). Additional interaction partners included complement component 4-binding proteins alpha and beta (C4BPA and C4BPB), CPN1 (among the proteins associated with 28-day mortality, Fig. 3c), and galectin-3-binding protein (LGALS3BP). Apart from apolipoprotein D (APOD), LGALS3BP was the only protein to be retrieved to a greater extent with spike glycoprotein from plasma of COVID-19 ICU patients compared to COVID-19-negative plasma (Fig. 7c, Supplementary Table 8).

LGALS3BP was markedly elevated in COVID-19 patients as discovered by DIA-MS and confirmed by ELISA, but unchanged between control and sepsis patients without COVID-19 (Fig. 7d). Strikingly, LGALS3BP was among the most elevated proteins when compared to sepsis ICU patients (Fig. 7e). Of the proteins revealed to bind spike, only LGALS3BP and members of the complement cascade were also specifically elevated in COVID-19 ICU patients. LGALS3BP abundance in COVID-19 patients closely correlated with proteins and regulators of the complement cascade (C6, C9, C4BPA, and C4BPB) (Fig. 7f, Supplementary Fig. 10). While LGALS3BP rises with COVID-19 severity[20], LGALS3BP levels were not predictive for 28-day mortality and declined over time (Fig. 3c, Supplementary Fig. 11). We, therefore, explored the functional effects of LGALS3BP in cell-based assays.

**LGALS3BP impairs SARS-CoV-2 spike-mediated cell–cell fusion and spike-pseudoparticle entry in vitro.** SARS-CoV-2 spike induces cell-cell fusion (syncytia formation) when spike, ectopically expressed on the membrane of host cells, binds to ACE2 receptors of adjacent cells[42–45]. To test the effect of LGALS3BP on spike-mediated syncytia formation, HEK293-ACE2

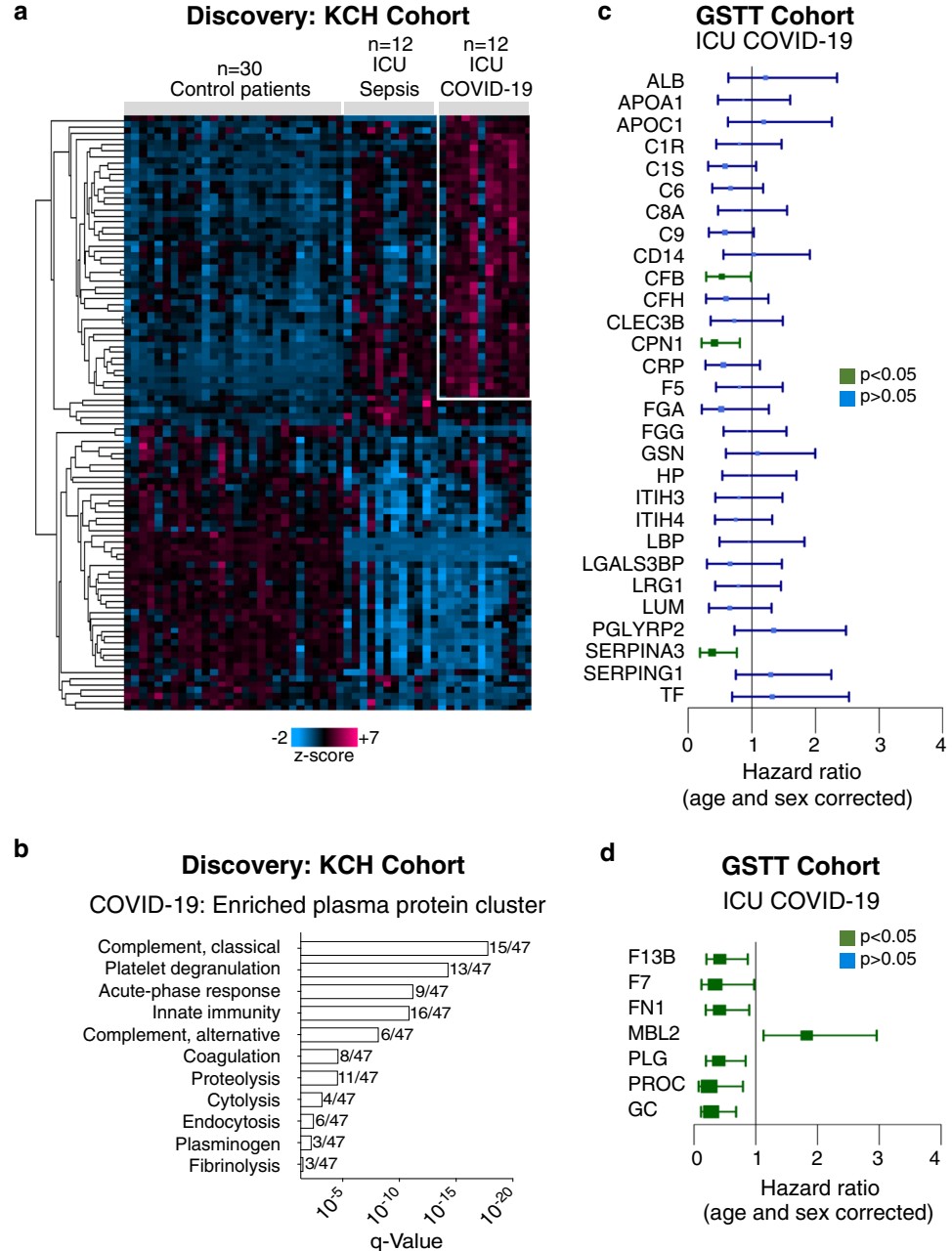

**Fig. 3 COVID-19 circulating proteome signature and associations with 28-day mortality. a** Plasma proteome profiling was conducted using a data-independent acquisition–mass spectrometry (DIA-MS) approach with spiked standards for 500 proteins. Hierarchical cluster analysis was conducted upon significantly changing plasma proteins across control patients before elective cardiac surgery ($n = 30$), ICU patients with sepsis ($n = 12$), and ICU patients with COVID-19 ($n = 12$, KCH). The heatmap highlights 47 proteins enriched in COVID-19. Kruskal–Wallis, Benjamini–Hochberg correction $q < 0.05$. **b** Gene ontology enrichment analysis was conducted upon these 47 proteins and significantly enriched pathways are represented. **c** Twenty-nine common proteins cross-referenced against two published proteomic studies, exploring protein markers of COVID-19 severity. The ability of these 29 proteins to predict 28-day mortality was explored in an independent ICU patient cohort ($n = 62$ patients, GSTT) by DIA–MS, and hazard ratios with 95% CI are shown. **d** Proteomic analysis by DIA-MS conducted upon the serum samples of the GSTT COVID-19 ICU cohort returned additional candidates that predict 28-day mortality as shown on hazard ratio plots with 95% CI ($n = 62$ patients, GSTT). Significance was determined through the Mann–Whitney $U$ test, correcting for age and sex and applying the Benjamini-Hochberg procedure. All statistical analyses are two-tailed.

and Vero cells were transfected with an LGALS3BP-coding plasmid, followed by transfection of a SARS-CoV-2 spike-coding plasmid 24 h later. Small interfering RNA (siRNA)-mediated knockdown of ACE2 served as a positive control (siACE2), while transfection of siNT1 (non-targeting control siRNA), pcDNA3 (plasmid backbone) and pmCherry (plasmid coding for mCherry) served as negative controls (Fig. 8a). As expected, siACE2 significantly reduced spike-mediated cell-cell fusion compared with

siNT1 (Fig. 8b–d). Strikingly, LGALS3BP overexpression also significantly reduced cell–cell fusion compared with pcDNA3 and pmCherry. This reduction in cell-cell fusion was dose-dependent with increasing amounts of LGALS3BP plasmid (Supplementary Fig. 12a–c). To rule out an effect of LGALS3BP on green fluorescent protein (GFP) expression, we co-transfected a GFP-coding plasmid demonstrating that the LGALS3BP-coding plasmid had no effect (Supplementary Fig. 12d, e). Overexpression of LGALS3BP

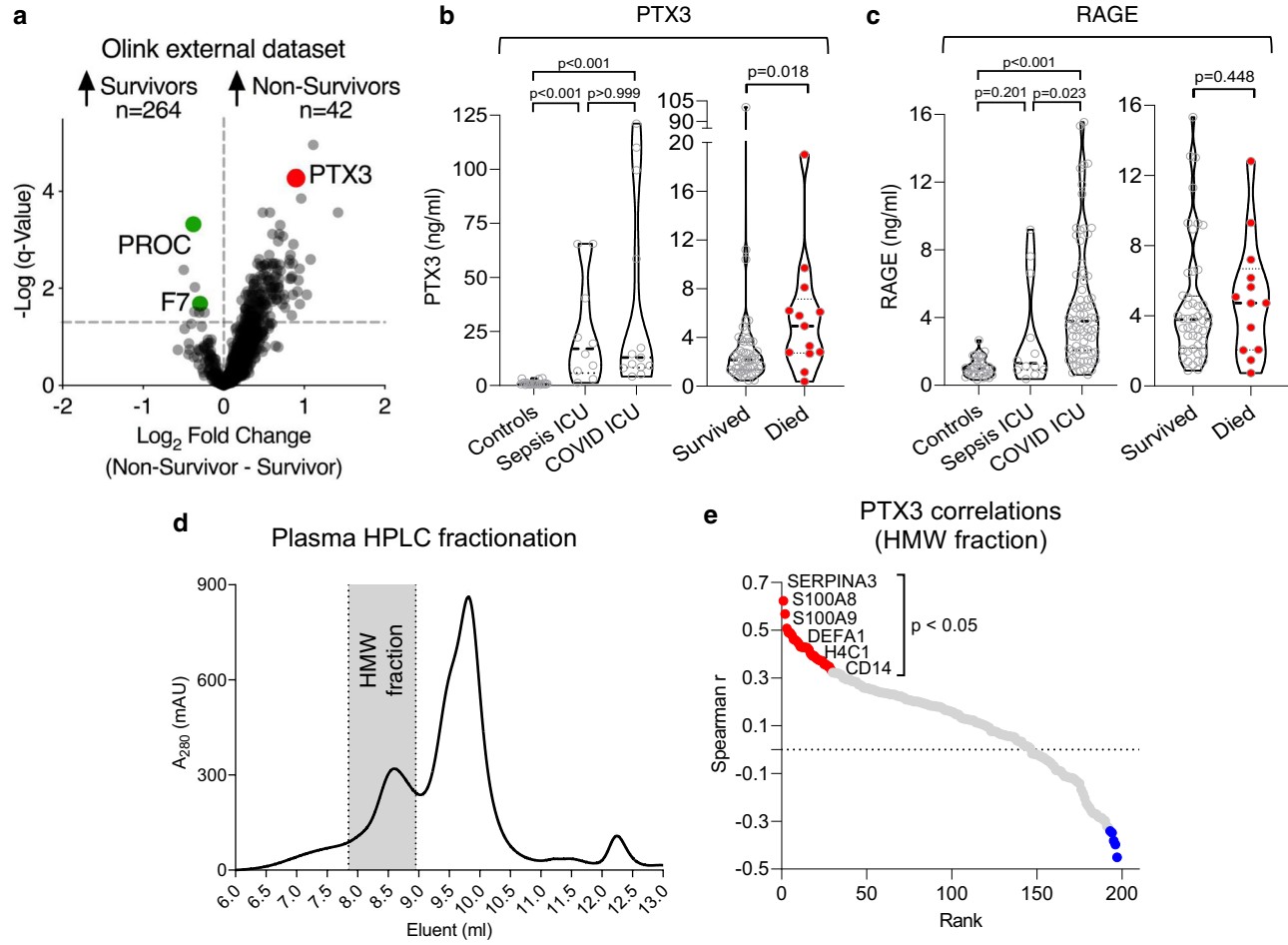

**Fig. 4 External protein marker validation and PTX3 selection. a** PTX3 was among the proteins most strongly associated with poor outcome among 1472 unique plasma proteins (Data provided by the MGH Emergency Department COVID-19 Cohort (Filbin, Goldberg, Hacohen) with Olink Proteomics)[38]. Of the 10 proteins we found to be associated with outcome in our DIA–MS data (Fig. 3c, d), PROC and F7 were the only proteins also measured in the external validation data, confirming an inverse association with mortality. The log2FC (adjusted for categorical age) was higher for PTX3: 0.8, adjusted $P$ value = 0.00044 compared with PROC: −0.4, adjusted $P$ value = 0.0006 and F7: −0.3, adjusted $P$ value = 0.03. **b** PTX3 measurements by ELISA (KCH and GSTT samples for COVID-19-ICU cohorts in left and right panel, respectively). **c** ELISA measurements for RAGE, as an established marker for ARDS. **d** High-performance liquid chromatography (HPLC) fractionation of plasma ($n = 35$-time points from 13 patients, KCH). PTX3-containing high molecular weight (HMW) fraction is shaded in gray. $A_{280}$ denotes the absorbance of the eluent at 280 nm. **e** Proteomics analysis of the HMW fraction. Significant Spearman correlations of PTX3 with neutrophil- and macrophage-related proteins. All statistical analyses are two-tailed.

and knockdown of ACE2 was verified by immunoblotting in HEK293-ACE2 cells (Supplementary Fig. 12f).

Next, we tested whether LGALS3BP affects the entry of pseudoparticles carrying the spike protein in addition to a GFP reporter, which is an established model of the SARS-CoV-2 entry pathway[46]. For this purpose, we transfected HEK293-ACE2 with an LGALS3BP-coding plasmid followed by the addition of pseudoparticles 24 h later (carrying spike or vesicular stomatitis virus G (VSV-G) protein as a control). siACE2 served as a positive control, while transfection of siNT1 and pcDNA3 served as negative controls (Fig. 8e). As expected, siACE2 significantly reduced cellular uptake of spike-pseudoparticles compared with siNT1 (Fig. 8f, g), while uptake of VSV-G particles remained unaffected (Fig. 8f, h). Strikingly, LGALS3BP overexpression also significantly reduced uptake of spike-pseudoparticles compared with pcDNA3 (Fig. 8f, g), while VSV-G uptake remained unaffected (Fig. 8f, h). In contrast, we did not observe a significant reduction in spike-pseudoparticle uptake when spike-pseudoparticles were pre-incubated with the supernatant from LGALS3BP-expressing cells (Supplementary Fig. 13a–c).

## Discussion

The main findings of our study include a 23% prevalence of SARS-CoV-2 RNAemia in ICU patients with COVID-19, an independent association of SARS-CoV-2 RNAemia with risk of 28-day mortality, and a proteomic trajectory characterized by four distinct protein clusters which mirrored the clinical status. Furthermore, we performed pulldown experiments with SARS-CoV-2 spike glycoprotein identifying LGALS3BP and complement system proteins as potential interaction partners. We highlight that overexpression of LGALS3BP impaired SARS-CoV-2 spike glycoprotein-induced syncytia formation and spike-pseudoparticle transduction efficiency.

SARS-CoV-2 RNAemia was observed in 23% of COVID-19 ICU patients within the first six days of admission to ICU, which is more frequent than its estimated prevalence (10% [95% CI: 5–18%], random-effects model)[7]. Likely explanations include: first, the fact that RNAemia is expected to be more common in ICU patients due to disease severity[7]. Second, we optimized detection by performing a two-step RT-qPCR protocol rather than the one-step RT-qPCR protocol used in previous studies in

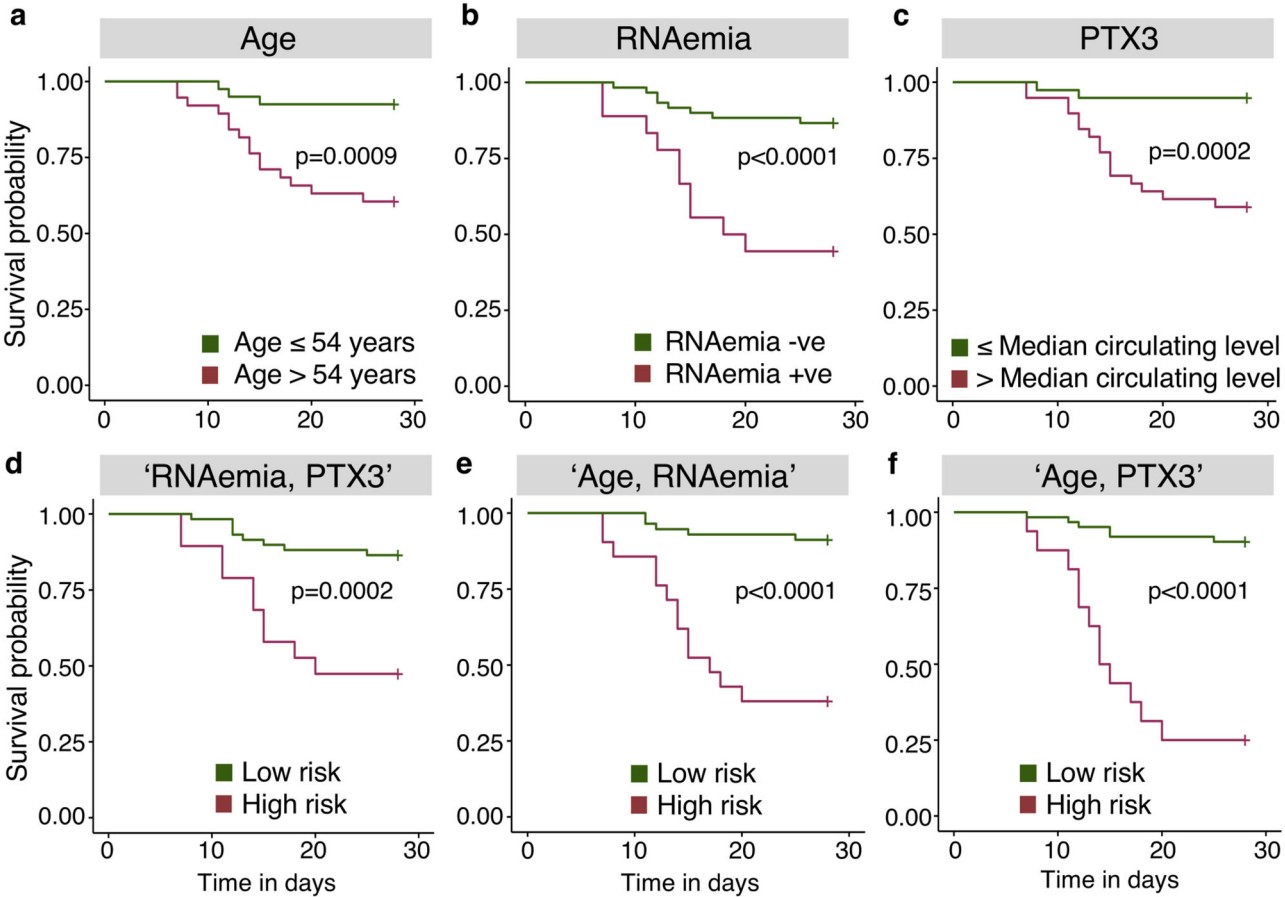

**Fig. 5 SARS-CoV-2 mortality prediction using machine learning. a** Kaplan–Meier plot for age (using the median age of 54 years). **b** Kaplan–Meier plot for SARS-CoV-2 RNAemia. As a single predictor, RNAemia provides the best stratification for survival. **c** Kaplan–Meier plot for PTX3 using the median levels of serum or plasma. **d**–**f** Kaplan–Meier plots for "RNAemia, PTX3", "Age, RNAemia", and "Age, PTX3" combined using support vector machine with radial basis function kernel (SVM RBF), a non-linear machine learning model. The machine learning model selected binary combinations of "Age, RNAemia" and "Age, PTX3" as the best predictors. Kaplan–Meier analysis is two-tailed. Nonsurvivors: $n = 18$; survivors: $n = 60$.

which RNAemia has been assessed thus far. Third, RNAemia was more frequent closer to the onset of symptoms[7] and when humoral response against SARS-CoV-2 was low. The latter observation was maintained after correcting for time since onset of symptoms. Thus, this is not a mere reflection of low humoral response in early sampling points. Using droplet digital PCR[15], RNAemia might become even more frequent but the clinical relevance of very low levels of RNAemia is unclear.

RNAemia within 6 days of ICU admission was strongly associated with 28-day mortality, which is a well-defined outcome measure in clinical trials[5,47]. Thus far, studies on RNAemia included predominantly non-ICU patients and associated RNAemia with disease severity[7]. Few studies also reported on the ability of RNAemia to predict mortality[8–10] but none of these studies specifically focused on ICU patients in which RNAemia is likely to be most informative. Our study focused on COVID-19 ICU patients ($n = 78$) with 28-day mortality as an outcome and included hospitalized, non-ICU COVID-19 patients ($n = 45$) as well as non-COVID-19 patients ($n = 55$). In comparison to RNAemia as assessed in our study (HR, 1.84 [95% CI: 1.22–2.77] adjusted for age and sex]), the mortality risk conferred by increased nasopharyngeal SARS-CoV-2 RNA levels was found to be small (HR, 1.07 [95% CI: 1.03–1.11], $n = 1145$)[48]. Correlation between nasopharyngeal and plasma viral load was previously found to be of moderate strength ($r = 0.32$)[9], suggesting that the viral load in the nasopharyngeal compartment only accounts for a minor part ($r^2 = 10.2\%$)[9] of the plasma variation. Thus, other

pathological processes are likely to contribute to RNAemia, independent of viral load. RNAemia could be a consequence of severe disease and might reflect the extent of viral dissemination. Notably, serum levels of CDH5, an endothelial-specific surface protein, differed between RNAemia positive versus negative ICU patients. RNAemia was also inversely associated with monocyte counts. A decrease in monocyte counts in COVID-19 patients has been attributed to extravasation and recruitment to lungs[11,49].

Similar to previous studies on RNAemia, the proteomic studies published to date focused on hospitalized COVID-19 patients. Proteome changes were associated with disease severity and healthy individuals were often used as controls[19–22]. In our study we highlight that few of the plasma protein changes associated with disease severity also predict outcome in ICU patients who are already critically ill, and included pre-pandemic sepsis ICU patients as an additional control. An argument could be made for using ARDS controls, due to similarities between non-COVID-19 ARDS and COVID-19 ARDS that we have previously reported[3,50]. However, in addition to our rationale for non-COVID-19 sepsis controls highlighted earlier, all sepsis patients used as controls were mechanically ventilated[51]; would meet the consensus definitions of ARDS[52]; and could have been enrolled in clinical trials of ARDS[53], since the most common etiology of ARDS is infection. The validation of the trajectory of differentially expressed proteins in COVID-19 and their association with outcome was done in serum samples of the larger GSTT COVID-19 ICU cohort ($n = 62$). The Olink measurements in the external

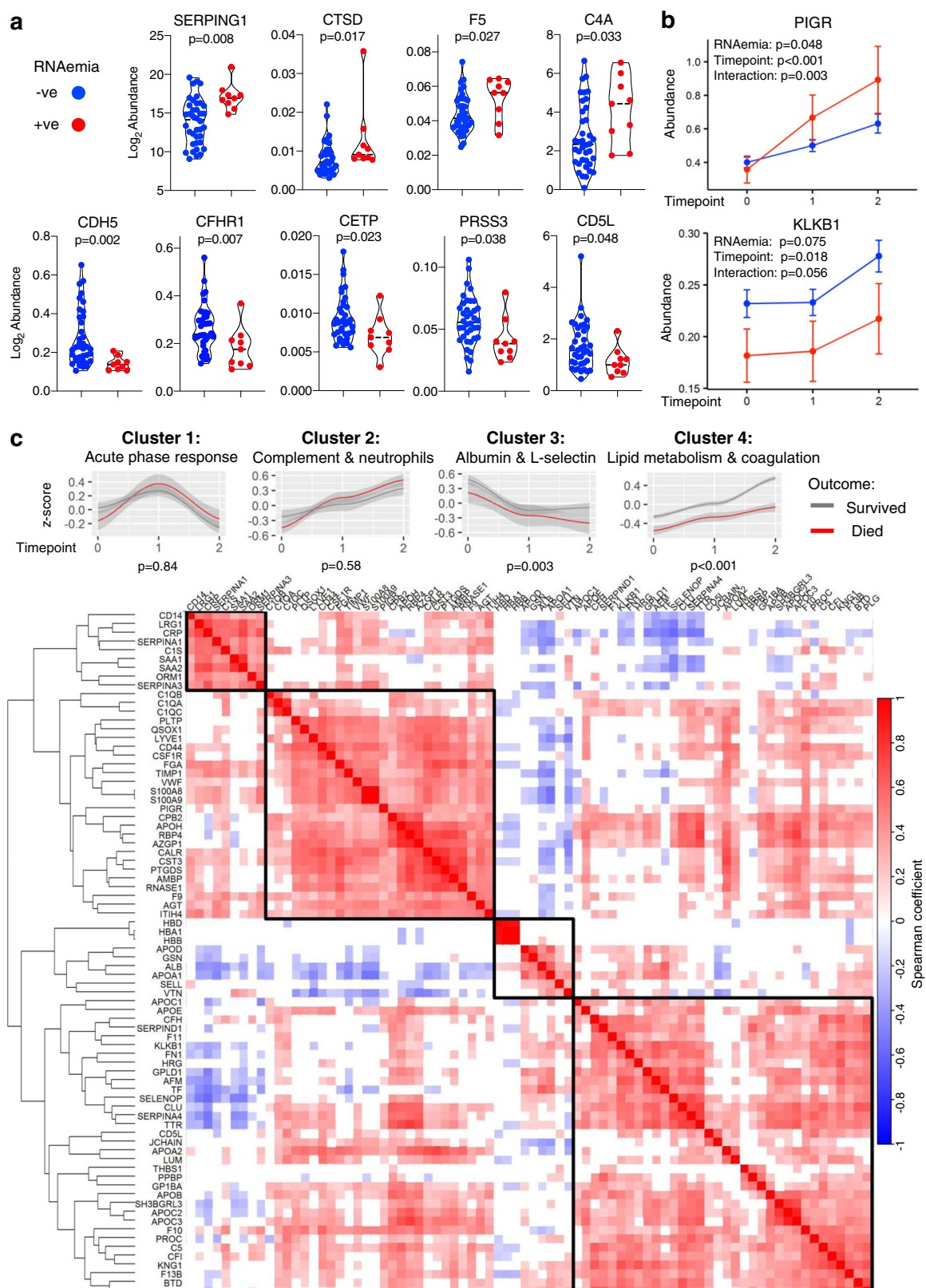

validation cohort were performed in plasma (Fig. 4a). Validation in plasma and in serum ensures that the protein changes are independent of the sample type, which is important for the clinical applicability of findings. The Olink platform covered two (of ten) proteins (PROC, F7) associated with 28-day mortality

measured by our DIA-MS approach, highlighting the complementarity of these different proteomics methods[54].

Our proteomics data reveal that complement activation is a core component of the overreaction of the immune system in response to COVID-19 (Fig. 3b), with elevated MBL2 being a predictor of

**Fig. 6 Circulating protein changes associated with SARS-CoV-2 RNAemia over time. a** DIA–MS analysis upon serum samples from the GSTT COVID-19 ICU cohort was used to determine proteins that associate with the presence of SARS-CoV-2 RNAemia at baseline ($n = 9$ positive, $n = 38$ negative). Proteins that were significantly associated with RNAemia at baseline are individually represented as violin plots. Significance was determined through the Limma linear model analysis using Benjamini and Hochberg's FDR correction. **b** Proteins with significantly different trajectories over time (baseline, week 1 —time point 1, week 2—time point 2) between RNAemia positive and negative patients ($n = 9$ positive patients, $n = 38$ negative patients with samples in each of the three-time points, totaling $n = 141$ samples). PIGR polymeric immunoglobulin receptor, KLKB1 kallikrein B1. The median and 95% CI of the median is shown. Unadjusted for multiple comparisons. **c** Serial serum samples from COVID-19 ICU patients (GSTT, baseline, week 1 and week 2, $n = 10$ nonsurvivors, $n = 37$ survivors with samples in each time point, totaling 141 samples) were analyzed by DIA–MS to determine protein changes over time in ICU. The heat map represents a hierarchical cluster analysis conducted upon a Spearman correlation network of significantly changing proteins over time in ICU, applying row-wise corrections for multiple testing using the Benjamini-Hochberg FDR correction. Comparison of the trajectories of protein clusters in COVID-19 ICU patients based on 28-day mortality is also shown. Gene ontology enrichment analysis was used to determine functional pathways associated with the distinct protein clusters identified. Listed are the protein clusters that show a significant change between 28-day survivors (gray) and nonsurvivors (red)—and having significant interaction with time points (baseline, week 1—time point 1, week 2—time point 2). Lines show nonlinear regression curves with gray bands indicating the 95% CI. $P$ values represent the significance of the outcome term in a fitted GAM model when correcting for age and sex. All statistical analyses are two-tailed.

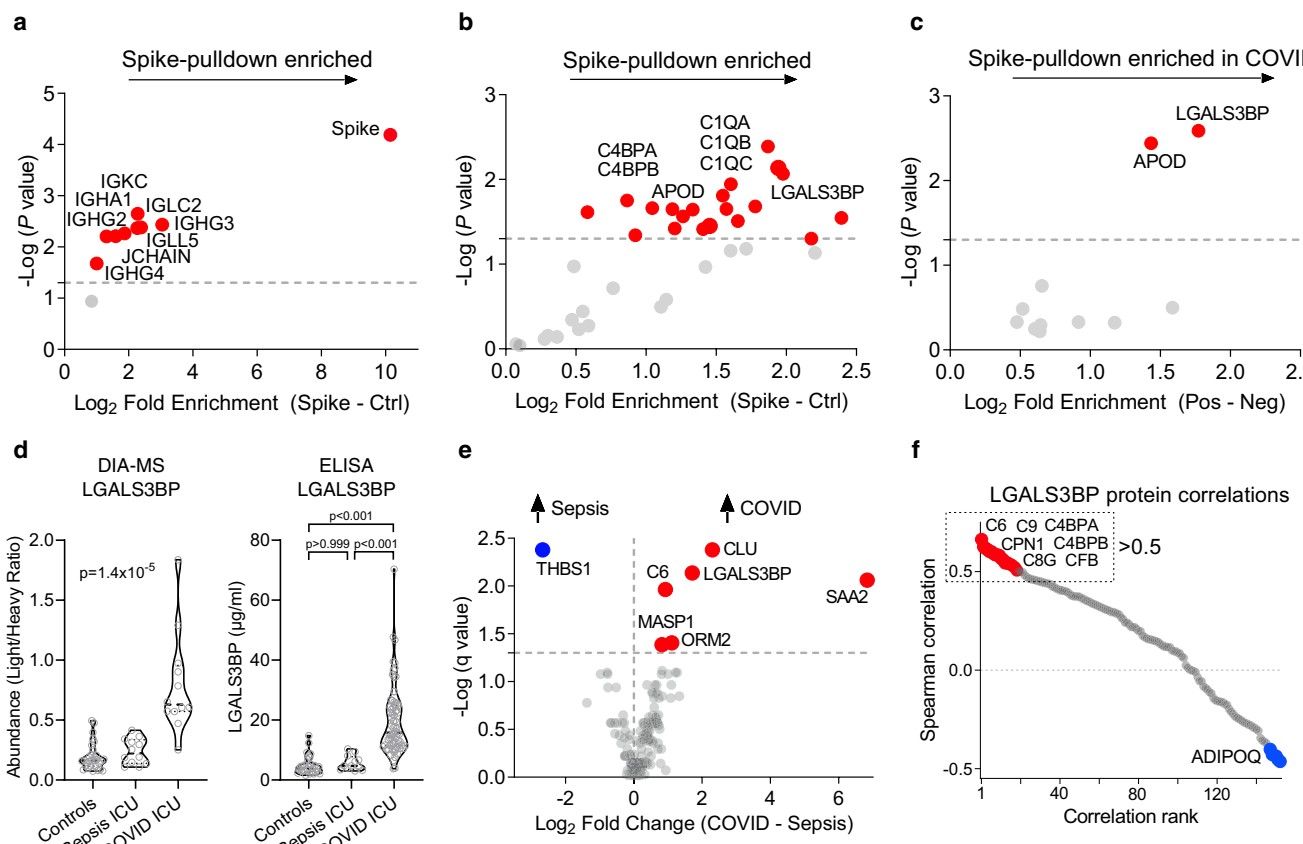

**Fig. 7 LGALS3BP interacts with SARS-CoV-2 spike glycoprotein. a** Magnetic bead-based affinity isolation of binding partners using His-tagged SARS-CoV-2 spike glycoprotein as a bait for proteins in SARS-CoV-2-positive patient plasma ($n = 8$). Volcano plot depicting significantly enriched constant chains of immunoglobulins. **b** Volcano plot depicting significantly enriched non-immunoglobulin proteins ($n = 8$). **c** Comparison of SARS-CoV-2 spike glycoprotein pulldown using plasma from COVID-19 ICU patients ($n = 8$) and non-COVID-19 patients ($n = 3$). Significance was determined by paired Student's $t$ tests for (**a**) and (**b**) and unpaired Student's $t$ tests for (**c**). **d** LGALS3BP levels across three patient cohorts as determined by DIA-MS or ELISA: control patients before undergoing elective cardiac surgery ($n = 30$), pre-pandemic sepsis ICU patients ($n = 12$) and COVID-19 ICU patients ($n = 74$). Kruskal–Wallis and Dunn's multiple comparisons tests were used to determine statistical significance. **e** Volcano plot representing protein changes between baseline plasma samples from patients in ICU with either sepsis ($n = 12$) or COVID-19 ($n = 12$). Significance was determined through the Mann–Whitney $U$ test with Benjamini-Hochberg's FDR correction. **f** Plasma proteins correlating to LGALS3BP after age and sex corrections in COVID-19 ICU patients ($n = 12$) are highlighted by a Spearman correlation matrix across the proteomic dataset. Proteins with a Spearman correlation coefficient greater than 0.5 were used for gene ontology pathway enrichment analysis (Supplementary Fig. 10). All statistical analyses are two-tailed.

28-day ICU mortality (Fig. 3d). Systemic complement activation has been associated with respiratory failure in hospitalized COVID-19 patients[55], and complement deficiencies appear to have protective effects on COVID-19-associated morbidity and mortality[56]. Similarly, C3 deficient mice developed less respiratory dysfunction following SARS-CoV infection[57]. It was recently reported[58] that circulating complement factors are high before seroconversion, while markers of systemic complement activation decrease after seroconversion, although this particular study did not include patients admitted to ICU with life-threatening

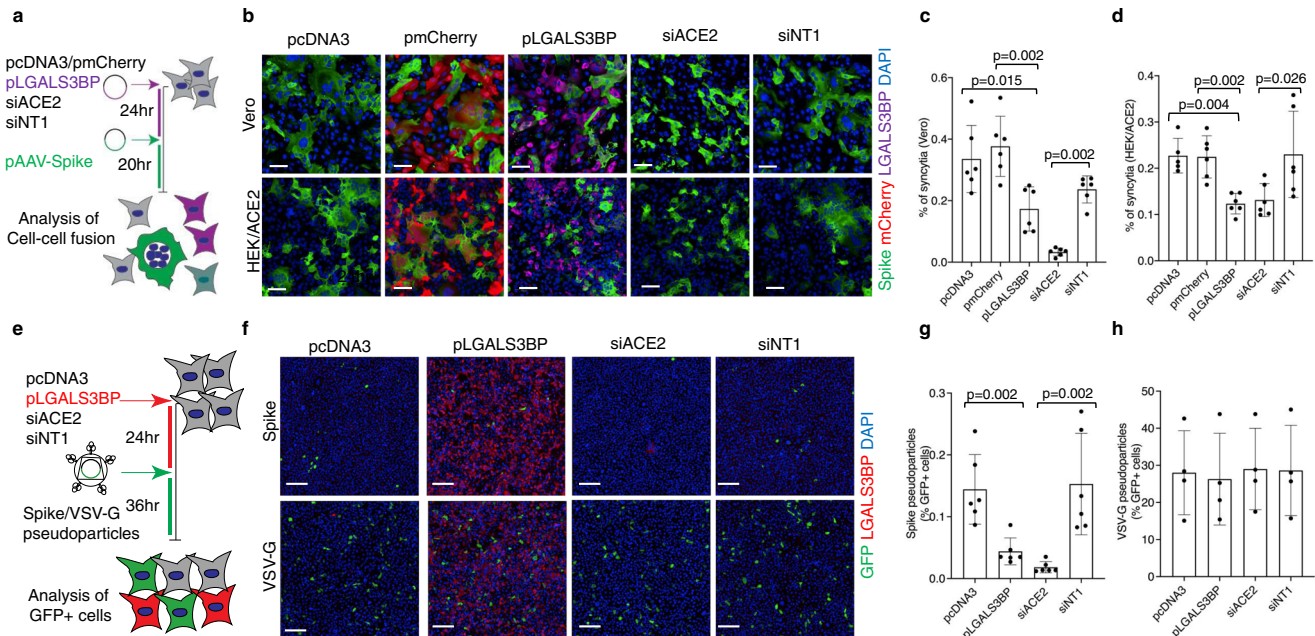

**Fig. 8 LGALS3BP overexpression impairs SARS-CoV-2 spike-mediated syncytia formation and cellular uptake of SARS-CoV-2 spike-pseudoparticles.**
**a** Schematic representation of the SARS-CoV-2 spike-mediated cell-cell fusion assay. **b–d** Vero and HEK293-ACE2 cells were transfected either with pcDNA3 (plasmid backbone), pmCherry (plasmid coding mCherry), pLGALS3BP (plasmid coding LGALS3BP), siACE2 (siRNA targeting ACE2), or siNT1 (non-targeting siRNA), followed by transfection of pAAV-Spike (plasmid coding SARS-CoV-2 spike) 24 h later. After 20 h, cells were stained with anti-LGALS3BP (violet), anti-Spike (green), and DAPI for nuclei (blue). Representative images are shown in (**b**), and quantifications are shown in (**c**) for Vero cells and in (**d**) for HEK293-ACE2 cells. Data (mean ± standard deviation; $n = 6$; Mann–Whitney $U$ test) are plotted as the percentage of fused cells (syncytia) normalized to the total number of cells. Scale bars in (**b**) represent 100 μm. **e** Schematic representation of the SARS-CoV-2 spike/VSV-G pseudoparticle transduction assay. **f–h** HEK293-ACE2 cells were transfected either with pcDNA3, pLGALS3BP, siACE2, or siNT1, followed by the addition of spike- or VSV-G pseudoparticles carrying a GFP reporter 24 h later. After 36 h, cells were stained with anti-LGALS3BP (red), anti-GFP (green), and DAPI for nuclei (blue). Representative images are shown in (**f**) and quantifications are shown in (**g**) for spike-pseudoparticles (mean ± standard deviation; $n = 6$; Mann–Whitney $U$ test) and in (**h**) for VSV-G pseudoparticles (mean ± standard deviation; $n = 3$). Data are plotted as the percentage of GFP-positive cells normalized to the total number of cells. Scale bars in (**f**) represent 200 μm. All statistical analyses are two-tailed.

COVID-19[58]. Interestingly, patients with RNAemia (Fig. 6a) showed dysregulation in several components of the complement (SERPING1, C4A, and CFHR1), the coagulation (F5) and the kinin-kallikrein system (KLKB1). Combined activation of these pathways is a hallmark of thromboinflammation[59,60].

Viral envelope glycoproteins are known to be an important trigger of the contact pathway of coagulation[59] and the complement system[61]. MBL2 binds to glycoproteins on the viral surface[62,63] and is a key molecule for the lectin pathway of complement activation in the circulation[64]. MBL2 levels were markedly elevated in COVID-19 ICU patients who died. High levels of MBL2 have previously been associated with lectin pathway-mediated tissue damage[65,66]. Furthermore, we found MASP1, the downstream mediator of MBL2, to be increased in plasma from COVID-19 ICU patients compared with sepsis patients. Moreover, PTX3 is important for activation (through MBL2 and C1q)[39] and regulation (through CFH and C4BPB)[67,68] of the complement system[64,69]. The MBL2/PTX3 complex can directly activate the complement system independent of antigen-antibody complexes. Consistent with our previous results on PTX3 in sepsis patients[31,70], PTX3 also emerged as a strong predictor for mortality in COVID-19 ICU patients. This is in agreement with other studies[38,40,71] reporting strong associations of PTX3 with COVID-19 mortality[71]. PTX3 is released from neutrophils upon activation[70,72], is abundant in macrophages[40], but also highly expressed in lung and adipose tissue (www.gtexportal.org/home/gene/PTX3). While anti-SARS-CoV-2 antibody levels were similar in COVID-19 ICU patients who survived and died, MBL2 and PTX3 were associated with poor outcome

pointing towards the importance of antibody-independent mechanisms of complement activation in COVID-19. Clinical trials with complement inhibitors are currently ongoing for COVID-19[59].

Besides members of the complement system, we demonstrate that LGALS3BP is a novel binding partner of SARS-CoV-2 spike glycoprotein. However, the direct interaction between LGALS3BP and SARS-CoV-2 spike remains to be confirmed as pulldown assays cannot rule out indirect binding to the bait protein. Interestingly, the N-terminal domain of the SARS-CoV-2 spike glycoprotein is highly homologous to human galectin-3[73], which may explain the interaction we observed with LGALS3BP. The presence of this 13-strand beta-sheet domain is not unique to the SARS-CoV-2 spike as it is also present in the spike glycoproteins of other members of the *Betacoronavirus* genus[74]. It has been proposed that this galectin domain was acquired from host cells and provided the virus with another cell attachment mechanism in its arsenal thus presenting an evolutionary advantage[75]. Thus, it has been suggested that galectin-3 inhibitors may be useful in the treatment of COVID-19[73,76]. LGALS3BP is prominently expressed in the lung[77], but also adipose tissue (www.gtexportal.org/home/gene/LGALS3BP) and possesses antiviral activity[78]. The rise in circulating LGALS3BP is not observed in non-COVID-19 sepsis ICU patients, highlighting the specificity for viral over bacterial infections. LGALS3BP directly interacts with adeno-associated viruses, inducing viral particle aggregation and impairment of transduction[79]. LGALS3BP also reduces the infectivity of human immunodeficiency virus (HIV) particles[80]. The antiviral effect on HIV is mediated predominantly through

intracellular effects[80]. Similarly, the supernatant from LGALS3BP-overexpressing cells did not inhibit SARS-CoV-2 spike-pseudoparticle entry. However, LGALS3BP overexpression reduced spike-mediated syncytia formation, and decreased spike-pseudoparticle entry, which are two functional readouts that we have recently used for drug discovery in COVID-19[45]. Thus, LGALS3BP may impede SARS-CoV-2 also through extracellular effects. LGALS3BP may be bound to the extracellular surface of the plasma membrane, similarly to other secreted proteins such as fibroblast growth factor-2[81] and HIV-1 tat protein[82]. The possibility of LGALS3BP binding to carbohydrates on the extracellular surface could explain its antiviral effects[83].

In summary, we report that RNAemia in COVID-19 ICU patients is associated with a higher risk of death, an observation that could potentially be a disease-specific enrichment biomarker[84] for antiviral medications, given the lack of benefit of these drugs in unselected ICU patients with COVID-19[85]. Proteomics analyses of blood samples from ICU patients with COVID-19 uncovered protein trajectories that mirror the recently reported immune trajectory in COVID-19 and add further granularity to COVID-19 biology, in particular with regard to complement activators of the innate immune system (MBL2/PTX3) being associated with mortality; and recovery of several liver-derived proteins being linked to survival[86]. Finally, our observation that LGALS3BP is a novel interaction partner of the SARS-CoV-2 spike glycoprotein has potential therapeutic implications.

## Methods

**Study design and recruitment**. An overview of the study design is presented in Supplementary Fig. 1. COVID-19 cohorts: COVID-19-positive patients, as confirmed by RT-qPCR of nasopharyngeal samples, who were admitted to the ICUs of Guy's and St Thomas' NHS Foundation Trust (GSTT) and King's College Hospital (KCH) between March 12, 2020, and July 1, 2020, were recruited for an observational cohort study with serial blood sampling and analysis of clinical outcomes. The primary outcome measure was defined as mortality 28 days after ICU admission. Serial blood sampling was performed within 24 h of admission to ICU and thereafter three measurements were taken during week 1, week 2, and again before discharge. In addition, we obtained plasma samples from COVID-19 patients upon hospitalization at GSTT (non-ICU COVID-19 cohort). Non-COVID-19 comparator cohorts: Plasma was collected from patients enrolled at the same time in the same KCH ICU as our COVID-19 ICU cohort but who repeatedly tested negative for nasopharyngeal SARS-CoV-2 (intra-pandemic, non-COVID-19 ICU cohort). Serial blood sampling of these samples was performed identically to our COVID-19 cohort. Additionally, pre-pandemic plasma samples from patients recruited at GSTT prior to the COVID-19 pandemic were available as controls. Firstly, this included serial plasma samples from sepsis ICU patients (pre-pandemic, non-COVID-19 ICU sepsis cohort) recruited between October 16, 2019, and February 26, 2020, collected upon admission and at three-time points thereafter. Sepsis was defined according to Sepsis-3 definitions (infection with organ dysfunction defined using SOFA score 2 or more points). The eligibility criteria for this cohort and the study protocol have been reported before[51]. Secondly, plasma samples from patients before elective cardiac surgery (pre-pandemic, non-COVID-19 control cohort) recruited between July 8, 2019, and September 9, 2019, The study was approved by an institutional review board (REC19/NW/0750 for all patients recruited at KCH; REC19/SC/0187 for patients recruited at GSTT of the COVID-19 ICU cohort, the pre-pandemic sepsis ICU cohort, the pre-pandemic control cohort; REC19/SC/0232 for patients recruited at GSTT of the non-ICU COVID-19 cohort). Written informed consent was obtained directly from patients (if mentally competent), or from the next of kin or professional consultee. The consent procedure was then completed with retrospective consent if the patient regained capacity.

**Inactivation of serum and plasma**. Plasma was collected in EDTA BD Vacutainer™ tubes (BD, 362799), whereas serum was collected in silica BD Vacutainer™ tubes (BD, 367820) and left to clot for 15 min. Plasma and serum tubes were then centrifuged at $2000 \times g$ for 15 min. Infectious samples were then transferred to a containment level 3 facility for safe inactivation. Samples destined for RNA extraction were inactivated by the addition of 100 μL of serum or plasma to 500 μL QIAzol (Qiagen, 79306), followed by 40 s of vortexing and 5 min incubation at room temperature. Samples destined for protein analysis were inactivated by the addition of 1% (v/v) Triton X-100 (Sigma, T8787) and 1% (v/v) tributyl phosphate (Sigma, 00675), followed by 15 s of vortexing and 4 h incubation at room

temperature. Heat treatment was not performed to avoid protein precipitation. All samples were then frozen at −80 °C until further processing.

**High-performance liquid chromatography fractionation of plasma**. High-performance size-exclusion chromatography of the KCH plasma samples ($n = 35$, from 13 patients) was performed using a TSKgel® G5000PW$_{XL}$ column (hydroxylated methacrylate, 10 μm particle size, 100 nm mean pore size; Tosoh Bioscience, 0008023) equipped with a TSKgel® PW$_{XL}$ guard column (hydroxylated methacrylate, 12 μm particle size, mixed pore size; Tosoh Bioscience, 0008033). Totally, 20 μL plasma was fractionated with PBS as a mobile phase at a flow rate of 0.6 ml/min. The high molecular weight (HMW) fraction (7.86–8.96 ml eluent) was denatured, reduced, alkylated, digested, and DDA–MS was performed.

**RNA extraction and heparinase treatment**. Total RNA was extracted using the miRNeasy Mini kit (Qiagen, 217004) according to the manufacturer's recommendations. Total RNA was eluted in 30 μL of nuclease-free H$_2$O by centrifugation at $8500 \times g$ for 1 min at 4 °C. To overcome the confounding effect of heparin on qPCR[87,88], RNA was treated with heparinase[89]. Briefly, 8 μL of RNA was added to 2 μL of heparinase 1 from Flavobacterium (Sigma, H2519), 0.4 μL RNase inhibitor (Ribo Lock 40 U/μL, ThermoFisher, EO0381) and 5.6 μL of heparinase buffer (pH 7.5) and incubated at 25 °C for 3 h.

**Reverse transcription-quantitative polymerase chain reaction (RT-qPCR)**. For detection of SARS-CoV-2 RNA we performed a two-step RT-qPCR using the LunaScript® RT SuperMix Kit (NEB, E3010) and the Luna Universal Probe qPCR Master Mix (NEB, M3004) according to the manufacturer's recommendations, apart from reducing the total qPCR reaction volume to 5 μL and loading a cDNA dilution of 1:4 instead of 1:8 when performing the qPCR reaction. Primer/probe sequences targeting the SARS-CoV-2 nucleocapsid (N) gene (N1 and N2) were predesigned by Integrated DNA Technologies (IDT, 10006821, 10006822, 10006823, 10006824, 10006825, and 10006826) according to the protocol for the detection of SARS-CoV-2 of the United States Centers for Disease Control and Prevention (US CDC), using 5′ FAM/ZEN™/3′ Iowa Black™ FQ probes (Supplementary Table 9). The qPCR reaction concentration for probe (125 nM), forward (500 nM) and reverse primers (500 nM) were used according to the US CDC protocol. A plasmid positive control (2019-nCoV_N Positive Control plasmid, IDT, 10006625) was measured on each qPCR plate. Reactions were loaded using a Bravo Automated Liquid Handling Platform (Agilent). qPCR was performed on a ViiA7 Real-Time PCR System (Applied Biosystems). Samples were considered positive for SARS-CoV-2 if the cycle quantification (Cq) value of either N1 or N2 was below 40. The abundance of SARS-CoV-2 RNA in patients who tested positive had a mean Cq of 34.4; range: 29.8–37.6. As reported before[90], N1 primers returned lower Cq values (higher abundance) than N2 primers (Supplementary Fig. 14).

**Measurement of anti-SARS-CoV-2 antibodies**. IgG antibodies against the SARS-CoV-2 spike S1 domain were measured by ELISA (Anti-SARS-CoV-2 IgG ELISA, Euroimmun, EI 2606-9601G) according to the manufacturer's recommendations. Since no international reference serum for anti-SARS-CoV-2 antibodies exists, calibration was performed in ratios, giving relative antibody quantification. Neutralizing antibodies against SARS-CoV-2 were measured using a Surrogate Virus Neutralization Test (SARS-CoV-2 sVNT Kit, GenScript, L00847)[36] according to the manufacturer's recommendations. This ELISA-based kit detects antibodies that are able to block the interaction between the SARS-CoV-2 spike RBD and the angiotensin-converting enzyme (ACE2) cell receptor. For technical validation of sVNT measurements in a subset of samples (38 samples from 16 ICU patients), neutralization potency was measured using HIV-1 (human immunodeficiency virus-1) based virus particles, pseudotyped with SARS-CoV-2 spike protein in a HeLa cell line stably expressing the ACE2 receptor[13]. Briefly, serial dilutions of serum samples were prepared with Dulbecco's Modified Eagle Medium (DMEM) containing 10% fetal bovine serum (FBS) and 1% penicillin-streptomycin; and incubated with pseudotype virus for 1 h at 37 °C in 96-well plates. The HeLa cells stably expressing the ACE2 receptor (provided by J. Voss, Scripps Research) were then added (12,500 cells per 50 μL per well) and the plates were incubated for 72 h. Infection levels were assessed in lysed cells with the Bright-Glo luciferase kit (Promega), using a Victor X3 Multilabel Reader (Perkin Elmer). Measurements were performed in duplicate and duplicates were used to calculate the ID$_{50}$. In this technical validation, a significant positive correlation was obtained for values returned by both assays ($r = 0.81$, $P < 0.0001$).

**In-solution protein digestion**. Totally, 10 μL of inactivated serum or plasma were denatured by the addition of urea (final concentration 7.2 M) and reduced using dithiothreitol (final concentration 5 mM) for 1 h at 37 °C and shaking at 180 rpm. Reduced proteins were cooled down to room temperature before being alkylated in the dark for 1 h using iodoacetamide (final concentration 25 mM). An aliquot equivalent to 40 μg of alkylated protein was added to a 0.1 M triethylammonium bicarbonate solution (pH 8.2) and digested for 18 h at 37 °C, shaking at 180 rpm using 1.6 μg of Trypsin/LysC (Promega, V5072). Digested peptide solutions were acidified using trifluoroacetic acid (TFA, final concentration 1%).

**Peptide clean-up and stable isotope-labelled standard (SIS) spike-in**. Peptide clean-up was achieved using a Bravo AssayMAP Liquid Handling Platform (Agilent). After conditioning and equilibration of the resin, acidified peptide solutions were loaded onto AssayMAP C18 Cartridges (Agilent, 5190-6532), washed using 1% acetonitrile (ACN), 0.1% TFA (aq) and eluted using 70% ACN, 0.1% TFA (aq). Eluted peptides were vacuum centrifuged (Thermo Scientific, Savant SPD131DDA) to dry and resuspended in 40 μL of 2% ACN, 0.05% TFA (aq). For clinical cohort analysis, 6 μL of cleaned peptide solution was added to two injection equivalents of PQ500 SIS mix (Biognosys, Ki-3019-96) using a Bravo Liquid Handling Platform (Agilent).

**DIA–MS analysis**. Peptides were analyzed using a high-performance liquid chromatography (HPLC)–MS assembly consisting of an UltiMate 3000 HPLC system (Thermo Scientific) which was equipped with a capillary flow selector and coupled via an EASY-Spray NG Source (Thermo Scientific) to an Orbitrap Fusion Lumos Tribrid mass spectrometer (Thermo Scientific). To generate DIA data for serum samples (GSTT COVID-19 ICU cohort) and plasma samples (KCH COVID-19 ICU cohort, the pre-pandemic sepsis ICU cohort and the pre-pandemic control patients before elective cardiac surgery), peptides were injected onto a C18 trap cartridge (Thermo Scientific, 160454) at a flow rate of 25 μL/min for 1 min, using 0.1% formic acid (FA, aq). The initial capillary flow rate was reduced from 3 to 1.2 μL/min in 1 min at 1% B. Peptides were then eluted from the trap cartridge and separated on an analytical column (Thermo Scientific, ES806A, at 50 °C) using the following gradient: 1–11 min, 1–5% B; 11–32 min, 5–18% B; 32–52 min, 18–40% B; 52–52.1 min, 40–99% B; 52.1–58 min, 99% B. The flow rate was increased to 3 μL/min and the column was washed using the following gradient: 58–58.1 min, 99–1% B; 58.1–59.9 min, 1–99% B; 59.9–60 min, 99–1% B. Finally, the column was equilibrated at 1% B for 6 min. In all HPLC-DIA-MS analyses, mobile phase A was 0.1% FA (aq) and mobile phase B was 80% ACN, 0.1% FA (aq). Precursor MS1 spectra were acquired using Orbitrap detection (resolution 60000 at 200 m/z, scan range 329–1201 m/z). Quadrupole isolation was used to sequentially scan 30 precursor m/z windows of variable width (Supplementary Table 10). Per isolation window, semi-targeted Orbitrap MS2 spectra (resolution 30000 at 200 m/z) were collected following higher-energy C-trap dissociation.

**MS database search for DIA–MS analysis**. PQ500 SIS-spiked DIA data from all serum and plasma samples of the GSTT COVID-19 ICU cohort, the KCH COVID-19 ICU cohort, the non-COVID-19 sepsis ICU cohort and the control patients before elective cardiac surgery were analyzed in Spectronaut v14 (Biognosys AG), using the provided PQ500 analysis plug-in. MS1 and MS2 mass tolerance strategies were set to dynamic. Retention time calibration was achieved using the spiked iRT peptides included in the PQ500 SIS mix. Precursor and protein Q-value cutoff was set to 0.01. Quantification was conducted at an MS2 level using peak areas and individual runs were normalized using the global strategy set to the median. All peptides for reported proteins were manually checked to ensure accurate peak integration across all samples. Peptides with a Q-value of more than 0.01 or a signal to noise ratio of less than 5 were marked as missing. Peptides with more than 30% missing values across all samples were filtered out and the remaining missing values were imputed using the K nearest neighbours (KNN) algorithm (K = 5)[91]. Spearman correlations of peptides belonging to the same protein were computed. In case more than two peptides per protein were detected, peptides were filtered if their correlation with the remaining peptides was less than r = 0.4. In case two peptides per protein were detected, the most abundant peptide was kept even when correlation was less than r = 0.4. Final protein abundance was calculated by summing up the quantified peptide abundances. Final quantitative comparisons were conducted using the light/heavy peptide abundance ratio. For validation of our DIA-MS data, we correlated levels to clinical measurements of albumin (n = 49, r = 0.68, P < 0.05) and C-reactive protein (n = 49, r = 0.83, P < 0.05) as examples of high and medium-abundant proteins.

**Enzyme-linked immunosorbent assay (ELISA)**. ELISAs for receptor for advanced glycation end-products (RAGE; R&D Systems, DRG00), galectin-3-binding protein (LGALS3BP; R&D Systems, DGBP30B), and pentraxin-3 (PTX3; Abcam, ab214570) were performed according to the manufacturer's instructions.

**SARS-CoV-2 spike protein pulldown**. His-tagged recombinant SARS-CoV-2 spike glycoprotein (RP-87680, ThermoFisher) was added to 1:2 PBS-diluted plasma from COVID-19 ICU patients (n = 8) or non-COVID-19 controls (n = 3) at 200 ng/μL and incubated overnight at 4 °C with intermittent mixing. His-tagged spike was then isolated by means of metal affinity magnetic beads (Dynabeads His-Tag Isolation and Pulldown, 10103D, ThermoFisher) and eluted in imidazole-containing phosphate buffer. Proteins in the pulldown isolates were denatured, reduced, alkylated and precipitated, as described above. Proteins interacting non-specifically with the solid phase were determined by incubating plasma samples with magnetic beads without the addition of His-tagged spike. Pulldown of His-tagged spike without the addition of plasma was performed as an additional control. Spike pulldown protein digestion followed the same protocol outlined above.

**Data-dependent acquisition (DDA)–MS analysis**. Proteins from the spike pulldown experiments were subject to in-solution tryptic digestion and C18 clean-up as described above. Tryptic peptides were analyzed by LC–MS/MS. An UltiMate 3000 HPLC system (Thermo Scientific) with a nanoflow selector was coupled via an EASY-Spray Source (Thermo Scientific) to a Q Exactive HF mass spectrometer (Thermo Scientific). Peptides were injected onto a C18 trap cartridge (Thermo Scientific, 160454) at a flow rate of 25 μL/min for 1 min, using 0.1% FA (aq). Peptides were eluted from the trap cartridge and separated on an analytical column (EASY-Spray C18 column, 75 μm × 50 cm, Thermo Scientific, ES803A, at 45 °C) at a flow rate of 0.25 μL/min using the following gradient: 0–1 min, 1% B; 1–6 min, 1–6% B; 6–40 min, 6–18% B; 40–70 min, 18–35% B; 70–80 min, 35–45% B; 80–81 min, 45–99% B; 81–89.8 min, 99% B; 89.8–90 min, 99–1% B; 90–120 min, 1% B. Mobile phase A was 0.1% FA (aq) and mobile phase B was 80% ACN, 0.1% FA (aq). Precursor MS1 spectra were acquired using Orbitrap detection (resolution 60,000 at 200 m/z, scan range 350–1600). Data-dependent MS2 spectra of the most abundant precursor ions were obtained after higher-energy C-trap dissociation and Orbitrap detection (resolution 15,000 at 200 m/z) with TopN mode (loop count 15) and dynamic exclusion (duration 40 s) enabled.

**MS database search for DDA–MS analysis**. Proteome Discoverer software (version 2.3.0.523, Thermo Scientific) was used to search raw SARS-CoV-2 spike glycoprotein pulldown data files against a human database (UniProtKB/Swiss-Prot version 2020 01, 20,365 protein entries) supplemented with SARS-CoV-2 spike glycoprotein (1 protein entry) using Mascot (version 2.6.0, Matrix Science). The mass tolerance was set at 10 ppm for precursor ions and 0.02 Da for fragment ions. Trypsin was used as the digestion enzyme with up to two missed cleavages being allowed. Carbamidomethylation of cysteines and oxidation of methionine residues were chosen as fixed and variable modifications, respectively.

**Cell culture**. HEK293T cells (ATCC CRL-3216) were cultured in Dulbecco's modified Eagle medium (DMEM) with 1 g/L glucose (Life Technologies) supplemented with 10% FBS (Life Technologies) plus a final concentration of 100 IU/ml penicillin and 100 μg/ml streptomycin, or without antibiotics where required for transfections. Vero (WHO) Clone 118 cells (ECACC 88020401) were cultured in Dulbecco's modified Eagle medium (DMEM, Life Technologies) with 1 g/L glucose (Life Technologies) supplemented with 10% heat-inactivated FBS (Life Technologies) plus a final concentration of 100 IU/ml penicillin and 100 μg/ml streptomycin, or without antibiotics where required for transfection. Cells were incubated at 37 °C, 5% CO$_2$.

**Plasmids**. Human ACE2-coding plasmid (Addgene, 1786), pLVTHM/GFP (Addgene, 12247), psPAX2 (Addgene, 12260), pMD2.G (Addgene, 12259) were obtained from Addgene and pCMV6-LGALS3BP (OriGene, RC204918) from ORIGENE. pcDNA3, pAAV-CMV-GFP and pAVV-mCherry were obtained from L. Zentilin (Molecular Medicine Lab, International Centre for Genetic Engineering and Biotechnology, Trieste, Italy). The SARS-CoV-2 spike coding sequence (NCBI accession number NC_045512.2) was codon-optimized and synthesized with a V-5 tag at the C-terminus and then cloned in to the pZac 2.1 AAV (pAAV-Spike) vector under the control of a cytomegalovirus promoter. The last 19 amino acids at the C-terminus of the SARS-CoV-2 spike protein contain an endoplasmic reticulum retention sequence, which reduces the yield of spike-pseudoparticle production. For pseudoparticle production, a pAA-Spike-d19-V5 expression vector was generated, deleting the 19 amino acid endoplasmic reticulum retention signal through PCR amplification (primer sequences are listed in Supplementary Table 9), and cloned into the pAAV-Spike-V5 vector. The construct was verified by sequencing.

**Antibodies**. Antibodies against the following proteins were used: ACE2 (Abcam, ab15348), LGALS3BP (Abcam, ab217572), SARS-CoV-2 spike protein (GeneTex GTX632604), V5-488 (Thermo Fisher Scientific, 377500A488), α-tubulin (Sigma-Aldrich T5168), mouse-HRP (Abcam ab6789) and rabbit-HRP (Abcam ab205718).

**Small interfering RNA and SARS-CoV-2-spike coding plasmid transfections**. SiRNAs (siACE2, M-005755-00-0005; siNT1, non-targeting siRNA) were reverse transfected in 96 well plates, with the transfection reagent (Lipofectamine RNAi-MAX, Life Technologies) and siRNAs (25 nM) diluted in Opti-MEM (Life Technologies) after 5 min of incubation at room temperature. The transfection mixes were incubated for 30 min at room temperature and then added to the 96-well plates (CellCarrierUltra 96, Perkin Elmer). Reverse transfection of plasmids (pcDNA3, pmCherry, pLGALS3BP) was performed in 96-well plates, using a different amount for each plasmid. Totally, 25 ng to 100 ng of plasmids were diluted in 25 μL of Opti-MEM (Life Technologies) and mixed with the transfection reagent (FuGENE HD, Promega) using a ratio of 1 μg pDNA: 3 μl FugeneHD. The transfection mixes were incubated for 25 min at room temperature and added to the 96-well plates (CellCarrierUltra 96, Perkin Elmer). After 30 min, 6.5 × 10$^3$ Vero cells or 8 × 10$^3$ HEK293-ACE2 cells were seeded in each well. 24 h after transfection, 75 ng of either pEC117-Spike-V5 or pCMV-EGFP expression plasmids were transfected using a standard forward transfection protocol. After 24 h, cells were fixed in 4% PFA and processed for immunofluorescence.

**Pseudoparticle production and transduction**. An HIV-1-based replication-deficient lentiviral system was used to produce VSV-G and SARS-CoV-2 spike pseudotyped particles[92]. The particles were produced in HEK293T cells by co-transfecting 10 μg PLVTHM(GFP) plasmid, 5 μg psPAX2 (packing vector) and 5 μg pMD2.G (VSV-G pseudotyped particles) or pAAV-SΔc19 (Spike pseudotyped particles) using FugeneHD (Promega) transfection reagent in a ratio of 1 μg pDNA: 3 μl FugeneHD[92]. Viral supernatants were collected 48 h after transfection and centrifuged at 2105×g for 10 min at 4 °C. The supernatants were then filtered with a 0.45 μm pore size filter, aliquoted and stored at −80 °C. For pseudoparticle transduction, 24 h before transduction, both siRNAs and plasmids were transfected into the HEK293 cell expressing ACE2 by using a reverse transfection protocol. 24 h after transfection, equal amounts of VSV-G or Spike pseudoparticles were added to each treatment. After 36 h, cells were fixed, nuclei were labelled with Hoechst and assessed for pseudoparticle transduction efficiency based on GFP expression.

**Immunofluorescence**. After fixation in 4% PFA for 10 min at room temperature, cells were washed two times in 100 μL/well (96-well plate) of 1× PBS and then permeabilized in same volumes of 0.1% Triton X100 (Sigma-Aldrich 1086431000) for 10 min at room temperature. Cells were then washed two times with 1× PBS and blocked in 2% bovine serum albumin (BSA) for 1 h at room temperature. After blocking, 45 μL/well (96-well plate) of diluted primary antibody (1:500 in 1% BSA SARS-CoV-2 spike antibody or V5-488, and Flag/LGALS3BP antibody were added to each well and incubated overnight at 4 °C. Cells were then washed two times in 1× PBS, the 1× PBS was removed and 45 μL/well of diluted (1:500 in 1% BSA) secondary antibodies were added to each well and incubated 2 h at room temperature. Cells were then washed two times in 1× PBS. Nuclear staining was performed by Hoechst 33342 (1:5000) according to manufacturer's instruction.

**Image acquisition and analysis of syncytia and pseudoparticle entry**. Syncytia: Image acquisition was performed using the Operetta CLS high content screening microscope (Perkin Elmer) with a Zeiss 20× (NA = 0.80) objective, a total of 25 fields were acquired per wavelength, well and replicate (~10,000–15,000 cells per well and replicate). Images were subsequently analyzed, using the Harmony software (version 4.9, PerkinElmer). Images were first flatfield-corrected and nuclei were segmented using the "Find Nuclei" analysis module (Harmony software, version 4.9, PerkinElmer). The thresholds for image segmentation were adjusted according to the signal to background ratio. The splitting coefficient was set in order to avoid splitting of overlapping nuclei (fused cells). The intensity of the green fluorescence (spike/GFP) was calculated using the 'Calculate Intensity Properties' module (Harmony software, version 4.9, PerkinElmer). All cells that scored a nuclear area greater than 4 times the average area of a single nucleus and simultaneously showed a green signal (spike) in the cytoplasm are considered as syncytia. In the case of manual quantification of syncytia, >3 fused nuclei were counted as syncytia, when the cytoplasm showed a green signal (spike). Data are expressed as the percentage of fused cells by calculating the average number of fused cells normalized to the total number of cells per well. Pseudoparticle entry: Mean intensities of the segmented nucleus in the DAPI channel and the Hoechst channel for each nucleus across all fields were extracted. Each assay plate included siACE2 (siRNA targeting ACE2) as a positive control as well as pcDNA3 and siNT1 (non-targeting siRNA) as negative controls. Briefly, nuclei were segmented based on Hoechst staining, and cells were then classified as positive or negative depending on the GFP signal. Data are expressed as the percentage of GFP-positive cells by calculating the average number of GFP-positive cells normalized to the total number of cells per well.

**Western blotting**. After 48 h of transfection with siRNAs or plasmids, supernatants were collected, and cell debris was removed by centrifugation. HEK293 cell membranes were isolated using the Mem-PER™ Plus Membrane Protein Extraction Kit (Thermo Fisher, 89842), according to the manufacturer's instructions. Equal amounts of total cellular proteins (15–20 μg), as measured by BCA assay (Thermo Fisher, 23227), were resolved by electrophoresis in 4–20% gradient polyacrylamide gels (Mini-PROTEAN, Biorad) and transferred to nitrocellulose/PVDF membranes (GE Healthcare). Membranes were blocked at room temperature for 60 min with PBST (PBS + 0.1% Tween-20) containing 5% skim milk powder (Cell signalling, 9999). Blots were then incubated (4 °C, overnight) with primary antibodies against ACE2 (diluted 1:1000), LGALS3BP (diluted 1:1000) and α-tubulin (diluted 1:10,000). Blots were then washed three times (8 min each) with PBST. For standard Western blotting detection, blots were incubated with either an anti-rabbit HRP-conjugated antibody (1:5000) or an anti-mouse HRP-conjugated antibody (1:10,000) for 1 h at room temperature. After washing three times at room temperature with PBST (10 min each), blots were developed with Enhanced Chemiluminescence (ECL, Amersham).

**Machine learning**. Machine learning was deployed to identify a prognostic classifier for COVID-19 ICU patients based on 27 clinical variables, RNAemia, as well as ELISA measurements of candidates selected from the literature (PTX3[31–34], RAGE[28–30], and LGALS3BP). To ascertain the translational value of our machine learning results, we did not include our DIA MS-based data. The RNAemia feature was defined as a binary feature that takes a true value when RNAemia was present within six days upon admission to ICU. Feature selection was undertaken using an ensemble approach - feature filter for singleton variables followed by a wrapper method to evaluate binary and triplet combinations of these shortlisted singleton features (Supplementary Fig. 7). Mann–Whitney U test of statistical significance (P value < 0.05) was used as feature selection criterium for singleton markers (Fig. 4b, Supplementary Table 1). Binary and triplet combinatorial feature search was performed using wrapper feature selection[93] with support vector machine (SVM) classifier using radial basis function (RBF) kernel. Feature combinations were evaluated using the average of sensitivity, PPV and area under the receiver operating characteristic curve (ROC AUC) metrics. Although the F1-score, i.e., the harmonic mean of sensitivity and precision (PPV), is a commonly used evaluation metric for imbalanced data, the drawback is that F1-score does not reflect the correct classification of the majority class, i.e., true negatives. Combining ROC AUC along with sensitivity and PPV addresses this limitation of standalone usage of the F1-score. SVM uses hyperplane (decision surface) leveraging only a percentage of training samples (support vectors), thus offering high generalization ability attributed to its near impervious characteristic to new samples[94]. Combinations were restricted to a maximum of triplets to enhance the ease of clinical implementation and avoid the risk of overfitting. In addition, tenfold cross-validation along with leave-one-out validation was used to avoid overfitting and test model generalization. The SVM Synthetic Minority Oversampling Technique was used to prevent the learning bias of SVM RBF towards the majority class[95]. Tuning of the SVM RBF external parameter i.e., C was performed using grid search. The Scikit-learn default i.e., "scale" was used for the SVM RBF gamma parameter[96]. A permutation test was performed to evaluate the null hypothesis that the classifier performance is by chance i.e., input variables and outcome labels are independent[97]. Hence, rejection of the null hypothesis implies that the classifier has found a real class structure (pattern) in the data. For technical validation of our "Age, RNAemia" model based on SVM RBF, we employed a permutation test for statistical significance of the classifier performance; and stability of feature importance in an alternate machine learning feature ranking model, i.e., Random forest with resampling. Age and RNAemia were ranked among the top five most important features based on mean importance across 100 resampling cycles of sensitivity analysis. A permutation test with 50 permutes i.e., repeating the classification procedure after random permuting of the outcome labels returned a significant P value (Supplementary Fig. 8). The implementation of machine learning was done using Scikit-learn 0.23.2 python package[96].

**Statistical analysis**. All statistical analyses were two-tailed. Shapiro–Wilk normality test was extensively applied in proteomics and clinical data and some features were found to be not normally distributed. For this reason, nonparametric methods were used throughout the manuscript for differential expression analysis and correlation analysis. Mann-Whitney U significance test was used for continuous variables and Fisher exact test for binary variables in Supplementary Tables 1, 3 and 4. Statistical comparisons on MS and ELISA data were performed applying the nonparametric Mann–Whitney U test on the preprocessed data removing the residuals of age and sex from fitted linear models. Spike pulldown data were analyzed by paired or unpaired Student's t-tests as appropriate, because of the low sample size which makes the use of nonparametric tests not suitable. Timepoint comparisons were performed using the nonparametric Kruskal–Wallis test. Linear mixed models analysis was performed to further evaluate the combinatorial effect of time and RNAemia, and time and outcome respectively in the expression of significant proteins, correcting in all cases for age and sex. Correlations between continuous variables were analyzed using Spearman correlation. Correlations between categorical and continuous variables were examined using point-biserial correlation. Correlations between categorical variables were examined using Cohen's Kappa correlation. Anti-SARS-CoV-2 antibody data and trajectories of protein clusters were fitted using Generalized Alternative Models (GAM), with P values reporting the effect of RNAemia or mortality in the model because the trajectories of the antibody data have been demonstrated to be nonlinear and the same was observed for the protein clusters. Survival analysis was performed using Cox regression and Kaplan–Meier plots leveraging the R "survival" package. As two groups, i.e., low and high risk were being compared, no adjustments for multiple comparisons were performed in the Kaplan–Meier analysis. All features were scaled to a mean of zero and a standard deviation of one. Proteins quantified by MS and clinical features with missing values ≥30% were dropped and not used for data analysis. This resulted in two clinical variables being dropped, i.e., eosinophils and basophils. The remaining features were imputed, as applicable, using KNN based imputation with K = 5 (Supplementary Table 11)[91]. To validate DIA–MS findings, a publicly available proximity-extension assay proteomics-based dataset was analyzed (Data provided by the MGH Emergency Department COVID-19 Cohort (Filbin, Goldberg, Hacohen) with Olink Proteomics)[38]. Differential expression analysis of proteins in survivors and non-survivors 28-days after hospitalization within the Olink dataset[38] was achieved through the Ebayes method of the limma package since RQ values of Olink measurements were normally distributed. Benjamini-Hochberg's FDR corrected q-values were calculated to correct for multiple testing in all parts of the analysis: differential expression, correlation, survival. Statistical analysis and associated figures were generated with R programming environment (version 4.02), Python

programming environment (version 3.8.6) and GraphPad software (version 8.4.3). Schematic diagrams were created with Biorender.com.

**Reporting summary**. Further information on research design is available in the Nature Research Reporting Summary linked to this article.

## Data availability

The authors declare that the data supporting the findings of this study are available within the article and its Supplementary Data. The mass spectrometry proteomics data have been deposited to the ProteomeXchange Consortium via the PRIDE partner repository with the dataset identifiers PXD024026 and PXD024089 (https://www.ebi.ac.uk/pride/). We used the following human protein database: UniProtKB/Swiss-Prot version 2020 01, 20,365 protein entries (https://www.uniprot.org).

The external validation data was provided by the MGH Emergency Department COVID-19 Cohort (Filbin, Goldberg, Hacohen) with Olink Proteomics[38] (https://info.olink.com/mgh-covid-study-overview-page?utm_campaign=Broad%2520%2520Explore%2520Covid%2520Study&utm_source=research-gate&utm_medium=MGH%2520post). Source data are provided with this paper.

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

## Acknowledgements

We thank Dr. Victor Corman for his assistance and helpful comments.

## Financial support

M.M. and A.M.S. are British Heart Foundation (BHF) Chair Holders with BHF program grant support (CH/16/3/32406, RG/16/14/32397, and CH/1999001/11735, RE/18/2/34213, respectively). C.G., E.R., and B.S. are funded by BHF PhD studentships (FS/18/60/34181, FS/17/65/33481, and FS/19/58/34895). M.H. is funded by an interdisciplinary PhD studentship from King's BHF Centre of Research Excellence. M.F. is funded by a National Institute of Academic Anesthesia BJA-RCOA PhD Fellowship WKR0-2018-0047. M.J.W.M. is grateful to the Biomedical Research Centre at Guy's and St. Thomas' NHS Foundation Trust for support. K.O.G. is supported by a UK Medical Research Council Clinical Research Training Fellowship (MR/R017751/1). B.M. was supported by an NIHR Academic Clinical Fellowship in Combined Infection Training. K.J.D. was supported by a King's Together Rapid COVID-19 Call award, by an MRC Discovery Award (MC/PC/15068), Huo Family Foundation and the Fondation Dormeur, Vaduz. The NIHR Collaboration for Leadership in Applied Health Research and Care South

London at King's College Hospital NHS Foundation Trust, awarded to J.D.E. who is also supported by a charitable donation from the Lower Green Foundation. R.F.L. is funded by a Spanish Ministry of Economy and Competitiveness Grant BFU2017-87316. M.G. is supported by the European Research Council (ERC) Advanced Grant 787971 "CuRE" and by Program Grant RG/19/11/34633 from the BHF. M.M.'s research was made possible through the support of the BIRAX Ageing Initiative and funding from the EU Horizon 2020 research and innovation program under the Marie Skłodowska-Curie grant agreement No. 813716 (TRAIN-HEART), the Leducq Foundation (18CVD02), the excellence initiative VASCage (Centre for Promoting Vascular Health in the Ageing Community, project number 868624) of the Austrian Research Promotion Agency FFG (COMET program–Competence Centers for Excellent Technologies) funded by the Austrian Ministry for Transport, Innovation and Technology; the Austrian Ministry for Digital and Economic Affairs; and the federal states Tyrol (via Standortagentur), Salzburg, and Vienna (via Vienna Business Agency), two BHF project grant supports (PG/17/48/32956 and SP/17/10/33319) and the BHF Centre for Vascular Regeneration with Edinburgh/Bristol (RM/17/3/33381). M.M. and S.R.B. acknowledge support as visiting professors as part of the Transcampus TU Dresden King's College London Initiative. The work of A.C.H. is supported by a Cancer ImmunoTherapy Accelerator award from CRUK; the Wellcome Trust (106292/Z/14/Z); the Rosetrees Trust; King's Together Seed Fund; The John Black Charitable Foundation; Royal Society Grant IES\R3\170319 and the Francis Crick Institute, which receives core funding from Cancer Research UK (FC001093), the MRC (FC001093) and the Wellcome Trust (FC001093). M.S.H. is supported by the National Institute for Health Research Clinician Scientist Award (CS-2016-16-011). The research was funded/supported by the National Institute for Health Research (NIHR) Biomedical Research Centre based at Guy's and St Thomas' NHS Foundation Trust and King's College London. The views expressed are those of the author(s) and not necessarily those of the NHS, the NIHR or the Department of Health.

## Author contributions

C.G., K.T., S.A.B., B.S., K.Th., E.R., L.E.S., M.R., X.Y., S.B., J.D.E., A.M.S., S.R.B., T.T., A.C.H., M.G., M.S.H., and M.M. contributed to the study design, data interpretation, and writing of the paper. C.G., K.T., S.A.B., B.S., H.A., K.Th., M.H., L.E.S., C.C., U.N., M.S., K.D., R.F.L., R.K.W., K.J.D. contributed to the laboratory data generation and analysis. A.N., M.F., M.J.W.M., K.O.G., G.A., S.N., S.F.M., F.T., B.Sa., B.M., and M.S.H. contributed to the participant recruitment, sample collection, sample processing, and clinical data collection. All authors reviewed the paper.

## Competing interests

King's College London has filed and licensed a patent application with regard to using PTX3 as a biomarker in sepsis. King's College London has filed a patent application on the methods used to detect SARS-CoV-2 Spike protein-induced syncytia as described in this paper. A.C.H. is a board member and equity holder in ImmunoQure, A.G., and Gamma Delta Therapeutics, and is an equity holder in Adaptate Biotherapeutics.

## Additional information

[1]King's College London British Heart Foundation Centre, School of Cardiovascular Medicine and Sciences, London, UK. [2]King's College Hospital NHS Foundation Trust, London, UK. [3]Peter Gorer Department of Immunobiology, School of Immunology and Microbial Sciences, King's College London, London, UK. [4]Department of Intensive Care Medicine, Guy's and St Thomas' NHS Foundation Trust, London, UK. [5]Department of Inflammation Biology, School of Immunology and Microbial Sciences, Faculty of Life Sciences and Medicine, King's College London, London, UK. [6]Institute of Liver Studies, King's College Hospital, London, UK. [7]Clinical Infection and Diagnostics Research group, Department of Infection, Guy's and St Thomas' NHS Foundation Trust, London, UK. [8]NIHR Biomedical Research Centre, Guy's and St Thomas' NHS Foundation Trust and King's College London, London, UK. [9]Structural Biology Programme, Spanish National Cancer Research Centre (CNIO), Madrid, Spain. [10]Medical Clinic I, University Hospital Carl Gustav Carus, Technical University Dresden, Dresden, Germany. [11]Experimental Transfusion Medicine, Faculty of Medicine Carl Gustav Carus, Technical University Dresden, Dresden, Germany. [12]Department of Internal Medicine III, University Hospital Carl Gustav Carus, Technical University Dresden, Dresden, Germany. [13]Department of Diabetes, School of Life Course Science and Medicine, King's College London, London, UK. [14]Institute for Transfusion Medicine, German Red Cross Blood Donation Service North East, Dresden, Germany. [15]The Francis Crick Institute, London, UK. [16]These authors contributed equally: Clemens Gutmann, Kaloyan Takov, Sean A. Burnap, Bhawana Singh, Hashim Ali. ✉email: manu.shankar-hari@kcl.ac.uk; manuel.mayr@kcl.ac.uk

