## [Peer Review File · Nature Communications]

Reviewers' Comments:

Reviewer #1:

Remarks to the Author:

This is a very interesting study that investigates the relationship between RNAemia with COVID19 clinical outcomes and provide very informative data on the pull-down experiments to interrogate physical interaction of SARS-Cov-2 spike glycoprotein and host proteins. I think there are few aspects that need to be sorted out. The information concerning the pull-down experiments is very interesting and illuminates on interactions that to my knowledge have not been developed before.

There is another aspect that I find more complicated as it is presented now. Biomarkers are clinical instruments and I am concerned that the information provided by the authors lacks important data to contribute credibility to the study.

- 1) How did you define sepsis? The reason I am concerned about this definition is because if well-defined, sepsis seems to be a poor reference for covid-19. Indeed, while covid-19 patients develop important reminiscences to ARDS, sepsis and ARDS have different (and quite opposing) systemic molecular landscapes. ARDS hyperinflammatory landscape associates with higher mortality; in sepsis, SRS2 seems to be protective. Please address these issues appropriately.
- 2) In the introduction, authors state that APACHE II and sofa are "not discriminatory". I am confused by this statement as these scores are outcomes-predicting instruments, not tools to "discriminate" diagnoses. That is the reason why I need clarity of the definitions of disease in non-covid patients.
- 3) The fact that neither APACHEII, SOFA, albumin or FIO2 at enrollment were significantly associated with mortality is quite shocking...are the authors sure of these data? Were the patients equally hypoxemia at enrollment? This is something I need confirmation; having treated hundreds of these patients makes that data very unusual. It is ONLY the authors discovery what can predict outcomes in these patients? Can the authors provide mechanical ventilation data, including the dynamic compliance and PEEP value of these patients? Again, there seems to be some problem with the scoring system here or I am missing something big.
- 4) Can the authors contribute some Covid-specific risk score such as the COVID-GRAM score (JAMA Intern Med. 2020;180(8):1081-1089. doi:10.1001/jamainternmed.2020.2033)?

Other issues:

- 5) Regarding "pre-pandemic, non-covid-19 ISU sepsis patients": When were these samples collected? Before the pandemic? When were these patients analyzed? If before the pandemic, how did authors account or the batch effect?
- 6) Please explain how circulating SARS-CoV-2 RNA (RNAemia) differs from viral load. Why should that covariable surprise given previous evidence that larger viral load associated with poorer outcomes. Is there any correlation between viral load and RNAemia? Have these been compared at all?

Reviewer #2:

Remarks to the Author:

The detection of SARS-CoV-2 RNA in plasma has previously been associated with more severe disease and outcomes. In this study, the authors extend this observation in COVID ICU pats by assessing the relationship of RNAemia with antibody responses and proteomic profiling. In total, 295 samples were available from 78 ICU patients, including 18 who died. Results were compared to samples from 45 hospitalized non-ICU covid pts (5 died) and 55 ICU and non-ICU patients without COVID. Key findings include: 1) RNAemia was more common in ICU pts and associated with a higher risk of 28-day mortality; 2) RNAemia was associated with lower spike IgG and neutralization levels; 3) identification

of plasma proteomic clusters associated with increased mortality; 4) galectin-3 binding protein binds to the spike glycoprotein and is enriched in covid pts; and 5) machine learning approaches showed

improved predictors of mortality with a combination of RNAemia and either age or PTX3.

Overall, this study represents a valuable contribution to the literature, further highlighting the clinical importance of SARS-CoV-2 RNA in plasma while extending the analysis to include both Ab and proteomic analysis. The finding of lower Ab/neutralization levels in those with RNAemia speaks to a potential mechanism underlying the phenomenon and the association with proteomic clusters provides insights on down-stream effects of the RNAemia. Important limitations of this study include 1) the primary focus on critically-ill patients, which limits the generalizability of the findings to the general hospitalized or outpt populations; 2) the viral RNA measurements were non-quantitative as no standard curves were included; 3) while much was made of the longitudinal nature of the dataset (each ICU participant had an average of 3.8 samples collected), relatively little is reported on the actual trajectories of the RNAemia and an analysis on the duration of positivity.

Other questions/comments:

- 16 ICU covid participants had neutralizing Ab studies. It's still not fully clear to me how these individuals were chosen and the presence of overlaps between these groups? This is important to help the reader interpret the results. Also, not every participant had proteomic analysis, but it's a bit confusing how many actually had proteomics performed. Could the authors add the Ns to the right side of Fig 1?
- If I'm interpreting Fig 3 correctly, the authors are splitting the 2 ICUs into a Discovery and Validation dataset? The KCH cohort only has 12 participants. It's unclear to me whether this is a large enough sample size? Are there differences in ICU participants between the two hospitals?
- Do you have any data showing the heparinase treatment overcomes its inhibitory effects on qPCR?

Reviewer #3:

Remarks to the Author:

The manuscript 'SARS-CoV-2 RNAemia and proteomic biomarker trajectory informs prognostication in COVID-19 patients admitted to intensive care' is a very interesting and basically well designed study, featuring longitudinal assessment of patient plasma from COVID-19 patients requiring or not requiring ICU support, compared to non-COVID-19 patients either in the ICU for sepsis or simply in hospital prior to surgery. The controls are appropriate, and the longitudinal aspects of the study are truly unique. Cohorts were obtained from two hospitals and used for different purposes: the KCH cohort provided plasma that was used for assessment of RNAemia and for proteomic discovery using DIA-MS, using spiked in synthetic peptides representing 500 proteins. Serum was collected from the GSTT cohort and used for validation of leads from the plasma experiments; the make-up of the two cohorts was also slightly different, as the GSTT cohort included serum samples from pre-pandemic healthy controls. The objective was to identify biomarkers predictive of mortality within 28 days of entering the ICU, which is a highly relevant clinical goal.

Technically, the experiments are well executed. The use of heparinase to enhance the accuracy of qRT-PCR is a good control, and the evidence supporting increased RNAemia as a predictor of mortality is very solid. The 'discovery' proteomics effort uncovered associations between COVID infection and increased abundance of 29 plasma proteins, three of which were statistically associated with 28-day mortality [CFB, CPN11, SERPINA3]. Equally interesting is the failure to validate three previously implicated proteins: LBP, CD14, and ITIH3. The statistical associations observed with the complement cascade, platelet degranulation, acute phase response, and coagulation all make biological sense, and are well described in the Discussion, which is appropriately referenced.

One of the most interesting aspects of the manuscript is the attempt to identify the components of protein complexes formed on the SARS-CoV-2 spike glycoprotein. Pull-down experiments were conducted with His-tagged spike protein in plasma from the KCH COVID-19 ICU patients, and 32 proteins were identified by DIA-MS. Although most of the associated proteins were expected, galectin-

3-binding protein [LGALS3BP] was identified and also found to be elevated in plasma from COVID-19 ICU patients, compared to either non-ICU patients or ICU patients with sepsis. Validation of this observation was via DIA-MS on the GSTT patients, and also by ELISA on the same plasma samples. LGALS3BP abundance in COVID-19 patients was statistically associated with proteins of the complement cascade, platelet degranulation, and the innate immune system, although no experiments were done to determine the functional significance of the association or the mechanism leading to elevated LGALS3BP. Machine learning was used to identify the best predictors of differential survival in Kaplan-Meier plots, indicating that RNAemia was the best single predictor, but Age+PTX3 was almost as significant as Age+RNAemia.

Despite the clinical interest of these observations and the general strengths of the work, there are several significant weaknesses in the report that detract from the overall impact of the work:

1. It is not clear why both serum and plasma samples were used, and it is not always clear which substrate was used for which proteomic experiment. This might have inadvertently biased the results. Why were no plasma samples collected from COVID-19 patients in the ICU at GSTT?
2. There are no controls shown for the His-tagged spike glycoprotein pull-down, and thus it is impossible to judge which of the observed protein associations are biologically significant and reproducible. Although controls are described in the legend to Supplementary Table 7, we are not given the results of those control experiments. It would be relevant to know just how 'dirty' the pull downs were
3. Validation of the LGALS3BP-spike association actually appears to be merely a confirmation that the serum of ICU patients with COVID-19 had higher levels of LGALS3BP than controls or sepsis patients. This does not confirm a functional role for LGALS3BP in the spike glycoprotein complex. Some orthogonal measurement on the pull-downs should have been shown, whether Western or ELISA.
4. The investigation of PTX3 as a prognostic factor for survival appears to come solely from prior work by the authors and others; no data on the abundance of PTX3 in either plasma or serum from the KCH and GSTT cohorts is shown in the main manuscript. Given the data that are shown in Supplementary Material [Figure S6b], PTX3 appears to be a rather non-specific marker of 'sick enough to be in the ICU'. Therefore, its inclusion in the machine learning models, and the significance of its performance in the machine learning models, is highly doubtful – it may simply be an indicator of being on a ventilator. The authors need to be very up front with these observations in the main manuscript
5. The concerns about the promotion of PTX3 as a prognostic factor for clinical outcome in COVID-19 are exacerbated by the lack of any experimental detail regarding the 'machine learning' method used to determine the efficacy of 'Age, PTX3' as a signature of COVID-19 mortality. The entire description of the machine learning exercise appears to be contained in Legends to Supplementary Figure 8 and Supplementary Table 9, with no justification given for use of the SVM RBF model. It is also not precisely clear how the 'leave one out validation' was conducted, as the main text simply refers to 'the external validation cohort of hospitalized COVID-19 patients described above'. It is not clear whether this refers to the identification of PTX3 as a prognostic marker, to the identification of an SVM signature, or both.

Reviewer #4:

Remarks to the Author:

In " SARS-CoV-2 RNAemia and proteomic biomarker trajectory inform prognostication in COVID-19 patients admitted to intensive care " Gutmann et al describe the analysis of viral RNA and circulating proteins in blood samples collected from intensive care patients suffering from COVID-19. While the study provides valuable insights into longitudinal molecular phenotypes of the disease, there are, unfortunately, some substantial shortcomings that need to be addressed.

Major concerns:

As a reviewer and first-time reader of the manuscript, I found the work to be rather erratic and an

assembly of possible routes to analyse the available samples and data. It remained unclear what the real value and contribution of the study would be and how others could learn from the findings. Even though the topic is timely, the team of renown expertise, and the presented data of good quality, I missed a coherent message. The work as whole was often presented as an assembly of different projects and capabilities. Many different types of analyses have been done, there are two inconsistent sample sets (one serum, one plasma), but no common core hypothesis to centre the project(s).

Another concern were the different levels of care taken to choose appropriate statistical methods. From simple t-tests to SVM approaches, there is no good alignment and justification for using these. In particular the proteomics part suffers from choosing the simplest but certainly not most appropriate tools, check for normality and account for the small sample size versus the larger number of analytes. I remain particularly critical to using static correlation analysis tools to infer causality, and not include informative co-variables into association by LMMs (or similar) to learn more about what the data actually adds on top of age, sex, CRP, ALT or other abundant blood proteins. Moreover, the dichotomous analyses suffer greatly from imbalanced group sizes (eg. 18 vs 60), making findings less likely to be replicated elsewhere. It is good to correct p-values for multiple testing, but please include also those from the correlation analyses.

For the proteomics part, and as other MS-based COVID-19 proteomics studies have indicated, most of the proteins altered in the circulation as a response to SARS-CoV-2 infections are derived from the liver. Even though proteins of the coagulation system have been highlighted, their main source is the liver and their activity and abundance can be influenced by heat treatment (precipitation).

From my point of view, it would have added more value to more rigorously compare protein levels with the clinical data that otherwise remained hidden in the supplementary. To properly judge these relationships, qq-plots should be provided for all variables with all proteins. From there, thorough power calculations will lead to more valuable lists.

Longitudinal analysis is challenging but given that most patient will receive treatment adjusted to their health status, time as a variable is less appropriate and most likely overruled by other factors (eg medication dose, ventilation, days in bed/ICU, comorbidity). All figures of Fig 4 should therefore also include the clinical traits and consider these as informative components. It may well be that the two groups listed in cluster 4 of Fig 4c are simply reflecting age+sex instead of survival.

While the IP experiments of the spike protein is interesting, there are a few issues that need to be solved. First, a crapome (see PMID: 23921808) needs to be defined as many of the proteins found may likely be common and abundant contaminants. Others like PMID: 31171813 used plasma instead of cell lysates and illustrated the advantage of z-scores calculated from a large number of IPs as a measure of enrichment over a population of common contaminants. Secondly, a mock protein needs to be used instead of bare beads to reflect a bait similar to spike. Thirdly, dilution series of serum/plasma samples need to be conducted to learn more about the dynamic of the interaction and whether enrichment is based on the very high abundance of LGALS3BP or the more selective affinity. Forth, it was unclear if heat treatment of the samples has induced the reported interaction. Again, PMID: 31171813 may provide some helpful leads on this matter.

It remained unclear why serum or plasma samples have been collected at the centres and why no common sample processing protocols have been agreed upon prior to the analyses. The two preparation types must be regarded as different and cannot serve to compare collections that use either of these. It remains impossible to state whether the phenotypes were different due to disease or sample collection.

Further, it was unclear if only the COVID-19 samples were heat treated. As has been shown by others, heat treatment affects the protein composition and deactivates not only the virus' infection potential but also the components of the coagulation/complement system. Hence, a comparison between

COVID19, sepsis, and pre-pandemic samples may, in addition to the phenotype, be further influenced and biased by sample pre-treatment. Age of sample (time in the freezer) is another important aspect to consider in such cross-sectional analysis (PMID: 31573204).

While the KM plots indicate some added value of the discussed variables, it remains unclear what PTX3 and RNAemia add to age, sex, etc. It is preferred to conduct conditional analysis than combinatorial models adding in variables for the best apparent performance. It is more important to understand the independent contribution of the variables and learn about the mechanism than conducting a race for the best mathematical AUCs.

It also remains unclear why PTX3 was included into this story after it has not been detected in the MS-based proteomics. Benchmarking the work with the publicly data sets is good but again, this has further blurred the manuscript's message.

Other comments:

Please avoid statements such as " To the best of our knowledge, this is the largest longitudinal assessment of RNAemia,... " as this main also be the only study and it does not increase the value given its pitfalls.

Why did the authors stop providing p-values after S-table 4? These two groups seem to be very different in their demographics.

S-table 1: Among the clinical variables, ALT levels of survivors vs non-survivors also appear different but, surprisingly, have passed the significance threshold. Please provide visual representation and explanation of this.

Figures:

Please add the number of samples to each group in all figures!

All heat maps are only partially informative because they lack dendrograms. It remains impossible to judge distances between the sub-clusters, where and why the groups were categorised as highlighted. Please sort all HR plots per ratio and not the alphabet

REVIEWER COMMENTS

Reviewer #1 (Remarks to the Author):

This is a very interesting study that investigates the relationship between RNAemia with COVID19 clinical outcomes and provide very informative data on the pull-down experiments to interrogate physical interaction of SARS-Cov-2 spike glycoprotein and host proteins. I think there are few aspects that need to be sorted out. The information concerning the pull-down experiments is very interesting and illuminates on interactions that to my knowledge have not been developed before.

We thank the reviewer for the positive comments. Indeed, a plasma protein pull-down experiment using SARS-CoV-2-spike glycoprotein has not been done before. In the revised version, we have now added functional data to demonstrate the antiviral activity of LGALS3BP see new Fig. 6.

There is another aspect that I find more complicated as it is presented now. Biomarkers are clinical instruments and I am concerned that the information provided by the authors lacks important data to contribute credibility to the study.

1) How did you define sepsis? The reason I am concerned about this definition is because if well-defined, sepsis seems to be a poor reference for COVID-19. Indeed, while COVID-19 patients develop important reminiscences to ARDS, sepsis and ARDS have different (and quite opposing) systemic molecular landscapes. ARDS hyperinflammatory landscape associates with higher mortality; in sepsis, SRS2 seems to be protective. Please address these issues appropriately.

Regarding sepsis definitions:

Sepsis was defined according to Sepsis-3 definitions (infection with organ dysfunction defined using SOFA score 2 or more points). The eligibility criteria for the cohort study and the study protocol have been published (please see: <https://doi.org/10.1177/1751143720966286>). We have now included this information in the section "Methods" -> "Study design and recruitment".

Regarding sepsis reference point:

The underlying principle of the sepsis-3 definition is that it is infection-related organ dysfunction. Infection could be viral, bacterial, fungal, protozoal. This concept was used in the Global Burden of Diseases paper by Rudd and Colleagues. Please see: [https://doi.org/10.1016/S0140-6736\(19\)32989-7](https://doi.org/10.1016/S0140-6736(19)32989-7)

Regarding different and quite opposing molecular landscapes:

We considered this point highlighted by the reviewer. Whilst there are molecular differences, we observed numerous similarities in the immune responses between COVID-19 and sepsis. Please see our recent work: Laing et al. *Nat Med* 2020: <https://doi.org/10.1038/s41591-020-1038-6> . Furthermore, the molecular subphenotypes of ARDS proposed by Calfee and Colleagues were replicated in COVID-19 ARDS, again highlighting similarities and mortality are within the confidence interval bounds. Please see: Sinha et al. *Lancet Respir Med* 2020 [https://doi.org/10.1016/S2213-2600\(20\)30366-0](https://doi.org/10.1016/S2213-2600(20)30366-0)

Regarding SRS phenotypes:

We recently summarized the subphenotypes of sepsis and ARDS. There are clinical and molecular phenotypes reported in the literature. The most consistent signal is within the ARDS subphenotypes. In contrast our report highlights up to 6 clinical subphenotypes of sepsis and between 2-6 molecular subphenotypes of sepsis. The SRS2 phenotype is one among these different phenotypes, which was derived using microarray data and using a relative comparison within a sepsis cohort, without a control cohort. Please see: Shankar-Hari M and Rubenfeld G. *Curr Opin Crit Care* (2019) <https://doi.org/10.1097/MCC.0000000000000641>

2) In the introduction, authors state that APACHE II and sofa are "not discriminatory". I am confused by this statement as these scores are outcomes-predicting instruments, not tools to "discriminate" diagnoses. That is the reason why I need clarity of the definitions of disease in non-COVID patients.

Thank you. We have revised the manuscript for clarity. We are using the term for outcome prediction.

3) The fact that neither APACHEII, SOFA, albumin or FIO2 at enrolment were significantly associated with mortality is quite shocking...are the authors sure of these data? Were the patients equally hypoxemia at enrolment? This is something I need confirmation; having treated hundreds of these patients makes that data very unusual. It is ONLY the authors discovery what can predict outcomes in these patients? Can the authors provide mechanical ventilation data, including the dynamic compliance and PEEP value of these patients? Again, there seems to be some problem with the scoring system here or I am missing something big.

Using UK COVID-19 ICU data, Prof Shankar-Hari was part of the UK team that assessed the prognostic variables using the TRIPOD guideline-based approach. The above statements hold true regarding association with mortality, but these scores do not discriminate survival.

Please see Ferrando-Vivas et al. *Crit Care Med* 2021: <https://doi.org/10.1097/CCM.0000000000004740>

0) Can the authors contribute some Covid-specific risk score such as the COVID-GRAM score (JAMA Intern Med. 2020;180(8):1081-1089. doi:10.1001/jamainternmed.2020.2033)?

We appreciate and thank the reviewer for pointing us to the COVID-GRAM score. This score is based on a study by Liang et al *JAMA Intern Med* 2020: <https://doi.org/10.1001/jamainternmed.2020.2033>. Hospitalized patients with varying disease severity were used as a study population. Its purpose is “to help identify patients with COVID-19 who may subsequently develop critical illness”.

We would like to highlight that all of our ICU patients were already critically ill and the applicability of the COVID-GRAM score is therefore unclear. Given that our main focus was on ICU patients, we collected data for the two most established outcome-predicting scores in ICU patients: APACHE II and SOFA. We would also like to draw attention of the reviewer to the point that the website that hosts the COVID-GRAM score (<http://118.126.104.170/>) is not secure and unfortunately does not meet the security compliance of King's College London to use this website.

4) Regarding “pre-pandemic, non-covid-19 ISU sepsis patients”: When were these samples collected? Before the pandemic? When were these patients analyzed? If before the pandemic, how did authors account or the batch effect?

To avoid batch effects, all samples presented in the current manuscript were analyzed in the same batch. Processed samples were stored at -80°C prior to analysis. Patient recruitment of the “pre-pandemic, non-COVID-19 ICU sepsis” cohort occurred between October 16, 2019 and February 26, 2020. Patient recruitment of the “pre-pandemic, non-COVID-19 control” cohort occurred between July 8, 2019 and September 9, 2019. We have now included these dates in the respective methods section for clarification. Both cohorts were recruited at Guy's and St Thomas' NHS Foundation Trust (GSTT), London, UK.

5) Please explain how circulating SARS-CoV-2 RNA (RNAemia) differs from viral load. Why should that covariable surprise given previous evidence that larger viral load associated with poorer outcomes. Is there any correlation between viral load and RNAemia? Have these been compared at all?

Regarding associations of RNAemia and nasopharyngeal viral load with mortality:

Increased nasopharyngeal viral load has indeed been associated with higher mortality (please see Pujadas et al, *Lancet Respir Med* (2020): [https://doi.org/10.1016/S2213-2600\(20\)30354-4](https://doi.org/10.1016/S2213-2600(20)30354-4). The increase in risk, however, was small (HR, 1.07 [95% CI, 1.03–1.11], $n = 1,145$). In our study, we demonstrate that the risk increase conferred by RNAemia is substantially higher even in critically ill patients admitted to ICU (HR, 1.84 [95% CI, 1.22–2.77] adjusted for age and sex). This expands on recent reports in hospitalized patients by Fajnzylber et al *Nat Comm* (2020): <https://doi.org/10.1038/s41467-020-19057-5> with a HR of 5.5, $P = 0.02$ and by Prebensen et al *Clin Infect Dis* (2020): <https://doi.org/10.1093/cid/ciaa1338> with a HR of 6.0 [95% CI, 2.3-15.6]. Thus, RNAemia appears to be a better predictor for poor outcome than nasopharyngeal viral load. The two recent references for RNAemia in

hospitalized patients have been added. Please note that in our comparison RNAemia in hospitalized patients is lower (4%) than in ICU patients (23%).

Regarding correlations between nasopharyngeal viral load and RNAemia:

The study by Fajnzylber et al *Nat Comm* (2020): <https://doi.org/10.1038/s41467-020-19057-5> assessed correlations between viral load in different body compartments: “nasopharyngeal vs oropharyngeal Spearman’s $r = 0.34$, $P = 0.03$; nasopharyngeal vs. sputum $r = 0.39$, $P = 0.03$, oropharyngeal vs. sputum $r = 0.56$, $P = 0.001$, Supplementary Fig. 2). Plasma viral load was modestly associated with both nasopharyngeal ($r = 0.32$, $P = 0.02$) and sputum viral loads ($r = 0.36$, $P = 0.049$), but not significantly associated with oropharyngeal viral loads. There was no significant association between urine viral load and viral loads from any other sample types”. This suggests that viral load in the nasopharyngeal compartment ($r^2=10.24\%$) or sputum ($r^2=12.96\%$) only account for a minor part of the variation in plasma, suggesting that RNAemia is not predominantly a consequence of high viral load in other compartments.

This is now better explained in the revised version of the manuscript.

Reviewer #2 (Remarks to the Author):

The detection of SARS-CoV-2 RNA in plasma has previously been associated with more severe disease and outcomes. In this study, the authors extend this observation in COVID ICU pats by assessing the relationship of RNAemia with antibody responses and proteomic profiling. In total, 295 samples were available from 78 ICU patients, including 18 who died. Results were compared to samples from 45 hospitalized non-ICU covid pts (5 died) and 55 ICU and non-ICU patients without COVID. Key findings include: 1) RNAemia was more common in ICU pts and associated with a higher risk of 28-day mortality; 2) RNAemia was associated with lower spike IgG and neutralization levels; 3) identification of plasma proteomic clusters associated with increased mortality; 4) galectin-3 binding protein binds to the spike glycoprotein and is enriched in covid pts; and 5) machine learning approaches showed improved predictors of mortality with a combination of RNAemia and either age or PTX3.

Overall, this study represents a valuable contribution to the literature, further highlighting the clinical importance of SARS-CoV-2 RNA in plasma while extending the analysis to include both Ab and proteomic analysis. The finding of lower Ab/neutralization levels in those with RNAemia speaks to a potential mechanism underlying the phenomenon and the association with proteomic clusters provides insights on down-stream effects of the RNAemia.

We thank the reviewer for the positive comments.

Important limitations of this study include 1) the primary focus on critically-ill patients, which limits the generalizability of the findings to the general hospitalized or outpt populations; 2) the viral RNA measurements were non-quantitative as no standard curves were included; 3) while much was made of the longitudinal nature of the dataset (each ICU participant had an average of 3.8 samples collected), relatively little is reported on the actual trajectories of the RNAemia and an analysis on the duration of positivity.

Regarding the primary focus on critically-ill patients:

In our opinion, the focus on critically-ill patients is a particular strength, because ICU patients remain understudied and have the highest mortality. Studies investigating RNAemia were predominantly focused on hospitalized patients with varying disease severity, *i.e.* by Fajnzylber et al *Nat Comm* (2020): <https://doi.org/10.1038/s41467-020-19057-5> and by Prebensen et al *Clin Infect Dis* (2020): <https://doi.org/10.1093/cid/ciaa1338>.

Similarly, previous studies investigating the circulating proteome of COVID-19 patients focused on hospitalized patients. Proteome changes were associated with disease severity. In our study we highlight that few of the plasma protein changes associated with disease severity also predict outcome in ICU patients who are already critically ill.

The underrepresentation of studies in ICU patients suggests that samples from hospitalized COVID-19 patients are more readily accessible. We would also like to highlight that no previous study has combined the measurements of RNAemia and proteomics. This, together with the focus on ICU patients, are only two of the unique aspects of our study. The pull-down experiments from plasma another as pointed out by reviewer 1. In the revised version, we have now also added functional data to demonstrate the antiviral activity of LGALS3BP – see new **Fig. 6**.

Regarding the viral RNA measurements:

In RNAemia-positive patients, abundance of viral RNA is low (mean Cq: 34.4; range: 29.8-37.6) consistent with studies by others (please see: <https://doi.org/10.12688/wellcomeopenres.16002.2>). Thus, we analyzed RNAemia as a categorical variable (detectable yes or no): its presence alone represents a defined risk factor and information on differences in the low abundance range of viral RNA observed in blood would not increase its predictive performance. Moreover, there is no consensus on which RNA standards to use for absolute SARS-CoV-2 quantification (copies/ml). cDNA standards cannot be used for RNA viruses due to variable reverse transcription efficiency. Copy numbers estimated by other experimental approaches can lead to reproducibility problems. Previous studies reporting absolute quantification, *i.e.*

- Pujadas et al *Lancet Respir Med* (2020): [https://doi.org/10.1016/S2213-2600\(20\)30354-4](https://doi.org/10.1016/S2213-2600(20)30354-4)) for instance used diluted RNA extracted from cell lines.

- Fajnzylber et al *Nat Comm* (2020): <https://doi.org/10.1038/s41467-020-19057-5> reported RNAemia both as a continuous and categorical variable, although their protocol does not mention whether they used an RNA or cDNA standard. In the study by Fajnzylber et al, the association of RNAemia with mortality was lower when analyzed as a continuous variable (HR: 2.4, $P = 0.04$), compared to when it was assessed as a binary variable (HR: 5.5, $P = 0.02$).
- In our study, we did not detect any differences in Cq-values between survivors and non-survivors (**Supplementary Fig. 13**), refuting a major dose-dependency within the low abundance range of viral RNA. It appears that virus dissemination in blood per se is a predictor of poor outcome.

Regarding the longitudinal nature of the dataset (each ICU participant had an average of 3.8 samples collected): Thank you for the suggestion. The longitudinal design is another strength of our study. Most proteomics studies to date are using samples from single time points. The trajectories of protein changes have already been highlighted in the manuscript. We now provide an additional Supplementary Figure (**Supplementary Fig. 2**), where we show that RNAemia frequency is higher early after symptom onset and becomes less frequent at later stages.

- 16 ICU covid participants had neutralizing Ab studies. It's still not fully clear to me how these individuals were chosen and the presence of overlaps between these groups? This is important to help the reader interpret the results. Also, not every participant had proteomic analysis, but it's a bit confusing how many actually had proteomics performed. Could the authors add the Ns to the right side of Fig 1?

Regarding antibody measurements:

38 samples from 16 ICU COVID-19 patients were used in an HIV-1-based pseudotype neutralization assay to validate the neutralization results we obtained by the surrogate virus neutralization test (sVNT). Given that this was just a technical validation, we selected these samples at random.

- IgG measurements were performed in the entire cohort.
- Neutralization measurements with the sVNT kit were not performed in the entire cohort because the delivery of an additional sVNT kit was substantially delayed. We therefore excluded some non-RNAemic survivors, because this was by far the largest patient group. In the meantime, we measured the remaining samples. As expected, the previously observed effect remained the same. The right panel of Figure 2e was updated accordingly
- We have now stated the exact N-numbers in the manuscript.
- Note that for antibody analysis only patients with information on the days post onset of symptoms (POS), age and sex were included (70 of 78 COVID-19 ICU patients, totaling 261 samples).

Regarding proteomics:

To obtain the most reliable proteomics results all samples were processed and analyzed in parallel. Four patients of the KCH cohort were recruited at a later time point, when the proteomics analysis had already been conducted. We wanted to avoid batch effects and did not include these additional four patients (**Fig. 3**). In our proteomics time course experiment (**Fig. 4**) we only included those patients from the GSTT cohort, for which samples from all three time points were available (47 of 61 patients). An explanation has been added to the Methods section. Exact N-numbers are now stated for each experiment.

Regarding N-numbers in Fig. 1:

We have added a detailed sample allocation figure as **Supplementary Fig. 1** in addition to providing the exact N-numbers in each Figure legend. We hope this is acceptable to the reviewer.

- If I'm interpreting Fig 3 correctly, the authors are splitting the 2 ICUs into a Discovery and Validation dataset? The KCH cohort only has 12 participants. It's unclear to me whether this is a large enough sample size? Are there differences in ICU participants between the two hospitals?

As highlighted in **Fig. 3a** and **Supplementary Fig. 4**, the differences between COVID-19 ICU patients and control patients are very pronounced. These n -numbers were sufficient as the comparison reproduced many protein changes previously linked to COVID-19 severity (100 protein changes, $q < 0.05$). A unique aspect of our study, however, that we used another non-COVID-19 sepsis cohort as additional control. For discovery, we compared plasma samples from KCH at baseline ($n = 12$), from patients undergoing elective cardiac surgery ($n = 30$) and from a pre-pandemic sepsis ICU cohort ($n = 12$). The validation of the trajectory of differentially expressed proteins

in COVID-19 and their association with outcome was done in the larger GSTT COVID-19 ICU cohort ($n = 62$). In total, we analyzed serial samples from 74 COVID-19 ICU patients by proteomics. For comparison, here are the n-numbers from other proteomics studies that included COVID-19 patients with severe disease:

Shen et al, Cell , 2020:	n=28 severe
Messner et al, Cell Systems , 2020:	n=15 severe (WHO-5 to 7)

With regards to differences between the two hospitals, KCH samples were plasma whereas GSTT collected serum. Discovery in plasma and validation in serum ensures that the reported protein changes are independent of the sample type, which is important for the generalizability of our findings. Another COVID-19 proteomics study placed on a preprint server also utilized this approach whereby a discovery cohort utilized plasma and serum was used in a separate validation cohort: <https://doi.org/10.1101/2020.11.09.20228015>.

- Do you have any data showing the heparinase treatment overcomes its inhibitory effects on qPCR?

The effect of heparinase treatment on RT-qPCR measurements of plasma/serum samples from heparinized patients is well documented by us and others (please see: <https://doi.org/10.1161/CIRCRESAHA.119.314937> ; <https://doi.org/10.1160/TH13-05-0368> ; <https://doi.org/10.1016/j.jviromet.2011.05.012>). As a precaution we treated all our COVID-19 samples with the heparinase protocol, so we cannot compare it to untreated samples.

Reviewer #3 (Remarks to the Author):

The manuscript 'SARS-CoV-2 RNAemia and proteomic biomarker trajectory informs prognostication in COVID-19 patients admitted to intensive care' is a very interesting and basically well-designed study, featuring longitudinal assessment of patient plasma from COVID-19 patients requiring or not requiring ICU support, compared to non-COVID-19 patients either in the ICU for sepsis or simply in hospital prior to surgery. The controls are appropriate, and the longitudinal aspects of the study are truly unique. Cohorts were obtained from two hospitals and used for different purposes: the KCH cohort provided plasma that was used for assessment of RNAemia and for proteomic discovery using DIA-MS, using spiked in synthetic peptides representing 500 proteins. Serum was collected from the GSTT cohort and used for validation of leads from the plasma experiments; the make-up of the two cohorts was also slightly different, as the GSTT cohort included serum samples from pre-pandemic healthy controls. The objective was to identify biomarkers predictive of mortality within 28 days of entering the ICU, which is a highly relevant clinical goal.

We thank the reviewer for the positive comments. Just to clarify the prepandemic sepsis controls were plasma samples. They were recruited at GSTT just before the COVID-19 outbreak in the UK.

Technically, the experiments are well executed. The use of heparinase to enhance the accuracy of qRT-PCR is a good control, and the evidence supporting increased RNAemia as a predictor of mortality is very solid. The 'discovery' proteomics effort uncovered associations between COVID infection and increased abundance of 29 plasma proteins, three of which were statistically associated with 28-day mortality [CFB, CPN11, SERPINAA3]. Equally interesting is the failure to validate three previously implicated proteins: LBP, CD14, and ITIH3. The statistical associations observed with the complement cascade, platelet degranulation, acute phase response, and coagulation all make biological sense, and are well described in the Discussion, which is appropriately referenced.

Again, we thank the reviewer for the insightful comments. We believe it is important to demonstrate that protein changes related to severity of COVID-19 do not necessarily inform on outcome in already critically-ill patients.

One of the most interesting aspects of the manuscript is the attempt to identify the components of protein complexes formed on the SARS-CoV-2 spike glycoprotein. Pull-down experiments were conducted with His-tagged spike protein in plasma from the KCH COVID-19 ICU patients, and 32 proteins were identified by DIA-MS. Although most of the associated proteins were expected, galectin-3-binding protein [LGALS3BP] was identified and also found to be elevated in plasma from COVID-19 ICU patients, compared to either non-ICU patients or ICU patients with sepsis. Validation of this observation was via DIA-MS on the GSTT patients, and also by ELISA on the same plasma samples. LGALS3BP abundance in COVID-19 patients was statistically associated with proteins of the complement cascade, platelet degranulation, and the innate immune system, although no experiments were done to determine the functional significance of the association or the mechanism leading to elevated LGALS3BP. Machine learning was used to identify the best predictors of differential survival in Kaplan-Meier plots, indicating that RNAemia was the best single predictor, but Age+PTX3 was almost as significant as Age+RNAemia.

Indeed, a plasma protein pulldown experiment using SARS-CoV-2-spike glycoprotein has not been done before. In the revised version, we have now added functional data to demonstrate the antiviral activity of LGALS3BP – see new Fig. 6.

1. It is not clear why both serum and plasma samples were used, and it is not always clear which substrate was used for which proteomic experiment. This might have inadvertently biased the results. Why were no plasma samples collected from COVID-19 patients in the ICU at GSTT?

While utilizing plasma samples to determine a COVID-19 related protein signature as shown in Fig. 3a, we then opted to determine whether the proteins within this signature were predictive of 28-day outcomes in the serum of patient samples from the, larger, GSTT cohort. The use of two sample matrices from COVID-19 patients we believe actually provides strength to the subsequent confirmed prognostic markers, particularly due to both plasma and serum collection being common practice in the clinic and therefore a marker that is consistent across both matrices

provides better generalizability of the findings. Similarly, a recent COVID-19 proteomics study placed on a preprint server also utilized this approach whereby a discovery cohort utilizing plasma led to the use of serum in a separate validation cohort (preprint available at <https://doi.org/10.1101/2020.11.09.20228015>).

A good example of matrix effects on biomarker concentrations is actually PTX3. PTX3 was determined to be one of the best prognostic markers in the O-Link validation dataset utilized in our study (preprint available at <https://doi.org/10.1101/2020.11.02.365536>). This dataset was generated from plasma and the fact that we see a similar strength in survival prediction by PTX3 in our cohort, including the GSTT serum samples, adds strength to the potential of PTX3 as a biomarker for COVID-19 survival. PTX3 levels, however, were different between serum and plasma (**Fig. 7b, Supplementary Fig. 10b**). This information was critical for the machine learning approach (**Supplementary Table 8**). Matrix effects on biomarkers would have gone unnoticed if only plasma or serum would have been used and will be important for determining reference values for clinical applicability.

2. There are no controls shown for the His-tagged spike glycoprotein pull-down, and thus it is impossible to judge which of the observed protein associations are biologically significant and reproducible. Although controls are described in the legend to Supplementary Table 7, we are not given the results of those control experiments. It would be relevant to know just how ‘dirty’ the pull downs were.

We thank the reviewer for raising this important point. All possible controls were performed:

- 1) The recombinant spike protein was pulled down and analyzed by proteomics to exclude contaminants that could have been carried over from the HEK293 cells in which spike was expressed. We identified 96 proteins which could have been carried over from the spike preparation. Note that we applied a stringent threshold, so any proteins enriched less than five times in our plasma pull-downs compared to the “mock” spike control were excluded from subsequent analyses. We have now included an additional **Rebuttal Table 1** to list all the carryover proteins which we excluded from analyses.
- 2) Magnetic bead pull-down leads to non-specific isolation of proteins binding to the solid phase. Spike plasma sample pull-downs were compared against “crapome” of the bead-only as suggested by Reviewer 4.
- 3) To further rule out unspecific binding, we have performed an additional control: bead-only plasma incubation without adding exogenous spike. This was done for each of the eight biological samples providing internal control for every biological replicate. Hence, we have identified the “crapome” in our own experiments which we have now included in the **Rebuttal Table 2**. This combined with a sample size of $n = 8$, provides solid evidence that the findings are not merely a contamination issue. Results shown on the Volcano plots on **Fig. 5** are taking all of the above into account.
- 4) To further address the reviewer’s concern, we have performed additional pull-downs using PCSK9 as another, irrelevant recombinant protein with the same magnetic bead pull-down approach (**Rebuttal Fig. 6**). LGALS3BP, immunoglobulin or complement proteins were not pulled down with PCSK9.

We now provide additional information in the revised version.

Rebuttal Table 1: Protein contaminants identified in SARS-CoV-2 spike glycoprotein pull-down. List of the 96 proteins detected within the pull-down of His-tagged SARS-CoV-2 spike glycoprotein preparation by MS that were subsequently excluded as contaminants in downstream analyses.

A2M	ECM1	KRT13	NID2
AGRN	ENO1	KRT14	NUCB1
AHSG	F5	KRT16	PAM
ALB	F8	KRT17	PAPLN
ALDOB	FBLN1	KRT2	PKM
ANG	FBN1	KRT4	PRDX1
ANXA2	FBN2	KRT5	PRSS1

APOA1	FLG	KRT6A	PRSS3
APOC3	FLG2	KRT6B	SAFB
AZGP1	GAPDH	KRT74	SBSN
BAG3	GC	KRT78	SERPINA4
C1QTNF1	GRN	KRT80	SF3A1
C4orf48	HBB	KRT9	SF3A2
CASP14	HRC	LAMB2	SF3A3
CAT	HRNR	LAMC1	SF3B4
CCNG1	HSPA1A	LBP	SLC39A10
CFH	HSPA5	LMNA	SLC39A6
COL18A1	HSPG2	LOXL2	STC2
COL2A1	IGFBP4	LTBP1	TF
CSTA	IGFBP5	LTBP3	TGM1
DCD	ITIH3	LTBP4	TGM3
DSC1	JUP	NELL1	TNC
DSG1	KRT1	NELL2	TUBA1B
DSP	KRT10	NID1	UACA

Rebuttal Table 2: “Crapome” proteins identified in pulldowns of plasma without addition of SARS-CoV-2 spike glycoprotein. List of the 82 proteins detected within the Dynabeads His-tag plasma pulldown preparation by MS without addition of recombinant SARS-CoV-2 spike.

A1BG	CD5L	FN1	PROS1
ABCB9	CFB	GSN	RNGTT
ACTN1	CFI	HBA1	S100A8
AGT	CLU	HP	S100A9
AMBP	CP	HPR	SAA1
APOA2	CPN2	HPX	SAA2
APOA4	CRP	IGFALS	SAA4
APOB	CRTAC1	IGHM	SERPINA1
APOE	DCDC1	ITIH1	SERPINA10
APOH	DEFA1	ITIH2	SERPINA3
APOL1	EFEMP1	ITIH4	SERPINA5
C1R	F11	KLKB1	SERPINA6
C1S	F12	KNG1	SERPINC1
C2	F2	LRG1	SERPIND1
C3	F9	MASP1	SERPINF2
C4A	FCN2	MASP2	SERPING1
C4B	FCN3	ORM2	THBS1

C5	FGA	PLG	TTR
C8A	FGB	PON1	VTN
C8B	FGG	PRG4	VWF
C9	FGL1		

3. Validation of the LGALS3BP-spike association actually appears to be merely a confirmation that the serum of ICU patients with COVID-19 had higher levels of LGALS3BP than controls or sepsis patients. This does not confirm a functional role for LGALS3BP in the spike glycoprotein complex. Some orthogonal measurement on the pull-downs should have been shown, whether Western or ELISA.

In the revised version, we have now added functional data to demonstrate the antiviral activity of LGALS3BP (see new **Fig. 6**) further demonstrating that LGALS3BP is not merely a contaminant. We agree that the observation that a greater level of LGALS3BP was co-isolated with spike from the plasma of COVID-19 patients when compared to sepsis patients is most likely due to the higher levels of LGALS3BP present. However, we would argue that this is an additional confirmatory experiment to demonstrate a dose-response effect for the interaction between LGALS3BP and spike. If this observation were only due to non-specific binding, we would also expect many of the other proteins elevated in COVID-19 to be pulled down to a greater extent in COVID-19-positive plasma, which is not the case. Additionally, as we pointed out above, if LGALS3BP had not been pulled down specifically by spike, then we would have observed this in our bead-only control samples which were performed for every biological replicate.

4. The investigation of PTX3 as a prognostic factor for survival appears to come solely from prior work by the authors and others; no data on the abundance of PTX3 in either plasma or serum from the KCH and GSTT cohorts is shown in the main manuscript. Given the data that are shown in Supplementary Material [Figure S6b], PTX3 appears to be a rather non-specific marker of ‘sick enough to be in the ICU’. Therefore, its inclusion in the machine learning models, and the significance of its performance in the machine learning models, is highly doubtful – it may simply be an indicator of being on a ventilator. The authors need to be very up front with these observations in the main manuscript.

Regarding data presentation in the main manuscript:

We thank the reviewer for bringing this to our attention. We agree with reviewer’s critique:

- We now present the ELISA measurements of PTX3 alongside RAGE as revised **Fig. 7b**.
- We now present the finding of PTX3 being among the best predictors for poor outcome among 1,526 plasma proteins, including many cytokines and chemokines, in the revised **Fig. 7a** (Filbin et al *bioRxiv* 2020: <https://doi.org/10.1101/2020.11.02.365536>).

Regarding selection of PTX3:

As specified in the machine learning (ML) methods section, we would like to clarify that the investigation of PTX3 as a prognostic factor is based on the statistical significance test. Our previous work on PTX3 in sepsis (Cuello et al *Mol Cell Prot* 2015: <https://doi.org/10.1074/mcp.M114.039446>; Burnap et al *Mol Cell Prot* 2021: <https://doi.org/10.1074/mcp.RA120.002305>) and that by others (Filbin et al *bioRxiv* 2020: <https://doi.org/10.1101/2020.11.02.365536>) corroborates this finding. We recognize that machine learning (ML) approach using baseline measurements upon ICU admission to predict 28-day mortality, as specified in online methods section, is not referred in main manuscript. Hence, we have now included the reference to this approach and included a ML methods illustration in **Supplementary Fig. 11**.

Regarding specificity of PTX3:

As shown in **Fig. 7b**, PTX3 baseline comparison in GSTT cohort (serum, $n = 62$) highlights that non-survivors have significantly higher PTX3 levels (P value 0.018 with age and sex correction), thus presenting it as a putative marker of mortality. All COVID-19 ICU patients were on ventilators.

5. The concerns about the promotion of PTX3 as a prognostic factor for clinical outcome in COVID-19 are exacerbated by the lack of any experimental detail regarding the ‘machine learning’ method used to

determine the efficacy of 'Age, PTX3' as a signature of COVID-19 mortality. The entire description of the machine learning exercise appears to be contained in Legends to Supplementary Figure 8 and Supplementary Table 9, with no justification given for use of the SVM RBF model. It is also not precisely clear how the 'leave one out validation' was conducted, as the main text simply refers to 'the external validation cohort of hospitalized COVID-19 patients described above'. It is not clear whether this refers to the identification of PTX3 as a prognostic marker, to the identification of an SVM signature, or both.

Regarding inclusion of ML in the main manuscript:

Once again, we are grateful to the reviewer for highlighting this. As mentioned in our response above, the machine learning (ML) method is described in the methods section provided after the references. However, we acknowledge that the main manuscript does not provide any reference to the methods section of machine learning. We have now addressed this by including an explanation in the main manuscript and **Supplementary Fig. 11** illustrating the steps towards final selection of prognostic marker(s). We hope that this sufficiently clarifies the feature and model selection process.

Regarding justification of SVM RBF:

We would like to draw the attention of the reviewer to the methods section that states "SVM uses hyperplane (decision surface) leveraging only a percentage of training samples (support vectors), thus offering high generalization ability attributed to its near impervious characteristic to new samples⁸²."

Regarding 'leave one out validation':

Leave-one-out validation was conducted on the discovery cohort, *i.e.* GSTT and KCH combined dataset. Leave-one-out is a special case of k-fold cross validation where k, *i.e.* number of folds is equal to the sample count. Thus, each sample is used once in the test set while the remaining samples shape the training set.

Regarding external validation cohort:

'*The external validation cohort of hospitalized COVID-19 patients described above*' refers to the study by MGH Emergency Department COVID-19 Cohort (Filbin et al *bioRxiv* 2020: <https://doi.org/10.1101/2020.11.02.365536>). The finding regarding PTX3 is now highlighted as Volcano plot in **Fig. 7a**.

Reviewer #4 (Remarks to the Author):

In “SARS-CoV-2 RNAemia and proteomic biomarker trajectory inform prognostication in COVID-19 patients admitted to intensive care” Gutmann et al describe the analysis of viral RNA and circulating proteins in blood samples collected from intensive care patients suffering from COVID-19. While the study provides valuable insights into longitudinal molecular phenotypes of the disease, there are, unfortunately, some substantial shortcomings that need to be addressed.

As a reviewer and first-time reader of the manuscript, I found the work to be rather erratic and an assembly of possible routes to analyse the available samples and data. It remained unclear what the real value and contribution of the study would be and how others could learn from the findings. Even though the topic is timely, the team of renown expertise, and the presented data of good quality, I missed a coherent message. The work as whole was often presented as an assembly of different projects and capabilities. Many different types of analyses have been done, there are two inconsistent sample sets (one serum, one plasma), but no common core hypothesis to centre the project(s).

We thank the reviewer for acknowledging the timeliness of the topic and the good quality of the data. As pointed out by the other reviewers, unique aspects of our study are 1) the investigation of the relationship between RNAemia with COVID-19 clinical outcomes, 2) the association with proteomic clusters to provide insights on downstream effects of RNAemia and clinical outcomes and 3) the pull-down experiments to interrogate physical interaction of SARS-CoV-2 spike glycoprotein and host proteins. Our core hypothesis is that circulating SARS-CoV-2 is a predictor for poor outcome and that associated plasma protein changes reveal causative pathways.

Following the reviewer’s criticism, we have rewritten parts of the Discussion, focusing on plasma proteins that interact with SARS-CoV-2, in particular activators of complement pathways and LGALS3BP:

Both MBL2 and PTX3 were among the best predictors for poor outcome. MBL2 binds to glycoproteins on the viral surface and is a key molecule for the *lectin pathway* of complement activation. PTX3, as a pattern recognition receptor, is known to form complexes with MBL2 (<https://doi.org/10.1074/jbc.M110.190637>) and to bind to viruses, including coronaviruses (<https://doi.org/10.1038/labinvest.2012.92>, <https://doi.org/10.4049/jimmunol.180.5.3391>). The MBL2/PTX3 complex directly activates the *classical complement activation*. Thus, complement activation via the innate immune response (MBL2/PTX3) is a key component of the overreactive immune system in COVID-19 ICU patients. Clinical trials testing complement inhibitors in COVID-19, such as the C5 inhibitor zilucoplan ([ClinicalTrials.gov](https://clinicaltrials.gov) Identifier: NCT04382755), are currently ongoing, and this is now mentioned in our discussion.

We have also added a new **Fig. 6** with functional data demonstrating the antiviral activity of LGALS3BP against SARS-CoV-2. LGALS3BP is another protein of the innate immune response and was identified by proteomics as potential interaction partner of SARS-CoV-2 spike glycoprotein. We now demonstrate that the expression of LGALS3BP lowers the uptake of SARS-CoV-2 spike glycoprotein and reduces syncytia formation, which we have recently used as a functional readout for identifying novel treatment targets for COVID-19 (Braga et al, *Nature*, in press).

Thus, our revised version conveys a more coherent message as requested by the reviewer.

Another concern were the different levels of care taken to choose appropriate statistical methods. From simple t-tests to SVM approaches, there is no good alignment and justification for using these. In particular the proteomics part suffers from choosing the simplest but certainly not most appropriate tools, check for normality and account for the small sample size versus the larger number of analytes. I remain particularly critical to using static correlation analysis tools to infer causality, and not include informative co-variables into association by LMMs (or similar) to learn more about what the data actually adds on top of age, sex, CRP, ALT or other abundant blood proteins. Moreover, the dichotomous analyses suffer greatly from imbalanced group sizes (eg. 18 vs 60), making findings less likely to be replicated elsewhere. It is good to correct p-values for multiple testing, but please include also those from the correlation analyses.

We are grateful to the reviewer for highlighting these points. All the details, corrections and additional analysis are now described in detail in the supplemental methods part of the manuscript with the most relevant information being included in the main manuscript.

Regarding normal distribution of data:

Non-parametric testing is consistently applied in all statistical comparisons of the manuscript except the analysis of the SARS-CoV-2 spike glycoprotein pulldown using plasma from COVID-19 ICU patients and non-COVID-19 sepsis ICU patients. In the latter case student t-test was used because of relatively low sample size ($n < 10$). We have now stated this rationale for t-test in spike pulldown data in the methods section.

From simple t-tests to SVM approaches, there is no good alignment and justification for using these.

Statistical significance tests along with correlation were used for exploratory analysis. SVM was used on baseline clinical, RNAemia and ELISA protein measures to enable non-linear learning towards the 28-day mortality prognosis prediction. Statistical significance test was used for singleton variables feature filter into SVM. We have now included **Supplementary Fig.11** to illustrate this. Proteomics repeated measures inferential analysis was conducted using linear mixed effects model. This is now specified in methods section.

Regarding proteomics small sample size versus the larger number of analytes.

In total, we analyzed serial samples from 74 COVID-19 ICU patients by proteomics. For comparison, here are the n-numbers from other proteomics studies that included COVID-19 patients with severe disease:

Shen et al, Cell , 2020:	n=28 severe
Messner et al, Cell Systems , 2020:	n=15 severe (WHO-5 to 7)

Regarding causality:

We now clearly state that correlation analysis cannot infer causality.

Regarding justification of statistical methods:

LMMs were used to explore the interaction survival and time of measurement and RNAemia and time of measurement correcting also for age and sex. Anti-SARS-CoV-2 antibody data and trajectories of protein clusters were fitted using Generalized Alternative Models (GAM), with *P* values reporting the effect of RNAemia or mortality in the model since the trajectories of the Anti-SARS-CoV-2 have been demonstrated to be non-linear while the same was observed for the protein clusters.

Regarding clinical co-variates:

Apart from age, none of the clinical variables were associated with 28-day mortality in COVID-19 ICU patients. Applying corrections for additional clinical parameters such as CRP etc. was not feasible because of the lack of some of these measurements at all timepoints. We now include an additional **Supplementary Fig. 5** and **Rebuttal Fig. 1-4** for the correlation of all significant proteins with clinical co-variates at baseline.

Rebuttal Fig. 1: Correlation of baseline clinical variables with serum proteins associated with COVID-19 (compared to sepsis and control patients) and proteins associated with 28-day ICU mortality in COVID-19 ICU patients (n = 62).

Rebuttal Fig. 2: Correlation of baseline clinical variables with serum proteins associated with COVID-19 (compared to sepsis and control patients) and proteins associated with 28-day ICU mortality in COVID-19 ICU patients (n = 62).

Rebuttal Fig. 3: Correlation of baseline clinical variables with serum proteins associated with COVID-19 (compared to sepsis and control patients) and proteins associated with 28-day ICU mortality in COVID-19 ICU patients (n = 62).

Rebuttal Fig. 4: Correlation of baseline clinical variables with serum proteins associated with COVID-19 (compared to sepsis and control patients) and proteins associated with 28-day ICU mortality in COVID-19 ICU patients (n = 62).

Regarding imbalanced group sizes (e.g. 18 vs 60) less likely to be replicated elsewhere:

Imbalanced group size in our study reflects the nature of COVID-19 ICU outcome, *i.e.* non-survivors *versus* survivors in the two hospitals, *i.e.* KCH and GSTT. As mentioned in the methods section, to ensure that the support vector machine (SVM) learning is not biased towards the minority class, we have applied SVM SMOTE. Further we have applied cross-validation to facilitate model generalization and restricted the biomarker combinations to a maximum of triplets to avoid the risk of overfitting. Nonetheless, further studies will be required to assess this strong associations of age, RNAemia and PTX3 with 28-day mortality in COVID-19 ICU patients.

Regarding multiple testing:

Corrections for multiple testing was applied for differential expression, functional enrichment, correlation and survival analysis.

For the proteomics part, and as other MS-based COVID-19 proteomics studies have indicated, most of the proteins altered in the circulation as a response to SARS-CoV-2 infections are derived from the liver. Even though proteins of the coagulation system have been highlighted, their main source is the liver and their activity and abundance can be influenced by heat treatment (precipitation).

Regarding liver origin:

The liver is indeed an important source of plasma proteins, in particular for proteins in cluster 4 of **Fig. 4c** (see response below). However, other proteins identified in this study are not of hepatic origin: For example, we have recently demonstrated that PTX-3 is released by neutrophils in response to endotoxin and deposited in the vasculature, thereby providing a link between systemic and vascular inflammation (Burnap et al, *Mol Cell Proteomics* 2021: <https://doi.org/10.1074/mcp.RA120.002305>). Similarly, LGALS3BP is most highly expressed in the lung (Loimaranta et al, *J Leukoc Biol* 2018: <http://doi.org/10.1002/JLB.3VMR0118-036R>).

Regarding heat treatment (precipitation):

In contrast to other MS-based COVID-19 proteomics studies, no heat treatment was used in our experiments for inactivation. Instead, we have added the WHO recommended detergents. The detergent mix was added to all serum/plasma samples, including controls. Control and disease samples were processed together. The detergents solubilize proteins and avoid precipitation.

From my point of view, it would have added more value to more rigorously compare protein levels with the clinical data that otherwise remained hidden in the supplementary. To properly judge these relationships,

qq-plots should be provided for all variables with all proteins. From there, thorough power calculations will lead to more valuable lists.

As mentioned above, following the reviewer's suggestion, we have generated correlation plots between significant proteins and clinical variables for all the associations where the FDR corrected q -value of the correlation was below 0.05 (new **Supplementary Fig. 5**).

Longitudinal analysis is challenging but given that most patient will receive treatment adjusted to their health status, time as a variable is less appropriate and most likely overruled by other factors (eg medication dose, ventilation, days in bed/ICU, comorbidity). All figures of Fig 4 should therefore also include the clinical traits and consider these as informative components. It may well be that the two groups listed in cluster 4 of Fig 4c are simply reflecting age+sex instead of survival.

With regards to treatment:

One of the strengths of our study was the fact it included only ICU patients. All of them were on ventilator support and treated according to standardized protocols. The GAM model analysis performed to compare the trajectories of patients who survived and died has been corrected for age and sex. Changes in the main manuscript and supplementary material text reflect the additional analyses and clarifications conducted for the reviewer.

With regards to cluster 4 in Fig. 4c:

We are currently performing further experiments integrating microRNA and protein measurements to further interpret the revealed protein clusters in COVID-19. Expanding on the reviewer's previous point about proteins of hepatic origin, we present the measurements of a liver-specific microRNA, miR-122. MiR-122 levels are similar in plasma and serum (see <https://doi.org/10.1093/eurheartj/ehw146>).

[redacted]

Rebuttal Fig. 5: miR-122 levels in COVID-19 ICU survivors and non-survivors.

While the IP experiments of the spike protein is interesting, there are a few issues that need to be solved. First, a crapome (see PMID: 23921808) needs to be defined as many of the proteins found may likely be common and abundant contaminants. Others like PMID: 31171813 used plasma instead of cell lysates and illustrated the advantage of z-scores calculated from a large number of IPs as a measure of enrichment over a population of common contaminants.

We appreciate that the reviewer concurs with the other reviewers regarding the novelty of the pulldown experiments for SARS-CoV-2 spike protein conducted. A limitation of the CRAPome database for its utilization in our study is the lack of reported plasma/serum-based experiments. However, as advised, we used the CRAPome repository containing control spectral counts for all human studies in the database (v1.1, $n = 411$). We initially filtered this list to retain proteins only identified in 20% or more experiments (83/411 experiments), resulting in a list of 744 proteins. This list of proteins was then cross-referenced against the 28 proteins we report to be significantly enriched in Spike pulldowns, resulting in an overlap of 4 proteins, which, perhaps non-surprisingly, were cytoskeletal and histone-related proteins (ACTB, FLNA, MYH9 and HIST2H2AC). The 4 suspected contaminant proteins have been excluded from **Fig. 5** and **Supplementary Table 6**. Furthermore, as another example, LGALS3BP was only identified in 25/411 experiments (6.1%), a frequency which we would deem not to equate to the classification of LGALS3BP as a common contaminant. **Rebuttal Table 1** and **2** show the protein contaminants identified in SARS-CoV-2 spike glycoprotein pulldowns; and the “Crapome” proteins, respectively.

Similarly, the key hit from our pulldown – LGALS3BP – was cross-referenced against the supplementary tables provided in the above mentioned publication (<https://doi.org/10.1038/s41598-019-43552-5>) where it was observed that LGALS3BP was enriched as an off-target co-isolate upon heat treatment: a procedure that we did not perform. Importantly, LGALS3BP was never listed as being enriched in the control arm of the reported 434 IP-MS experiments.

Due to the CRAPome repository lacking experiments conducted in human plasma and serum we also generated our own “crapome” (**Rebuttal Tables 1** and **2**). We performed several controls to ensure that contaminants are excluded and spike interactome identified is genuine as follows:

- 1) Exogenously introduced spike glycoprotein preparation could have contained contaminants carried over from the HEK293 cells in which spike was expressed. To eliminate this, we have used a duplicate “mock” pulldown of a spike-only preparation and identified 96 proteins which could have been carried over from the spike preparation. Note that we applied a stringent threshold, so any proteins enriched less than five times in our plasma pulldowns compared to the “mock” spike control were excluded from subsequent analyses. We have now included an additional **Rebuttal Table 1** to list all the carryover proteins which we excluded from analyses.
- 2) Magnetic bead pulldown leads to non-specific isolation of proteins binding to the solid phase. To eliminate this confounder, we have performed a control, bead-only plasma incubation without adding exogenous spike. This was done for each of the eight biological samples providing internal control for every biological replicate. Hence, we have identified the “crapome” in our own experiments which we have now included in the **Rebuttal Table 2**. This combined with a sample size of $n = 8$, provides solid evidence that the findings are not merely a contamination issue.

Secondly, a mock protein needs to be used instead of bare beads to reflect a bait similar to spike.

[redacted]

However, the selection of a mock protein is challenging. It is difficult to have a similar bait to that of SARS-CoV-2 spike glycoprotein. For instance, an example protein to select would be the extracellular region of another viral envelope protein that is also glycosylated. The addition of such a protein may result in similar immunoglobulin enrichment as we observe in **Fig. 5a** and therefore the subsequent pulldown of complement factors and also lectin-binding proteins, such as LGALS3BP. In this circumstance, we believe bead-only controls act as an even more stringent control due to the beads being more likely to bind proteins non-specifically with the lack of a His-tagged protein to bind. Thus, we believe a mock protein does not actually act as the best control in these circumstances, and we included the PSCK9 experiment in the rebuttal for the reviewer but not in the main manuscript.

Thirdly, dilution series of serum/plasma samples need to be conducted to learn more about the dynamic of the interaction and whether enrichment is based on the very high abundance of LGALS3BP or the more selective affinity.

We would argue that our observation that a greater level of LGALS3BP was pulled down from COVID-19 plasma supplemented with SARS-CoV-2 spike glycoprotein compared to sepsis and control patients' plasma supplemented with spike, in essence acts as a dose-response and highlights a specificity in the interaction between LGALS3BP and spike protein. In support of this, if this observation were only due to the elevated LGALS3BP in COVID-19, we would also expect many more of the other proteins elevated in COVID-19 to be pulled down, which was not the case. Additionally, as we pointed out in the response to the previous point, if LGALS3BP had not been pulled down specifically, then we would have observed this in our bead-only control samples which were performed for every single biological replicate. In the revised version, we have now included functional experiments demonstrating that LGALS3BP impairs SARS-CoV-2 spike-mediated cell-cell fusion and pseudoparticle entry, further supporting the specificity of our pulldown experiment.

Forth, it was unclear if heat treatment of the samples has induced the reported interaction. Again, PMID: 31171813 may provide some helpful leads on this matter.

None of our samples were heat treated. This is now clearly stated in the methods section "Inactivation of serum and plasma" as "Heat treatment was not performed to avoid protein precipitation".

It remained unclear why serum or plasma samples have been collected at the centres and why no common sample processing protocols have been agreed upon prior to the analyses. The two preparation types must be regarded as different and cannot serve to compare collections that use either of these. It remains impossible to state whether the phenotypes were different due to disease or sample collection.

For the proteomics, we always compared plasma and serum separately.

- We utilized plasma samples to determine a COVID-19 related protein signature. **Fig. 3a** is based on the comparison of controls (plasma, $n = 30$), Sepsis (plasma, $n = 12$), KCH COVID-ICU (plasma $n = 12$).
- We determined whether the proteins within this signature were predictive of 28-day outcomes in the serum of patient samples from the larger GSTT serum cohort. **Fig. 3c** is based on the comparison of GSTT COVID-ICU serum samples ($n = 62$ patients, 240 samples).

All samples used for proteomics were processed in parallel, including all receiving the same detergent-based inactivation. For comparison, here are the n-numbers from other proteomics studies that included COVID-19 patients with severe disease:

Shen et al, Cell , 2020:	n=28 severe
Messner et al, Cell Systems , 2020:	n=15 severe (WHO-5 to 7)

The use of two sample matrices from COVID-19 patients can also be considered a strength of our study. Both plasma and serum collection are common clinical practice. It is important if protein levels are consistent across both matrices, *i.e.* for PTX3 we observed a clear matrix effect, which has gone unnoticed in previous publications using only one matrix (see Brunetta et al, *Nat Immunol* 2021: <https://doi.org/10.1038/s41590-020-00832-x>). Similarly, a recent COVID-19 proteomics study placed on a preprint server also utilized this approach whereby a discovery cohort utilizing plasma led to the use of serum in a separate validation cohort (preprint available at <https://doi.org/10.1101/2020.11.09.20228015>).

PTX3 was determined to be one of the best prognostic markers in an external Olink proteomics validation dataset utilized in our study (preprint available at <https://doi.org/10.1101/2020.11.02.365536>). This dataset was generated from *plasma*. We see a similar strength in survival prediction by PTX3 in the *serum* samples of GSTT patients adding strength to PTX3 as a biomarker for COVID-19 survival. These findings are now highlighted in the revised **Fig. 7a and 7b**.

Further, it was unclear if only the COVID-19 samples were heat treated. As has been shown by others, heat treatment affects the protein composition and deactivates not only the virus' infection potential but also the components of the coagulation/complement system. Hence, a comparison between COVID19, sepsis, and pre-pandemic samples may, in addition to the phenotype, be further influenced and biased by sample pre-treatment. Age of sample (time in the freezer) is another important aspect to consider in such cross-sectional analysis (PMID: 31573204).

Regarding heat treatment:

None of the samples within our study were heat treated. As described within the methods, to eliminate potentially active viral particles samples, all samples, including control and sepsis patient samples, were treated with detergent, as per the guidance of the CRICK COVID-19 consortium:

“Samples destined for protein analysis were inactivated by addition of 1% (v/v) Triton X-100 (Sigma, T8787) and 1% (v/v) tributyl phosphate (Sigma, 00675), followed by 15 s of vortexing and 4 h incubation at room temperature.”

Thus, there is no bias of pre-treatment. To avoid batch effects, all proteomics samples were processed together.

Regarding age of samples (time in the freezer):

Plasma and serum samples were kept at -80°C for less than one year. At -80°C, storage effects within less than a year are negligible, especially for the relatively abundant plasma proteins detected by DIA-MS. Patient recruitment of the “pre-pandemic, non-COVID-19 ICU sepsis” cohort occurred between October 16, 2019 and February 26, 2020; whilst recruitment of the “pre-pandemic, non-COVID-19 control” cohort occurred between July 8, 2019 and September 9, 2019. This is now clarified in the revised manuscript.

While the KM plots indicate some added value of the discussed variables, it remains unclear what PTX3 and RNAemia add to age, sex, etc. It is preferred to conduct conditional analysis than combinatorial models adding in variables for the best apparent performance. It is more important to understand the independent contribution of the variables and learn about the mechanism than conducting a race for the best mathematical AUCs.

Regarding conditional analysis:

We are taking an ensemble approach whereby feature filter is applied for singleton variables followed by a wrapper method to these shortlisted singleton features. We recognize that this may not have been clear from the description in our methods section. Hence, we have now illustrated this in a **Supplementary Fig. 11**.

In the wrapper method a deterministic search procedure is applied in the space of binary and triplet feature subsets for each of the shortlisted singleton feature from the previous step, *i.e.* feature filter. With 27 clinical variables, RNAemia and 3 ELISA protein measurements, this study offers a finite search space for both binary and triplet feature subsets. As mentioned in methods section, combinations were restricted to a maximum of triplets to enhance ease of clinical implementation and avoid the risk of overfitting. This deterministic search also enabled us to look at the shortlisted features and its combinations both from clinical and classification performance perspective.

We also explicitly looked at the correlation of shortlisted binary signatures, *i.e.* ‘Age, RNAemia’ and ‘Age, PTX3’. As illustrated in Fig 2c, Age and RNAemia do not have significant correlation. We recognize that we have not shared correlation of proteins with clinical variables. This is now illustrated in **Suppl. Fig. 5** and shows that PTX3 and age don’t have significant correlation.

Regarding understanding the individual variable contribution:

We would like to draw the attention of the reviewer to the independent contribution of the features as shown in **Supplementary Table 8**. Amongst the statistically significant variables ($P < 0.05$, **Fig. 7b, Supplementary Table 3**), RNAemia emerged as the best individual predictor based on the average of sensitivity, positive predicted value (PPV) and ROC AUC. As rightly pointed out by the reviewer, this study data is imbalanced reflecting the true patient scenario. Hence, for model evaluation we have used the average of sensitivity, PPV and ROC AUC, as illustrated in **Supplementary Table 8**. While F1-score, *i.e.* harmonic mean of sensitivity and precision (PPV) is a commonly used evaluation metric for imbalanced data, the drawback is that F1-score does not reflect the correct classification of the majority class, *i.e.* true negatives. Combining ROC AUC along with sensitivity and PPV addresses this limitation of standalone usage of F1-score. Additionally, we have used SVM Synthetic Minority Oversampling Technique (SMOTE) to prevent learning bias of SVM RBF towards the majority class.

While RNAemia was the best single predictor, its sensitivity was low (**Supplementary Table 8**). Age was the next best singleton predictor, however PPV of 44% demonstrates low probability confidence in predicting mortality, *i.e.* the positive class. In binary setting, two signatures emerged with comparable classification performance: 'Age, RNAemia' and 'Age, PTX3' (**Supplementary Table 8**). Although the triplet combination of age, FiO₂ and RNAemia achieved a ROC of ~86% with a sensitivity of 72.22% and a specificity of 88.33% (**Supplementary Table 8**), the gain in PPV was nominal with no uplift to specificity when compared to 'Age, RNAemia' suggesting the binary combination to be an optimal signature to choose.

Supplementary Fig. 12 illustrates the technical validation of the 'Age, RNAemia' model based on SVM RBF - a permutation test for statistical significance of the classifier performance, and stability of feature importance in an alternate machine learning feature ranking model, *i.e.* Random forest with resampling.

It also remains unclear why PTX3 was included into this story after it has not been detected in the MS-based proteomics. Benchmarking the work with the publicly data sets is good but again, this has further blurred the manuscript's message.

We agree with the reviewer. We now include a graphical representation of the Olink external dataset (**Fig. 7a**), highlighting all proteins measured and their association with 28-day outcome, in which PTX3 can be clearly seen to be highly significantly associated with outcome. Alongside, we now include the PTX3 levels in COVID-19 survivors and non-survivors as well as the comparison of PTX3 levels in control, sepsis and COVID-19 patients within the main figure prior to showing the KM plots (**Fig. 7b**). PTX3 levels over time and association with RNAemia is shown in **Supplementary Fig. 10**.

Our aim was to benchmark RNAemia against the best protein biomarkers. We selected RAGE as an established biomarker for ARDS and PTX3 as a biomarker for survival in sepsis. Apart from our own work on PTX3 in sepsis (Please see: Cuello et al, *Mol Cell Prot* 2014: <https://doi.org/10.1074/mcp.M114.039446> and Burnap et al *Mol Cell Prot* 2021: <https://doi.org/10.1074/mcp.RA120.002305>), PTX3 was one of the most significant proteins to be positively associated with 28-day mortality in COVID-19 patients in external Olink dataset and recently confirmed as a good predictor of COVID-19 outcome (see Brunetta et al, *Nat Immunol*, 2021: <https://doi.org/10.1038/s41590-020-00832-x>). Thus, we thought it was justified to measure PTX3 alongside RAGE in our cohorts by ELISA, due to their abundance being too low to be consistently detected by our DIA-MS analyses.

Please avoid statements such as "To the best of our knowledge, this is the largest longitudinal assessment of RNAemia,... " as this main also be the only study and it does not increase the value given its pitfalls.

We follow the reviewer's advice and have deleted this statement.

Why did the authors stop providing p-values after S-table 4? These two groups seem to be very different in their demographics.

We thank the reviewer for this suggestion. We have now inserted the "non-COVID-19, non-ICU controls" (formerly **Supplementary Table 5**) into **Supplementary Table 4** together with the "Pre-pandemic non-COVID-19 ICU sepsis patients" and "Intra-pandemic non-COVID-19 ICU patients". *P* values for all three comparisons are now provided.

S-table 1: Among the clinical variables, ALT levels of survivors vs non-survivors also appear different but, surprisingly, have passed the significance threshold. Please provide visual representation and explanation of this.

As requested, here is the visualization of ALT baseline characteristics as specified in **Supplementary Table 1** with median, IQR and *P* value along with actual data points jitter by survivors and non-survivors.

Rebuttal Fig. 7: ALT baseline levels in COVID-19 ICU non-survivors and survivors.

Please add the number of samples to each group in all figures!

Thank you for pointing this out. We have now added the respective numbers.

All heat maps are only partially informative because they lack dendrograms. It remains impossible to judge distances between the sub-clusters, where and why the groups were categorized as highlighted.

We thank the reviewer for this comment. We added the missing dendrograms in **Fig. 3a**

and **4c**. **Please sort all HR plots per ratio and not the alphabet**

To aid readability, the HR plots were sorted in the same order as the baseline characteristics for COVID-19 ICU patients in **Supplementary Table 1** (demographics, co-morbidities, acute care parameters, blood biochemistry, and blood count parameter). Hazard Ratio plots in **Fig. 2a** and **2b** also align with the sequence of baseline characteristics in **Supplementary Table 1**. This is now explained in the figure legends. We hope this is acceptable to the reviewer.

Reviewers' Comments:

Reviewer #1:

Remarks to the Author:

The authors addressed my concerns appropriately. Thanks

Reviewer #2:

Remarks to the Author:

Certain limitations are still present (e.g., lack of viral RNA quantification, small Ns in the KCH cohort), but the results do add important information to the field. I have no additional comments.

Reviewer #3:

Remarks to the Author:

In this revised manuscript and the accompanying rebuttal letter, the authors have done a generally adequate job of clarifying issues brought up by all four reviewers if those issues were primarily related to clarification of methods and procedures, and in that respect, the manuscript is substantially improved. More substantial issues, such as the clinical relevance of sepsis versus ARDS as a control for COVID-19 and the decision to use serum for some proteomic studies and plasma for other proteomic studies have generally been answered in a defensive manner, with little acknowledgement of the impact that such concerns will have on the potential impact of the manuscript. In short, it is not simply sufficient to justify one's position to the reviewers; potential issues of experimental design affecting impact must be dealt with up front in the discussion portions of the manuscript. In general, such discussions were missing or superficial. For example, simply stating that some other group has used the same approach of discovery in plasma and validation in serum is not an adequate response, particularly since the reference is a non-peer-reviewed preprint.

One of the major findings of this manuscript is the association of LGALS3BP with the SARS-Cov2 spike protein in pull-down assays. Concerns regarding the specificity of this pull down reaction raised by two reviewers were addressed, with both a more detailed explanation of experimental design and the application of 'crapome' analysis to the pull-down data. This is an adequate response that is well accepted in the field, and the addition of Rebuttal Tables 1 and 2 are substantial additions to the manuscript. However, the gold-standard experiment would be a reciprocal pull-down using LGALS3BP as the bait and examining the presence of spike protein in the pull down. This experiment was not done, although the authors did develop expression vectors for the spike protein in the context of the functional studies provided as additional evidence of the significance of the spike protein-LGALS3BP interaction.

The ability of co-expressed LGALS3BP to block fusion of ACE2-expressing and spike protein expressing cells is highly interesting and a very useful addition to the work. However, there appears to be a bit of a missed opportunity, in the reliance on imaging of cell fusion as the sole read-out, rather than exploring methods (such as FRET) that would specifically confirm the presence of LGALS3BP-spike protein interactions in an orthogonal fashion.

The major disappointment with the resubmission is the relatively minor revisions to the Discussion, which do not appear to address the intent of comments from the clinical and statistical reviewers. For example, Reviewer 1's concerns about the distinct role of hyperinflammatory responses in sepsis versus ARDS should have been addressed in the Discussion, as well as the rebuttal letter, as it specifically relates to the potential relevance and impact of this work. There is still no clear clinical application called out for this work – what would be the clinical action generated by a result indicating poor survival? The suggestion that RNAemia positivity might motivate anti-viral therapies begs the

questions raised by reviewers 1 and 2 about whether RNAemia is simply a surrogate for high viral load. While the Discussion provides new information related to the known functions of identified proteins associated with the complement system, it does not really address the issue of whether the association of RNAemia with poor survival reflects the consequences of high viral load, or whether there are other pathological processes activated in poor surviving patients that results in a 'leaky vascular system', such that SARS-Cov2 RNA more easily enters the blood, independent of viral load. This is really the question at the core of this study, and the motivating factor behind identifying protein factors that associate with poor outcome. The use of the term 'downstream' in the authors' statement in the Discussion ['the association with proteomic clusters to provide insights on downstream effects of RNAemia and clinical outcomes'] suggests that the authors assume the RNAemia causes poor outcome, rather than serving as an indicator of other factors that are themselves pathological. A more clear statement of alternative interpretations would have been helpful.

Reviewer #4:

Remarks to the Author:

First, I would like to thank the authors for their efforts to revise their manuscript.

In response to my previous comments, I as reviewer #4, found the revised version - unfortunately - only marginally improved.

The main changes made to the manuscript (that concern my initial suggestions) were to modify the discussion instead of further streamlining the message. The authors added yet another (cell biology focussed) section into the clinically oriented manuscript. Rather than keeping the focus and work on a cohesive story with an inter-connected thread to highlight the main value of the work (= present clinical study), the authors further diluted the main message(s) they wanted to give. This became most obvious in the open ended abstract that left the reader with making up their own minds about how to interpret the different observations. I can understand if the other requests to modify the work may have been exhausting, and the pandemic affecting everyone, but I cannot provide a more positive verdict when judging the response to my comments. Again, I strongly suggest to restructure the results and keep a focus on the core analyses rather than adding yet another set of data that is not aligned.

Some more of the remaining concerns are listed below:

1) Focus and message

The authors responded: "Our core hypothesis is that circulating SARS- CoV-2 is a predictor for poor outcome and that associated plasma protein changes reveal causative pathways.Thus, our revised version conveys a more coherent message as requested by the reviewer. "

I cannot agree that the points of concerns have been addressed sufficiently and I cannot agree that the authors provide enough data to ground their findings into the causal pathways.

As another example, very little is mentioned about MBL2 in the results section but they expand on this protein in the discussion. It is the opposite for PTX3. Again, such inconsistencies in focus confuse the story: The authors start off with RNAemia, then add proteomics to predict outcome and present trajectories, then perform pull-down, then perform cell based assays to study the pull-down, and then switch back to outcome prediction with RNAemia but with another ELISA. While I do not doubt the quality of the data, the current manuscript still does not satisfy my request to present a coherent message that has a clear red thread and focus on the most valuable insights rather than adding in yet another possible analysis (and because the data exists).

Fig 7 is labeled as "Selection of protein biomarkers". Besides a strong wish to avoid the word "biomarker" I wonder if this puts all previous proteomics data into another less valuable category?

2) Clinical data in longitudinal samples

The authors responded: "CRP etc. was not feasible because of the lack of some of these measurements at all timepoints." This is a surprising statement as it indicates that they could not even detect CRP in the MS analyses and use this as a basis for further analyses (after correlating clinical data from baseline). CRP seemed to be detectable in the pull-down with blank beads (see Rebuttal Table 2). Rather than following the initial theme and strengthening the clinical data, the authors did not expand on the requested matter but included a section on cellular interactions. This is per se informative, but for me, this does not fit and address the concerns I raised before.

3) Association tests:

The authors responded: "We now include an additional Supplementary Fig. 5 and Rebuttal Fig. 1-4 for the correlation of all significant proteins with clinical co-variates at baseline." Thank you for adding this to the rebuttal but how does this support/address the main hypothesis? How did they anchor this information in the main manuscript? I would also have appreciated to see this data being managed with a little more care when it comes to the outliers (eg bilirubin value of 400, PTX value of 100) and consider other tools for binary variables (eg COPD). There are no units added to these clinical variables either, so it becomes difficult to compare these.

4) Tissue origins:

The authors responded: "However, other proteins identified in this study are not of hepatic origin." and present their interpretation on the expression of PTX3 and LGALS3BP. Considering the wealth of cross-tissue analyses, I recommend to use database that present a global view on protein expression. Using GTEx, for example, both PTX3 and LGALS3BP are expressed in many other tissues. See <https://www.gtexportal.org/home/gene/PTX3> and See <https://www.gtexportal.org/home/gene/LGALS3BP>

5) IP experiments:

The authors responded: "A limitation of the CRAPome database for its utilization in our study is the lack of reported plasma/serum-based experiments." I have read this with surprise because I cannot understand why they did not use the suggested study with plasma and the reported data on frequencies in plasma? Regarding the mock protein: Why did the authors not use VSV-G, other viral proteins from HIV, or the currently so widely available S and N domains from SARS-CoV-1, MERS or other endemic (corona)viruses? This could illustrate the LGALS3BP is either specific to SARS-CoV-2 or not. Using a human protein as a bait is of limited value.

6) Sample size of other studies:

I am sure the authors have witnessed the flood of MS, Olink and Somalogic based COVID-19 studies and it is impossible to keep references up to date when a whole community is working on a single topic. However, there are two that I would like the authors to comment on and include. <https://elifesciences.org/articles/64827> and <https://elifesciences.org/articles/65508> The two early

studies they mentioned are smaller in size and have been followed up already. See, for example, <https://www.medrxiv.org/content/10.1101/2020.11.09.20228015v1>

REVIEWER COMMENTS

Reviewer #1 (Remarks to the Author):

The authors addressed my concerns appropriately. Thanks.

We thank the reviewer for the positive comment.

Reviewer #2 (Remarks to the Author):

Certain limitations are still present (e.g., lack of viral RNA quantification, small Ns in the KCH cohort), but the results do add important information to the field. I have no additional comments.

We thank the reviewer for the positive comments.

Reviewer #3 (Remarks to the Author):

In this revised manuscript and the accompanying rebuttal letter, the authors have done a generally adequate job of clarifying issues brought up by all four reviewers if those issues were primarily related to clarification of methods and procedures, and in that respect, the manuscript is substantially improved.

We thank the reviewer for the positive comments.

More substantial issues, such as the clinical relevance of sepsis versus ARDS as a control for COVID-19 and the decision to use serum for some proteomic studies and plasma for other proteomic studies have generally been answered in a defensive manner, with little acknowledgement of the impact that such concerns will have on the potential impact of the manuscript. In short, it is not simply sufficient to justify one's position to the reviewers; potential issues of experimental design affecting impact must be dealt with up front in the discussion portions of the manuscript. In general, such discussions were missing or superficial. For example, simply stating that some other group has used the same approach of discovery in plasma and validation in serum is not an adequate response, particularly since the reference is a non-peer-reviewed preprint.

We are grateful for the reviewer's feedback.

Regarding sepsis versus ARDS as control:

We consider it a strength of our study that we include pre-pandemic sepsis ICU patients as additional control (Fig 3a). We have now included the following statement in the introduction:

"Sepsis is defined as organ dysfunction caused by a dysregulated host response to infection. As SARS-CoV-2 infection causes organ dysfunction (pulmonary and extrapulmonary), there is overlap in immunological changes between SARS-CoV-2 infection and sepsis, forming the rationale for using SARS-CoV-2-negative ICU sepsis patients as additional comparators."

We cite our recent work as well as work by others:

Shankar-Hari et al. *JAMA* 2016: <https://doi.org/10.1001/jama.2016.0289>

Singer et al. *JAMA* 2016: <https://doi.org/10.1001/jama.2016.0287>

Wilson, Shankar-Hari *Chest* 2021: <https://doi.org/10.1016/j.chest.2021.01.023>

Gupta A, et al, *Nat Med* 2020: <https://doi.org/10.1038/s41591-020-0968-3>

Laing et al. *Nat Med* 2020: <https://doi.org/10.1038/s41591-020-1038-6>

Furthermore, we have added a section to the discussion commenting on the use of sepsis patients over ARDS and the similarities between the two diseases:

“An argument could be made for using ARDS controls, due to similarities between non-COVID-19 ARDS and COVID-19 ARDS that we have previously reported^{3,48}. However, in addition to our rationale for non-COVID-19 sepsis controls highlighted earlier, all sepsis patients used as controls were mechanically ventilated⁴⁹; would meet the consensus definitions of ARDS⁵⁰; and would have been enrolled in clinical trials of ARDS⁵¹, since the most common etiology of ARDS is infection.”

Regarding plasma and serum:

We now highlight PROC and F7 in the Volcano plot of the external Olink validation cohort (revised Fig. 4a). These were the only proteins identified by our DIA-MS approach that were also quantified by the Olink Explore 1536 panel. The Olink measurements were performed in plasma. The DIA-MS analysis of the GSTT-cohort was done in serum. A reduction of PROC and F7 was significantly associated with 28-day mortality in both cohorts. The following statement has been added to the Discussion:

“The validation of the trajectory of differentially expressed proteins in COVID-19 and their association with outcome was done in serum samples of the larger GSTT COVID-19 ICU cohort (n = 62). The Olink measurements in the external validation cohort were performed in plasma (Fig. 4a). Validation in plasma and in serum ensures that the protein changes are independent of the sample type, which is important for the generalizability of findings. The Olink platform covered two (of ten) proteins (PROC, F7) associated with 28-day mortality measured by our DIA-MS approach, highlighting the complementarity of these different proteomics methods⁵⁴.”

One of the major findings of this manuscript is the association of LGALS3BP with the SARS-Cov2 spike protein in pull-down assays. Concerns regarding the specificity of this pull down reaction raised by two reviewers were addressed, with both a more detailed explanation of experimental design and the application of ‘crapome’ analysis to the pull-down data. This is an adequate response that is well accepted in the field, and the addition of Rebuttal Tables 1 and 2 are substantial additions to the manuscript.

We thank the reviewer for the positive comments.

However, the gold-standard experiment would be a reciprocal pull-down using LGALS3BP as the bait and examining the presence of spike protein in the pull down. This experiment was not done, although the authors did develop expression vectors for the spike protein in the context of the functional studies provided as additional evidence of the significance of the spike protein-LGALS3BP interaction. The ability of co-expressed LGALS3BP to block fusion of ACE2-expressing and spike protein expressing cells is highly interesting and a very useful addition to the work. However, there appears to be a bit of a missed opportunity, in the reliance on imaging of cell fusion as the sole read-out, rather than exploring methods (such as FRET) that would specifically confirm the presence of LGALS3BP-spike protein interactions in an orthogonal fashion.

Again, we thank the reviewer for this very positive feedback. We agree that the functional studies are a very useful addition to the work. Based on our findings for RNAemia, a reciprocal pulldown using LGALSBP as the bait would not retrieve sufficient quantities of viral spike protein to be detected by our MS approach. We agree with the reviewer that further work is needed to characterize the LGALS3BP-spike interaction, but we believe that exploring further methods such as FRET is beyond the scope of the current manuscript. We also try to accommodate the criticism of reviewer 4, who felt that the manuscript should focus on clinical findings rather than cell-based assays.

The major disappointment with the resubmission is the relatively minor revisions to the Discussion, which do not appear to address the intent of comments from the clinical and statistical reviewers. For example, Reviewer 1’s concerns about the distinct role of hyperinflammatory responses in sepsis versus ARDS should have been addressed in the Discussion, as well as the rebuttal letter, as it specifically relates to the potential relevance and impact of this work. There is still no clear clinical application called out for this work – what would be the clinical action generated by a result indicating poor survival? The suggestion that RNAemia positivity might motivate anti-viral therapies begs the questions raised by reviewers 1 and 2 about whether RNAemia is simply a surrogate for high viral load. While the Discussion provides new

information related to the known functions of identified proteins associated with the complement system, it does not really address the issue of whether the association of RNAemia with poor survival reflects the consequences of high viral load, or whether there are other pathological processes activated in poor surviving patients that results in a 'leaky vascular system', such that SARS-Cov2 RNA more easily enters the blood, independent of viral load. This is really the question at the core of this study, and the motivating factor behind identifying protein factors that associate with poor outcome. The use of the term 'downstream' in the authors' statement in the Discussion ['the association with proteomic clusters to provide insights on down-stream effects of RNAemia and clinical outcomes'] suggests that the authors assume the RNAemia causes poor outcome, rather than serving as an indicator of other factors that are themselves pathological. A more clear statement of alternative interpretations would have been helpful.

Regarding sepsis versus ARDS as control:

As mentioned above, we highlight the similarities and differences in sepsis *versus* ARDS in the revised manuscript.

Regarding viral load versus other pathological processes:

We have included a statement in the Result section:

"To explore whether the association of RNAemia with 28-day mortality may reflect distinct pathological processes, we identified proteins that associate with RNAemia at baseline (Fig. 6a) and over time (Fig. 6b)."

We have also added a statement on viral load in the Discussion by referring to the recent study by Fajnzyblber et al. (SARS-CoV-2 viral load is associated with increased disease severity and mortality. **Nat Commun.** 2020: <https://doi.org/10.1038/s41467-020-19057-5>):

"Correlation between nasopharyngeal viral load and plasma viral load was of moderate strength ($r = 0.32$), suggesting that the viral load in the nasopharyngeal compartment only accounts for a minor part ($r^2 = 10.2\%$) of the plasma variation. Thus, other pathological processes are likely to contribute to RNAemia, independent of viral load."

Reviewer #4 (Remarks to the Author):

First, I would like to thank the authors for their efforts to revise their manuscript.

In response to my previous comments, I as reviewer #4, found the revised version - unfortunately - only marginally improved. The main changes made to the manuscript (that concern my initial suggestions) were to modify the discussion instead of further streamlining the message. The authors added yet another (cell biology focussed) section into the clinically oriented manuscript. Rather than keeping the focus and work on a cohesive story with an inter-connected thread to highlight the main value of the work (= present clinical study), the authors further diluted the main message(s) they wanted to give. This became most obvious in the open ended abstract that left the reader with making up their now minds about how to interpret the different observations. I can understand if the other requests to modify the work may have been exhausting, and the pandemic effecting everyone, but I cannot provide a more positive verdict when judging the response to my comments. Again, I strongly suggest to restructure the results and keep a focus on the core analyses rather than adding yet another set of data that is not aligned.

Following this reviewer's request, we have modified the abstract and restructured the results. All clinical findings are now presented before the cell biology experiments. However, we do believe it is a strength our manuscript that we are not only associating RNAemia and proteins with clinical outcome but also perform mechanistic experiments by identifying binding partners to spike protein and subsequently demonstrating antiviral activity for SARS-CoV-2. We are not aware of any proteomic study to date that has done similar mechanistic follow-up experiments or compared protein *versus* RNAemia measurements. The additional cell-based work was appreciated by reviewers 1-3. We also provide a revised discussion to focus our message. We thank the reviewer for the comments, which helped to improve the manuscript.

Some more of the remaining concerns are listed below:

1) Focus and message

The authors responded: “Our core hypothesis is that circulating SARS- CoV-2 is a predictor for poor outcome and that associated plasma protein changes reveal causative pathways..... Thus, our revised version conveys a more coherent message as requested by the reviewer. “ I cannot agree that the points of concerns have been addressed sufficiently and I cannot agree that the authors provide enough data to ground their findings into the causal pathways. As another example, very little is mentioned about MBL2 in the results section but they expand on this protein in the discussion. It is the opposite for PTX3. Again, such inconsistencies in focus confuse the story: The authors start off with RNAemia, then add proteomics to predict outcome and present trajectories, then perform pull-down, then perform cell based assays to study the pull-down, and then switch back to outcome prediction with RNAemia but with another ELISA. While I do not doubt the quality of the data, the current manuscript still does not satisfy my request to present a coherent message that has a clear red thread and focus on the most valuable insights rather than adding in yet another possible analysis (and because the data exists).

We have now restructured the results as requested and revised the discussion, expanding on the interaction of MBL2 and PTX3 and their synergistic effect on classical complement activation, independent of antibody-antigen complexes. Since anti-SARS-CoV-2 antibody levels were similar in patients who survived and died, the innate immune response (MBL2/PTX3) and its role in complement activation may play a more important role in determining outcome in COVID-19 ICU patients.

“The MBL2/PTX3 complex can directly activate the complement system independent of antigen-antibody complexes. Consistent with our previous results on PTX3 in sepsis patients^{31,69}, PTX3 also emerged as strong predictor for mortality in COVID-19 ICU patients. This is in agreement with other studies^{38,40,70} reporting strong associations of PTX3 with COVID-19 mortality⁷⁰. PTX3 is released from neutrophils upon activation^{69,71}, abundant in macrophages⁴⁰, but also highly expressed in lung and adipose tissue (<https://www.gtexportal.org/home/gene/PTX3>). While anti-SARS-CoV-2 antibody levels were similar in COVID-19 ICU patients who survived and died, MBL2 and PTX3 were associated with poor outcome pointing towards the importance of antibody-independent mechanisms of complement activation in COVID-19. Clinical trials with complement inhibitor are currently ongoing for COVID-19⁵⁸.”

Fig 7 is labelled as “Selection of protein biomarkers”. Besides a strong wish to avoid the word “biomarker” I wonder if this puts all previous proteomics data into another less valuable category?

We thank the reviewer for this feedback and have changed the labelling (now **Fig. 4**) to “*External protein marker validation and PTX3 selection*”. We avoid the word “biomarker” throughout the manuscript, including the title. Our main conclusion is that RNAemia was comparable in performance to the best protein predictors of 28-day mortality. This is now stated in the abstract. We also highlight PROC and F7 in the Volcano plot of the external Olink validation cohort (revised **Fig. 4a**). These were the only two proteins identified by our DIA-MS approach that were also quantified by the Olink Explore 1536 panel. A reduction of PROC and F7 was significantly associated with 28-day mortality in both cohorts.

2) Clinical data in longitudinal samples

The authors responded: “CRP etc. was not feasible because of the lack of some of these measurements at all timepoints.” This is a surprising statement as it indicates that they could not even detect CRP in the MS analyses and use this as a basis for further analyses (after correlating clinical data from baseline). CRP seemed to be detectable in the pull-down with blank beads (see Rebuttal Table 2). Rather than following the initial theme and strengthening the clinical data, the authors did not expand on the requested matter but included a section on cellular interactions. This is per se informative, but for me, this does not fit and address the concerns I raised before.

A misunderstanding has occurred. With “*the lack of some of these measurements at all timepoints*” we referred to the clinical CRP measurements (which were available only at baseline) and not our proteomics data. CRP was

consistently detected by proteomics in all timepoints. The revised **Supplementary Fig. 5** provides the correlation of the baseline clinical variables to the explored proteomics markers at baseline.

3) Association tests:

The authors responded: “We now include an additional **Supplementary Fig. 5** and **Rebuttal Fig. 1-4** for the correlation of all significant proteins with clinical co-variates at baseline.”

Thank you for adding this to the rebuttal but how does this support/address the main hypothesis? How did they anchor this information in the main manuscript? I would also have appreciated to see this data being managed with a little more care when it comes to the outliers (eg bilirubin value of 400, PTX value of 100) and consider other tools for binary variables (eg COPD). There are no units added to these clinical variables either, so it becomes difficult to compare these.

We have now anchored **Supplementary Fig. 5** in the main manuscript by further annotating the proteins with links to our main Figures.

We have stated in the methods section states that we use point-biserial correlation for correlations between continuous and binary variables; Cohen’s kappa for correlations between binary variables; and Spearman correlation for correlations between continuous variables. We have now also added this information to all figure legends where applicable.

In response to the reviewer’s request, we have inspected outliers visually in the correlograms and removed them for each correlation after performing the Grubbs test to confirm the presence of an outlier, as shown in the revised **Rebuttal Fig. 1-3** (previous Rebuttal Fig. 1-4) and revised **Supplementary Fig. 5**. Units for clinical variables are now also shown on these figures as requested. Thank you for this remark.

Rebuttal Fig. 1: Correlation of baseline clinical variables with serum proteins associated with COVID-19 (compared to sepsis and control patients) and proteins associated with 28-day ICU mortality in COVID-19 ICU patients (n = 62).

Rebuttal Fig. 2: Correlation of baseline clinical variables with serum proteins associated with COVID-19 (compared to sepsis and control patients) and proteins associated with 28-day ICU mortality in COVID-19 ICU patients (n = 62).

Rebuttal Fig. 3: Correlation of baseline clinical variables with serum proteins associated with COVID-19 (compared to sepsis and control patients) and proteins associated with 28-day ICU mortality in COVID-19 ICU patients (n = 62).

4) Tissue origins:

The authors responded: “However, other proteins identified in this study are not of hepatic origin.” and present their interpretation on the expression of PTX3 and LGALS3BP. Considering the wealth of cross-tissue analyses, I recommend to use database that present a global view on protein expression. Using GTEx, for example, both PTX3 and LGALS3BP are expressed in many other tissues. See <https://www.gtexportal.org/home/gene/PTX3> and See <https://www.gtexportal.org/home/gene/LGALS3BP>

Thank you for this helpful suggestion, we incorporated the GTEx links in the discussion. Apart from the lung, levels of expression of both PTX3 and LGALS3BP are also high in adipose tissue.

5) IP experiments:

The authors responded: “A limitation of the CRAPome database for its utilization in our study is the lack of reported plasma/serum-based experiments.” I have read this with surprise because I cannot understand why they did not use the suggested study with plasma and the reported data on frequencies in plasma?

We did take the suggested study into account, see previous rebuttal:

“Similarly, the key hit from our pulldown – LGALS3BP – was cross-referenced against the supplementary tables provided in the above mentioned publication (<https://doi.org/10.1038/s41598-019-43552-5>) where it was observed that LGALS3BP was enriched as an off-target co-isolate upon heat treatment: a procedure that we did not perform. Importantly, LGALS3BP was never listed as being enriched in the control arm of the reported 434 IP-MS experiments.”

However, we would like to stress that the pulldown conducted in our study is not an immunoprecipitation and relies upon the use of anti-HIS beads and cobalt-based chemistry for the interaction with HIS-tagged tagged Spike (Dynabeads™ His-Tag Isolation and Pulldown). If we were to take all proteins in the above paper with a z-score of >2.5 (Supplementary excel) in non-heat treated IPs, and filter for proteins defined as either OFF-TARGET, CONTROL or NO-TARGET, the list of potential contaminants includes 330 proteins. Cross-referencing this list

against our list of 19 Spike enriched proteins (excluding spike and immunoglobulins) would result in 5 proteins potentially being defined as contaminants: APCS, APOD, C4BPA, C4BPB and HABP2. However, the question remains as to whether these proteins are indeed contaminants or do bind Spike either directly or indirectly. Given the extensive controls we have included, the fact that the patients' immunoglobulins bind to Spike, and that no exogenous immunoglobulins were introduced in this experiment, we believe proteins such as complement regulators (C4BPA and C4BPB) may not be merely contaminants in this setting.

Regarding the mock protein: Why did the authors not use VSV-G, other viral proteins from HIV, or the currently so widely available S and N domains from SARS-CoV-1, MERS or other endemic (corona)viruses? This could illustrate the LGALS3BP is either specific to SARS-CoV-2 or not. Using a human protein as a bait is of limited value.

LGALS3BP is known to bind to certain viral proteins, *i.e.* from HIV, as mentioned in the discussion. Our experiments demonstrate for the first time that LGALS3BP can also bind to the Spike protein from SARS-CoV-2. To compare the interaction partners of the Spike protein from different coronaviruses is beyond the scope of the present study. Instead, we have streamlined the message of the current manuscript as requested by the reviewer.

6) Sample size of other studies:

I am sure the authors have witnessed the flood of MS, Olink and Somalogic based COVID-19 studies and it is impossible to keep references up to date when a whole community is working on a single topic. However, there are two that I would like the authors to comment on and include. <https://elifesciences.org/articles/64827> and <https://elifesciences.org/articles/65508> The two early studies they mentioned are smaller in size and have been followed up already. See, for example, <https://www.medrxiv.org/content/10.1101/2020.11.09.20228015v1>

Thank you for these suggestions. We have now included these studies in our discussion. We also highlight that the Olink Explore 1536 platform only covers two of the proteins associated with 28-day mortality as measured by our DIA-MS approach (**Fig. 3c, 3d, 4a**). Thus, we consider these two approaches as complementary, as reviewed by us recently:

Joshi A, Rienks M, Theofilatos K, Mayr M. Systems biology in cardiovascular disease: a multiomics approach. *Nat Rev Cardiol.* 2020 Dec 18. <https://doi.org/10.1038/s41569-020-00477-1>